# Impact of changes in climate and $CO_2$ on the carbon storage potential of vegetation under limited water availability using SEIB-DGVM version 3.02

Shanlin Tong[1,2,3], Weiguang Wang[2,3*], Jie Chen[1*], Chong-Yu Xu[4], Hisashi Sato[5], Guoqing Wang[6]

[1]State Key Laboratory of Water Resources and Hydropower Engineering Science, Wuhan University, Wuhan, 430072, Peoples R China

[2]State Key Laboratory of Hydrology-Water Resources and Hydraulic Engineering, Hohai University, Nanjing, 210098, Peoples R China

[3]Key Laboratory of Water Big Data Technology of Ministry of Water Resources, Hohai University, Nanjing, 210098, Peoples R China

[4]Department of Geosciences, University of Oslo, Oslo, N-0316, Norway

[5]Japan Agency for Marine-Earth Science and Technology, Yokohama, 236-0001, Japan

[6]Nanjing Hydraulic Research Institute, Nanjing, 210029, Peoples R China

*Ccorrespondence to: Weiguang Wang (wangweiguang2016@126.com); Jie Chen (jiechen@whu.edu.cn)

**Abstract**

Documenting year-to-year variations in carbon storage potential in terrestrial ecosystems is crucial for the determination of carbon dioxide ($CO_2$) emissions. However, the magnitude, pattern and inner biomass partitioning of carbon storage potential, and the effect of the changes in climate and $CO_2$ on inner carbon stocks, remain poorly quantified. Herein, we use a spatially explicit individual based-dynamic global vegetation model to investigate the influences of the changes in climate and $CO_2$ on the enhanced carbon storage potential of vegetation. The modelling included a series of factorial simulations using the CRU dataset from 1916 to 2015. The results show that $CO_2$ predominantly leads to a persistent and widespread increase in light-gathering vegetation biomass carbon stocks (LVBC) and water-gathering vegetation biomass carbon stocks (WVBC). Climate change appears to play a secondary role in carbon storage potential. Importantly, with the intensification of water stress, the magnitude of the light- and water-gathering responses in vegetation carbon stocks gradually decreases. Plants adjust carbon allocation to decrease the ratio between LVBC and WVBC for capturing more water. Changes in the pattern of vegetation carbon storage was linked to zonal limitations in water, which directly weakens and indirectly regulates the response of potential vegetation carbon stocks to a changing environment. Our findings

differ from previous modelling evaluations of vegetation that ignored inner carbon dynamics and
demonstrates that the long-term trend in increased vegetation biomass carbon stocks is driven by $CO_2$
fertilization and temperature effects that are controlled by water limitations.
**1 Introduction**
As a result of the changes in climate and atmospheric carbon dioxide ($CO_2$), the terrestrial ecosystem
carbon cycle exhibits remarkable trends in interannual variations, which induce uncertainty in estimated
carbon budgets (Erb et al., 2018; Keenan et al., 2017). Recent studies assessing interannual fluctuations
in terrestrial carbon sinks have shown that the land carbon cycle is the most uncertain component of the
global carbon budget (Ahlstrom et al., 2015; Piao et al., 2020; Jung et al., 2017; Humphrey et al., 2018;
Gentine et al., 2019; Humphrey et al., 2021). These uncertainties result from an incomplete understanding
of vegetation biomass carbon production, allocation, storage, loss, and turnover time (Bloom et al., 2016).
The extent and distribution of vegetation carbon storage is central to our understanding of how to
maintain a balanced land carbon cycle. Changes in terrestrial vegetation carbon storage have a significant
effect on atmospheric $CO_2$ concentrations and determine whether biomes become a source or sink of
carbon (Erb et al., 2018; Humphrey et al., 2018; Terrer et al., 2021). Therefore, investigating the
processes producing changes in carbon storage is key to improving the accuracy of estimated terrestrial
carbon budgets, and to tap the greenhouse-gas moderation potentials of vegetation (Ipcc, 2007; Roy et
al., 2001).

The atmospheric $CO_2$ concentration is affected by the vegetation carbon stock, while the long-term trend
of vegetation carbon storage capacity is also affected by the changes in climate and $CO_2$. Since the
beginning of industrialization, there has been a noticeable enhancement in the plant capacity of storing
and sequestering carbon, which is needed for stabilizing greenhouse gas concentrations and mitigating
global warming (Chen et al., 2019; Pan et al., 2011; Le Noë et al., 2020; Magerl et al., 2019; Bayer et al.,
2015; Harper et al., 2018). Due to the interaction between terrestrial vegetation and a changing
environment, both photosynthesis and respiration of the vegetation also changed. To better absorb $CO_2$
and sunlight required for photosynthesis, vegetated zones are gradually covered by vegetation with
higher plant height and wider leaf area (Erb et al., 2008). This change has coincided with a widespread

change in other vegetation features, including a positive increase in annual gross primary productivity and a greening of the biosphere (Madani et al., 2020; Zhu et al., 2016). The spatiotemporal distribution and environmental drivers in total carbon storage potential have been well documented on the basis of model estimates and satellite-based assessments (Erb et al., 2007; Erb et al., 2018; Bazilevich et al., 1971; Saugier et al., 2001; Bartholome and Belward, 2005; Olson et al., 1983; Pan et al., 2013; Ajtay et al., 1979; Ruesch and Gibbs, 2008; Kaplan et al., 2011; Shevliakova et al., 2009; Prentice et al., 2011; West et al., 2010; Hurtt et al., 2011). In contrast, the variability of inner components of carbon storage potential has not been extensively studied. Without an accurate assessment of the dynamics of each fraction, attribution of carbon storage potential to environmental drivers is highly uncertain. Consequently, partitioning potential vegetation carbon storage and revealing its inner processes are essential to accurately comprehend the current state of carbon storage capacity and reveal the influence of various drivers on the long-term trend of carbon storage potential.

The change of carbon storages in vegetation inner components is not only affected by environmental factors, but also controlled by allocation scheme of assimilated carbon. Fractional dynamics of the carbon stock are widely used as a key indicator to investigate the responses of vegetation to environmental drivers, which also reflect the response strategies of vegetation in environments with different water limitations (Yang et al., 2010). In arid regions, vegetation utilizes a tolerance strategy to allocate biomass, storing more biomass carbon in roots to resist enhanced water stress (Chen et al., 2013). Conforming to the optimal partitioning hypothesis, plants store more carbon in shoots and leaves in environments where water is more available and shift more carbon to roots when water is more limited (Yang et al., 2010; Mcconnaughay and Coleman, 1999). Water availability controls both carbon allocation and storage and can potentially transform zones characterized by a positive response to changes in climate and $CO_2$ to zones exhibiting a negative response. For example, global warming positively stimulates plant productivity (Keenan et al. 2017), while Madani et al. (2020) found that productively showed a negative response to temperature in tropical zones due to increasing water stress. With increased warming, water limitations are predicted to increasingly reduce the proportion of leaves' biomass, and decrease plant photosynthesis (Ma et al., 2021). Water limitations have a strong regulating effect on the spatial pattern of change in vegetation carbon storage, demonstrating the effects of the changes in climate and $CO_2$ on

the dynamics of the plant organs are affected by the terrestrial water gradient. Thus, it is important to
systematically investigate the distinct responses of carbon storage potential to changes in climate and
$CO_2$ under differing conditions of water stress.

As documented above, many studies have investigated the total changes in zonal and global terrestrial
storage of carbon, while few studies have examined trends in the components partitioning of vegetation
carbon storage. Large gaps in our knowledge of the effects of various drivers on the partitioning of carbon
stocks in vegetation biomass remain. Meanwhile, plants adjust carbon allocation scheme to adapt to
environmental change. With increased warming, an increase in the magnitude of water stress may
dramatically change or even reverse the impact of these drivers on inner components of carbon storage
(Ma et al., 2021). Evaluating the response pattern of carbon stocks to various drivers under conditions of
limited water is elemental for clearly documenting the response mechanism of vegetation carbon storage
potential.

Here, we use a spatially explicit individual-based dynamic global vegetation model (SEIB-DGVM),
along with the components partitioning method to (1) systematically determine the long-term variability
of carbon storage potential and understand its response mechanisms, and (2) estimate trends in
partitioning of potential biomass carbon stocks of vegetation biomass. Throughout this study, the
potential biomass carbon stock, biomass carbon stored in vegetation without anthropogenic disturbance,
is recognized as an indicator of the potential of carbon storage by natural vegetation. Using a set of
factorial simulations to isolate responses to environmental change, we analyse the contributions of
multiple driving factors to the trends of two fractions of carbon stock at large scales individually. We
then conceptualize the role of water availability through an aridity index (AI), in which hydrological
zones are subdivided by their degree of aridity. By comparing the differences in the magnitude of
response between the fractions of light- and water-gathering carbon stocks for varying degrees of water
availability, we assess the effect of water limitations on the response pattern of potential carbon stocks
to changes in climate and $CO_2$.

**2 Model description, experimental design, observational data, and evaluation metrics**

In this section, we provided a list of data source (Sect. 2.1), an overview of the modelling concept (Sect. 2.2), the representation of biomass carbon stock partitioning in the SEIB-DGVM (Sect. 2.3), an overview of the experimental scheme used in the model simulations (Sect. 2.4), and an overview about data source and pre-processing of observation dataset for model evaluation (Sect. 2.5).

**2.1 Forcing Data**

Long-term daily meteorological time-series data are required to run model simulations, including precipitation, daily range of air temperature, mean daily air temperature, downward shortwave radiation at midday, downward longwave radiation at midday, wind velocity, and relative humidity. These data were obtained from the Climatic Research Unit (CRU) time series 4.00 gridded dataset (degree 0.5°) for the period 1901–2015 (Harris et al., 2020). Because the CRU dataset is a monthly based dataset, the monthly meteorological data were converted into daily climatic variables by supplementing daily climatic variability within each month using the National Centre for Environmental Prediction (NCEP) daily climate dataset. The NCEP data, displayed using the T62 Gaussian grid with 192 × 94 points, was interpolated into a 0.5° grid (which corresponds to the CRU dataset) using a linearly interpolation method. By combining the CRU data, with the interpolated NCEP dataset, we were able to directly obtain the most of driving meteorological data (details in Sato et al. (2020)). Neither the CRU nor NCEP datasets included downward shortwave and longwave radiation at midday. Thus, daily cloudiness values in the NCEP were used to calculate radiation values using empirical functions (Sato et al., 2007). These data were all aggregated to a daily timescale with 0.5° resolution to run SEIB-DGVM.

Atmospheric $CO_2$ concentrations were collected from Sato et al. (2020), which contains reconstructed $CO_2$ concentrations between 1901 and 2015. The statistical reconstruction of global atmospheric $CO_2$ was used in this analysis. These reconstructions were based on present annual $CO_2$ concentrations recorded from the Mauna Loa monitoring station. These data assume atmospheric $CO_2$ concentration was 284 ppm in 1750, and statistically interpolates atmospheric $CO_2$ concentrations to fill the gap from 1750 to 2015.

The physical parameters of the soil used in the model include soil moisture at the saturation point, field
capacity, matrix potential, wilting point and albedo. These data were obtained from the Global Soil
Wetness Project 2.
**2.2 Overview of modelling concept in SEIB-DGVM**
Model SEIB-DGVM version 3.02 (Sato et al., 2020) was employed in this study. This is a process-based
dynamic global vegetation model driven by meteorological and soil data. It is an explicit and
computationally efficient carbon cycle model designed to simulate transient effects of environmental
change on terrestrial ecosystems and land-atmosphere interactions. It describes three groups of processes:
land-based physical processes (e.g., hydrology, radiation, aridity), plant physiological processes (e.g.,
photosynthesis, respiration, litter), and plant dynamic processes (e.g., establishment, growth, mortality).
Twelve plant functional types (PFTs) were classified. During the simulation, a sample plot was
established at each grid cell, and then the growth, competition, and mortality of each the individual PFTs
within each plot were modelled by considering the specify conditions for that individual as it relates to
other individuals that surround it (Sato et al., 2007).

SEIB-DGVM treats the relationships between soil, atmosphere, and terrestrial biomes in a consistent
manner, including the fluxes of energy, water, and carbon. Based on specified climatic conditions and
soil properties, SEIB-DGVM simulates the carbon cycle, energy balance, and hydrological processes.
SEIB-DGVM utilizes three computational time steps: (1) During the growth phase, the metabolic
procedures including photosynthesis, respiration, and carbon allocation are executed for each individual
tree every simulation day. (2) The monthly process of tree growth including reproduction, trunk growth,
and expansion of a cross-sectional area of the crown are executed. (3) On the last day of each year, the
height of the lowest branch increases as a result of purging crown disks, or self-pruning of branches, at
the bottom of the crown layer. The simulated unit of the model is a 30 m × 30 m spatially explicit 'virtual
forest'. A grass layer was placed under the woody layer, and provides for a comprehensive, spatially
explicit quantification of terrestrial carbon sinks and sources. The soil depth was set at 2 m and was
divided into 20 layers, each with a thickness of 0.1 m. The photosynthetic rate of a single-leaf was
simulated following a Michaelis-type function (Ryan, 1991). Respiration was divided into two types:
growth respiration and maintenance respiration. Growth respiration is defined as a construction cost for
plant biosynthesis, which is quantified by the chemical composition of each organ (Poorter, 1994).
Maintenance respiration of live plants occurs every day regardless of the phenological phase, and is
controlled by the temperature and nitrate content of each organ (Ryan, 1991). For a wide variety of plant
organs, the maintenance respiration rate is linearly related to the nitrogen content of living tissue. The
relative proportions of nitrogen in each organ for any PFT are linearly correlated. N-deposition isn't
included in SEIB-DGVM. Atmospheric $CO_2$ was envisioned to be absorbed by photosynthesis of woody
PFTs and grass PFTs. This assimilated carbon flux was then allocated into all the plant organs (leaf,
trunk, root, and stock), where maintenance respiration and growth respiration occur. The hydrology
module treats precipitation, canopy interception, transpiration, evaporation, meltwater, and penetration.
**2.3 Carbon stock of vegetation biomass partitioning**
**2.3.1 Parameterization of daily allocation**
Flexible allocation schemes about resources and biomass are set up in the framework of the SEIB-DGVM
biogeochemical model. Based on the updated observation data, the allocation schemes of Boreal Needle-
leaved summer-green trees and Tropical Broad-leaved evergreen trees are improved at SEIB-DGVM
V3.02. Allocation schemes of other PFTs are the same as the original version. Atmospheric $CO_2$ is
assimilated by the photosynthesis of both woody and grass foliage, and then is added into the non-
structural carbon of the plant. This non-structural carbon of photosynthetic production is allocated to all
the plant organs (foliage, trunk, root, and stock), supplying what is needed for the maintenance and
growth of each organ. When the non-structural carbon is greater than 0 during the growth phase, the
following dynamic carbon allocation is executed for each individual plant at the daily time scale, such
that:
(1) When the fine root biomass ($mass_{root}$) of wood or grass does not satisfy minimum requirements for
fulfilling functional balance ($mass_{leaf}/FR_{ratio}$), the mass of non-structural carbon is allocated to the root
biomass to supplement the deficit. Here, $mass_{leaf}$ is the leaf biomass, and $FR_{ratio}$ is the ratio of $mass_{leaf}$ to
$mass_{root}$ satisfying the functional balance.
(2) The stock biomass is supplemented until it is equal to leaf biomass. This scheme is active after the
first thirty days of the growing phase.
(3) Woody leaf biomass is constrained by three limitations of the maximum leaf biomass, which are
calculated as follows:
$$max_1 = \left(crown_{area} + \pi crown_{diameter} crown_{depth}\right)\frac{LA_{max}}{SLA} \tag{1}$$
$$max_2 = ALM_1 \frac{\pi\left(dbh_{heartwood}/2 + dbh_{sapwood}/2\right)^2 - \pi(dbg_{heartwood}/2)^2}{SLA} \tag{2}$$
$$max_3 = \frac{mass_{available}}{RG_f} \tag{3}$$
$$mass_{leaf} = \min(max_1, max_2, max_3) \tag{4}$$
where $max_1$, $max_2$, and $max_3$ are, respectively, maximum leaf biomass for a given crown surface
area, cross-sectional area of sapwood, and non-structural carbon; $SLA$ is a constant of PFTs leaf area
($m^2\ g^{-1}$); $LA_{max}$ is the plant functional type specific maximum leaf area per unit crown surface area
excluding the bottom layer ($m^2\ m^{-2}$); $ALM_1$ represents the area of transport tissue per unit biomass, and
is a constant (dimensionless). If the mass$_{leaf}$ is less than the minimum $(max_1, max_2, max_3)$, the mass of
non-structural carbon is allocated into leaf biomass to supplement the deficit.
When the leaf area index of grass equals the optimal leaf area index, it stops to allocate non-structural
carbon to grass leaf, which is calculated as:
$$lai_{opt} = \frac{\ln par_{grass} - \ln\left\{\frac{p_{sat}}{lue}\left[\left(1 - \frac{cost/SLA}{0.09093 \times dlen \times p_{sat}}\right)^{-2} - 1\right]\right\}}{eK} \tag{5}$$
where $lai_{opt}$ is the optimal leaf area index ($m^2\ m^{-2}$); $par_{grass}$ is the grass photosynthetically active
radiation ($\mu$mol photon $m^{-2}\ s^{-1}$); $p_{sat}$ is the light-saturated photosynthetic rate ($\mu CO_2\ m^{-2}\ s^{-1}$); $lue$ is
the light-use efficiency of photosynthesis (mol $CO_2$ mol photon$^{-1}$); $cost$ is the cost of maintaining
leaves per unit leaf mass per day (g DM g DM$^{-1}$ day$^{-1}$); $dlen$ is day length (hour); and $eK$ is light
attenuation coefficient at midday.
(4) When non-structural carbon is less than 10 g dry mass (DM) PFT$^{-1}$ or annual NPP is less than 10 g
DM PFT$^{-1}$ in the previous year, the following daily simulation processes (5-6) will be skipped.
(5) When total woody biomass is more than 10 kg DM, which defines the minimum tree size for
reproduction, 10% of non-structural carbon is used for every daily process of reproduction, including
having flowers, pollen, nectar, fruits, and seeds. These organs are not explicitly modelled in SEIB-
DGVM.
(6) During the simulation of trunk growth, the remaining non-structural carbon is allocated to sapwood
biomass. There is no direct allocation to heartwood, which is transformed slowly from sapwood biomass.
For grass PFTs biomass, the densities of all organs comprising the biomass never decline below 0.1 g
DM m$^{-2}$ even if the environment is deteriorated for grass survival. A more detailed description of SEIB-
DGVM is given by Sato et al. (2007).

To control plant phenology and the rate of photosynthesis as a function of the limitation in terrestrial
water, the physiological status of the limitation of terrestrial water is calculated as:
$$p_{sat} = PMAX\,ce_{tmp}\,ce_{co_2}\,ce_{water} \tag{6}$$
$$ce_{water} = \sqrt{stat_{water}} \tag{7}$$
$$stat_{water} = \frac{max(pool_{w(1)}/Depth_{(1)},\ pool_{w(2)}/Depth_{(2)}) - W_{wilt}}{W_{fi} - W_{wilt}} \tag{8}$$
where $p_{sat}$ is the single-leaf photosynthetic rate of tree PFTs and grass PFTs ($\mu$mol $CO_2$ m$^{-2}$ s$^{-1}$);
$PMAX$ is the potential maximum of photosynthetic rate ($\mu$mol mol$^{-1}$ $CO_2$ m$^{-2}$ s$^{-1}$); $ce_{tmp}$ and $ce_{co_2}$ are
the temperature and $CO_2$ concentration effect coefficient (dimensionless), separately; $ce_{water}$ is the
water effect coefficient (dimensionless); $stat_{water}$ is the physiological status of the terrestrial water
limitation, which ranges between 0.0–1.0, dimensionless; $pool_{w(n)}$ is the water content in soil layer n,
mm; $Depth_{(n)}$ is the depth of the soil layer n, mm; $W_{wilt}$ is soil moisture at the wilting point, m m$^{-1}$;
and $W_{fi}$ is soil moisture at field capacity, m m$^{-1}$. When the temperature of all soil layers is less than 0°
C, $stat_{water}$ is equal to 0.
**2.3.2 Carbon stock partitioning method**
SEIB-DGVM allocates and stores the biomass carbon in four pools of woody PFT (foliage, trunk, root,
and stock) and three pools of grass PFT (foliage, root, and stock). To investigate the fractional variability
of carbon sequestration potential between the pools, we partitioned potential vegetation carbon stocks
based on the physiological function of the plant (Figure A1). The root-shoot ratio (R/S) has been used to
distinguish and investigate the ratio of below-ground biomass (root biomass) and above-ground biomass
(shoot biomass) (Zhang et al., 2016). In this study, we adjusted the method of calculating the R/S ratio
by distinguishing between the light-gathering vegetation biomass carbon stock (LVBC) and the water-
gathering vegetation biomass carbon stock (WVBC). LVBC represents the biomass carbon invested by
plant is used to gather sunlight, including biomass carbon from woody foliage, woody trunk, and grass
foliage. WVBC represents biomass carbon used to gather water, including biomass carbon from woody
fine roots and grass fine roots, excluding the stock pool. Stock biomass is used for foliation after dormant
phase and after fires, which is reserve resource in each individual tree. Fine root biomass is just a tiny
fraction to the total biomass, but is has a very high turnover rate and determines the capacity of vegetation
to absorb soil water. Thus,
$$\frac{LVBC}{WVBC} = \frac{Tmass_{leaf} + Tmass_{trunk} + Gmass_{leaf}}{Tmass_{root} + Gmass_{root}} \times 100\% \qquad (9)$$
where $LVBC$ is light-gathering vegetation biomass carbon stock (kg C m$^{-2}$); $WVBC$ is water-gathering
vegetation biomass carbon stock (kg C m$^{-2}$); $Tmass_{leaf}$ is the leaf biomass carbon stock of woody
vegetation (kg C m$^{-2}$); and $Tmass_{trunk}$ is the trunk biomass carbon stock of trees (kg C m$^{-2}$), including
both branch and structural roots. This biomass is simplistically attributed to light-gathering vegetation
organs and is used primarily to support the plant. $Gmass_{leaf}$ is the leaf biomass carbon stock of grass
(kg C m$^{-2}$); whereas $Tmass_{root}$ and $Gmass_{root}$ are functional root (fine roots) biomass carbon stocks
of trees and grass, separately (kg C m$^{-2}$), which absorb water and nutrition from soil.
**2.4 Experimental design**
**2.4.1 Setup of model runs**
SEIB-DGVM simulations begin with seeds of selected PFTs planted in bare ground. The establishment
of PFTs seeds are determined by the climatic conditions in each grid cell. We inputted the transient
climate data from 1901 to 1915 to spin up the model in a repetitive loop. No obvious trend in climatic
factors was observed during this period (Tei et al., 2017). A spin-up period of 1050 years was necessary
to bring the terrestrial vegetation carbon cycle into a dynamic equilibrium. To reach quasi-equilibrium
in the vegetation biomass, about 1000 years of simulation was required as a spin-up procedure.
**2.4.2 Factorial simulation scheme**

**Table 1.** List of factorial simulations used in this study

| Factorial simulation | CO$_2$ concentration | Precipitation | Temperature | Radiation | Other drivers |
|---|---|---|---|---|---|
| S1 | √ | √ | √ | √ | √ |
| S2 | √ | | | | |
| S3 | √ | √ | | | |
| S4 | √ | | √ | | |
| S5 | √ | | | √ | |
| S6 | √ | | | | √ |

Note: In factorial simulation S1, historical atmospheric CO$_2$ concentration and historical climate fields from the

CRU data set were used. In simulation S2, only historical atmospheric $CO_2$ concentration was used, and climate variables of the transient period (1901–1915) were repeatedly input. In simulation S3 (or S4, S5), only historical atmospheric $CO_2$ concentrations and precipitation (or temperature, radiation) were input, and climate variables of the transient period (1901–1915) were repeatedly input. In the last simulation S6, only historical atmospheric $CO_2$ concentrations and other climate variables were input, including wind velocity and relative humidity.

In order to further quantify the relative contributions of varying atmospheric $CO_2$ concentrations,
precipitation, temperature, radiation, and other factors (wind velocity and relative humidity), we
performed six factorial simulations. In simulation S1, atmospheric $CO_2$ concentration and all of climate
variables were varied. In simulation S2, only atmospheric $CO_2$ concentration was varied, and climate
variables were held constant (Climate variables of the transient period (1901-1915) were repeatedly
inputted). In simulation S3 (or S4, S5), atmospheric $CO_2$ and precipitation (or temperature, radiation)
were varied, and other climate variables were held constant. In simulation S6, atmospheric $CO_2$, wind
velocity, and relative humidity were varied, and other climate variables were held constant. Finally, S2
was used to evaluate the effects of $CO_2$ fertilization on carbon stock variation. The differences of S2-S3,
S2-S4, S2-S5, and S2-S6 were used to evaluate the response of carbon stock growth to precipitation,
temperature, radiation, and other drivers, respectively.
**2.4.3 Non-parametric test methods**
Each driving factor (atmosphere $CO_2$, precipitation, temperature, and radiation) has a different influence
on the carbon stock, so it is difficult to make a simple pre-assumption about the population distribution
pattern for factorial simulations. We used the non-parametric Mann-Kendall and Sen's slope estimator
statistical tests (Gocic and Trajkovic, 2013) to assess the ability of SEIB-DGVM to simulate the response
patterns of carbon storage potential to a change in climate and $CO_2$ concentrations. We regressed the
simulated hundred-year mean global average carbon stock time series to reveal the accumulative
influences of the single variables based on the factorial simulations where only one or two drivers were
varied. As shown in Figures A2 and A3, detection trends of LVBC and WVBC for all driving factors
performed statistically well (in agreement at the 95% confidence intervals), indicating this analytical
method was suitable for trend attribution at the global scale.
**2.4.4 Distinguishing hydrological regions**

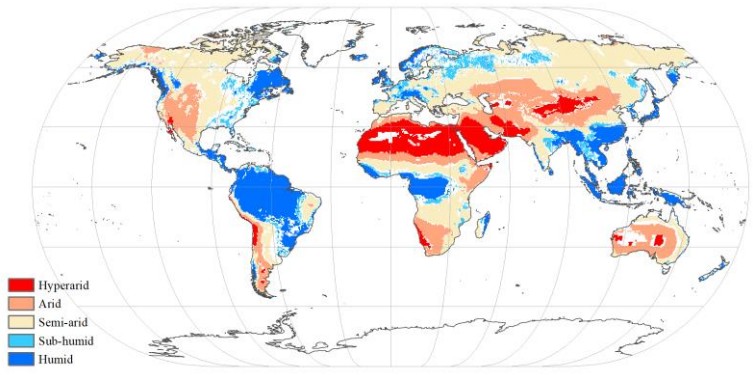

**Figure 1. Global spatial patterns of water availability.** Spatial variations in water availability were

categorized based on the multiyear average aridity index (AI), defined as the ratio of the multiyear

mean precipitation to the potential evapotranspiration. Categories include: hyper-arid (AI ≤ 0.05), arid

(0.05 < AI ≤ 0.2), semi-arid (0.2 < AI ≤ 0.5), sub-humid (0.5 < AI ≤ 0.65), and humid (AI > 0.65).

The white grid cell were not assigned hydrological category.

Locally available water strongly regulates and limits the response of carbon stocks to changes in climate
and $CO_2$. We used aridity index (AI) to distinguish between the global hydrological regions for
comparing the long-term trend in carbon stocks over different hydrological environments, and for
quantifying the influences of each hydrological environment on the variations in the trends. The AI was
defined as:
$$AI = \frac{\bar{P}}{\overline{ET_p}} \qquad\qquad (10)$$
where $\bar{P}$ is the multiyear mean precipitation (mm year$^{-1}$); and $\overline{ET_p}$ is the multiyear mean potential
evapotranspiration (mm year$^{-1}$), which was calculated by the Penman-Monteith model (Monteith and
Unsworth, 1990). As in a previous study (Chen et al., 2019), five hydrological regions were categorized
based on AI value. Under the influences of climate change, the hydrological condition was changed in
some grid cells (Figure A4). For example, the grid cell classified as sub-humid zone in the period of
1916-1945 was redefined as semi-arid zone in the period of 1986-2015. In this study, gird cells with
consistent hydrological condition between the period of 1916-1945 and the period of 1986-2015 were
selected and classified (Figure 1).
**2.5 Observation dataset for model evaluation**
A global time series of potential vegetation carbon was modelled by the SEIB-DGVM between 1916-
2015. In terrestrial vegetation biomes, there is a high correlation between biomass carbon stock density
and NPP per unit (Erb et al., 2016; Kindermann et al., 2008) (Figure A1). Thus, we collected NPP
observation dataset and used NPP as a proxy of the carbon stock to assess model accuracy. Ecosystem
Model-Data Intercomparison (EMDI) builds upon the accomplishments of the original worldwide
synthesis of NPP measurements and associated model driver data prepared by Global Primary Production
Data Initiative. We obtained the monitoring station data from the EMDI working group, and then
compared their data with modelled multiyear average NPP in the period of 1916-1999 (Figure 2).

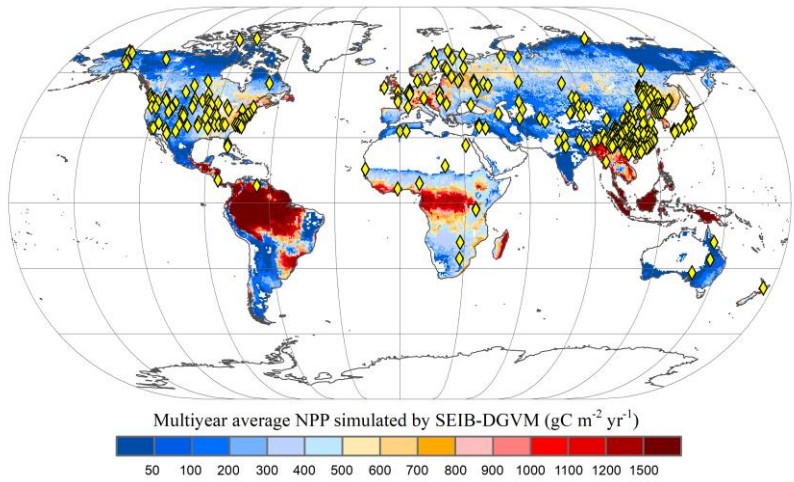

**Figure 2. Multiyear average NPP simulated by SEIB-DGVM and EMDI global site distribution**.

Yellow rhombuses indicate the monitoring stations of the EMDI.

However, *in-situ* observations are sparse for global spatial-temporal validation. Therefore, we used the
MOD17A3 products to further verify the simulated potential NPP in the twenty first century. These data
were collected by the Moderate Resolution Imaging Spectroradiometer and are some of the most widely
used data to assess the accuracy of global model simulations (Gulbeyaz et al., 2018). The natural
vegetation zones refer to the hypothetical condition that would prevail in an assumed absence of
anthropogenic activity, but under historical climate fields (Erb et al., 2018; Haberl et al., 2014). The
potential NPP is defined as that assimilated carbon stored in natural vegetation without the disturbance
of anthropogenic activities (Erb et al., 2018).

In order to distinguish the distribution of vegetation grid cells without anthropogenic disturbance, we
obtained global land cover types in the period 2001-2015 from MCD12C1 (Table A1). We included grid
cells whose largest vegetation component was evergreen needleleaf forest, evergreen broadleaf forest,
deciduous needleleaf forest, deciduous broadleaf forest, mixed forest, closed shrublands, open
shrublands, woody savannas, savannas or grasslands. Other grid cells were excluded from our analysis.

Part of grid cells covered by grassland were grazed by livestock, leading to the decrease of NPP of grass
PFTs. There is a weak anthropogenic disturbance in rangeland, while managed pasture is intensely grazed
by livestock. To remove pasture area with strong anthropogenic disturbance, we obtained land-use
forcing data from Land-Use Harmonization (LUH2) to map the distribution of managed pasture data
from 2001 to 2015 (Hurtt et al., 2020). As shown in Figure A5, grassland in eastern Asia, western Europe,
south central Africa, and western South America were severely affected by grazing. For exhibit the
disturbance of managed pasture, we calculated the mean fraction of managed pasture within the
corresponding 0.5° grid unit. When the fraction of managed pasture over 10%, the grid cell was
considered to be affected by managed pasture. To reduce the interference effects of livestock grazing,
we first removed the grid cells affected by managed pasture. Then, we map the distribution of natural
vegetation grid cells without anthropogenic disturbance (Figure A6). This exclusion method is only used
for potential NPP comparison.
**3 Results and discussion**
**3.1 Evaluation of SEIB-DGVM**
Figure 3 illustrates the comparison between model simulated and observed multi-year mean NPP during
1916-1999. The determined coefficient ($R^2$) between EMDI observed and estimated multiyear average
NPP of 669 *in-situ* observations is 0.54, which is significant at the p=0.01 level. The slope of the
regressed line is 0.70 during the twentieth century.

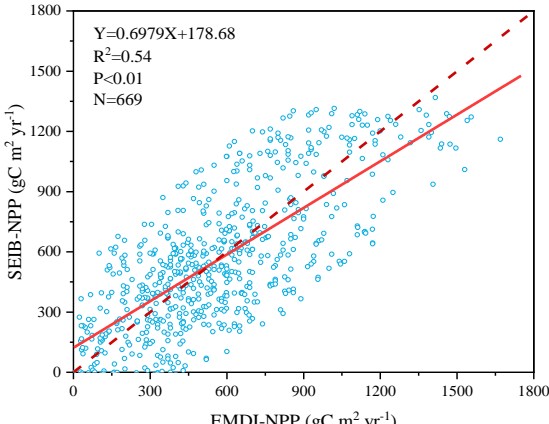

**Figure 3. Comparison of multiyear average NPP calculated by SEIB-DGVM and EMDI for the twentieth century.** The solid line is the best fit curve; and the dashed line represents a perfect correspondence in the results of the two.

Based on land cover types dataset from 2001 to 2015, we obtained NPP-MOD17A3 data in natural
vegetation zones without anthropogenic disturbance at the same period. Figure 4 shows that the modelled
NPP from the SEIB-DGVM exhibited a high degree of consistency with the NPP-MOD17A3 data in
natural vegetation zones over the period ($R^2$=0.63, p<0.05). The general spatiotemporal agreement
between the simulated NPP derived from SEIB-DGVM with *in-situ* observations and derived from
satellites reveals that it is reasonable to use the SEIB-DGVM simulations to evaluate the same
mechanisms controlling global potential biomass carbon stocks of vegetation.

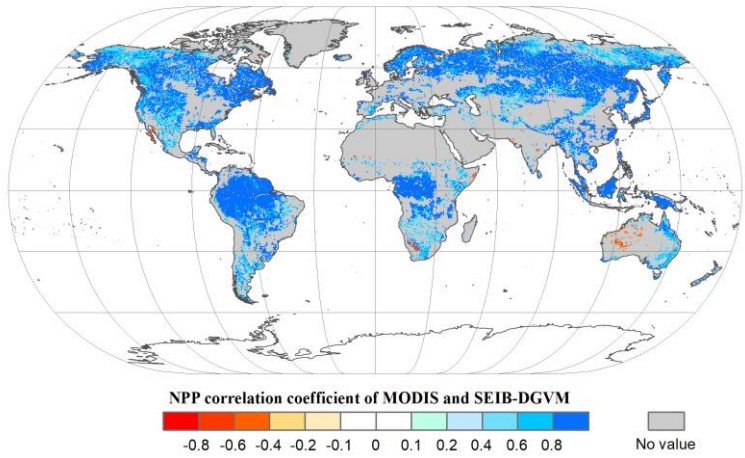

**Figure 4. Spatial patterns in the potential NPP correlation coefficients (P<0.05) between SEIB-DGVM and MODIS between 2001–2015.** These data were used to validate SEIB-DGVM.

Finally, the modelled result of potential vegetation biomass carbon stock was compared with current
existing data from the literature and state-of-the-art datasets. Figure 5 shows that the modelled results are
within the range of potential carbon stocks, which indicate that the SEIB-DGVM reliably simulated the
carbon stock dynamics.

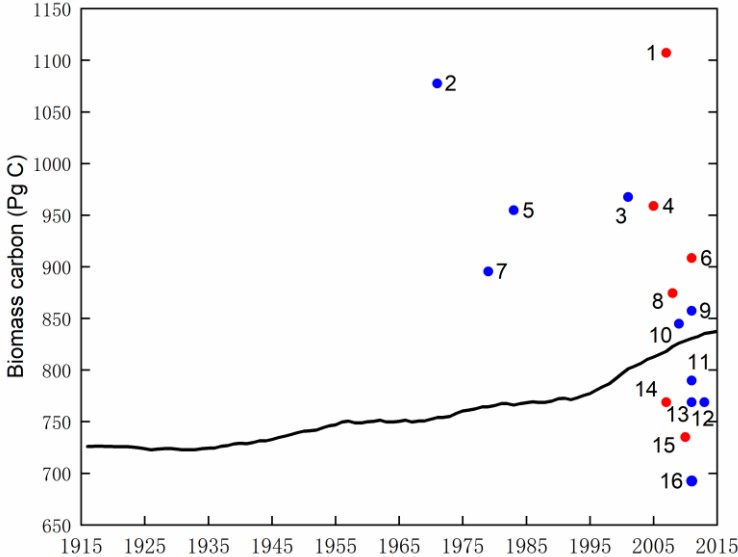

**Figure 5. Estimates of the potential vegetation biomass carbon stock from the literature (blue plot), state-of-the-art datasets (red plot) and this study (black line).** Datasets are from the following studies: (1)(Erb et al., 2018; Erb et al., 2007), (2)(Bazilevich et al., 1971), (3)(Saugier et al., 2001), (4)(Erb et al., 2018; Bartholome and Belward, 2005), (5)(Olson et al., 1983), (6)(Erb et al., 2018; Pan et al., 2011), (7)(Ajtay et al., 1979), (8)Erb et al., 2018; Ruesch and Gibbs, 2008), (9)(Kaplan et al., 2011), (10)(Shevliakova et al., 2009), (11)(Kaplan et al., 2011), (12)(Pan et al., 2013), (13)(Prentice et al., 2011), (14)(Erb et al., 2018; Erb et al., 2007), (15)(Erb et al., 2018; West et al., 2010), (16)(Hurtt et al., 2011).

**3.2 Enhanced carbon stocks and its fractions**
We distinguished the changes of LVBC and WVBC from total vegetation carbon stocks. The historical
temporal trends over the period are shown in Figure 6a. The potential vegetation carbon stock exhibits a
net increase of $119.26 \pm 2.44$ Pg C in the last century ($\pm 2.44$ represents intra-annual fluctuation in carbon
stock, which is the difference between maximum value and minimum value of carbon stock within the
year). Based on Pearson correlation analysis, this increasing trend of annual average carbon stock
exhibits a robust agreement with the dramatic increase in atmospheric $CO_2$ concentration ($R^2$=0.9677,
p<0.001), suggesting that the carbon stock is strongly affected by $CO_2$ fertilization. Meanwhile, the
positive correlation between the carbon stock and $CO_2$ generally extends across LVBC ($R^2$=0.9669) and
WVBC ($R^2$=0.9622). After the value of the global terrestrial carbon stock and trends were partitioned
among the vegetation functional classes, we see that LVBC increases 116.18 ± 2.34 Pg C (or ~15.60%),
which explains 97.42% of total carbon stock increasing trend and dominates the positive global carbon
stock trend; WVBC also increases 3.08 ± 0.14 Pg C (or ~18.03%) over the past century.

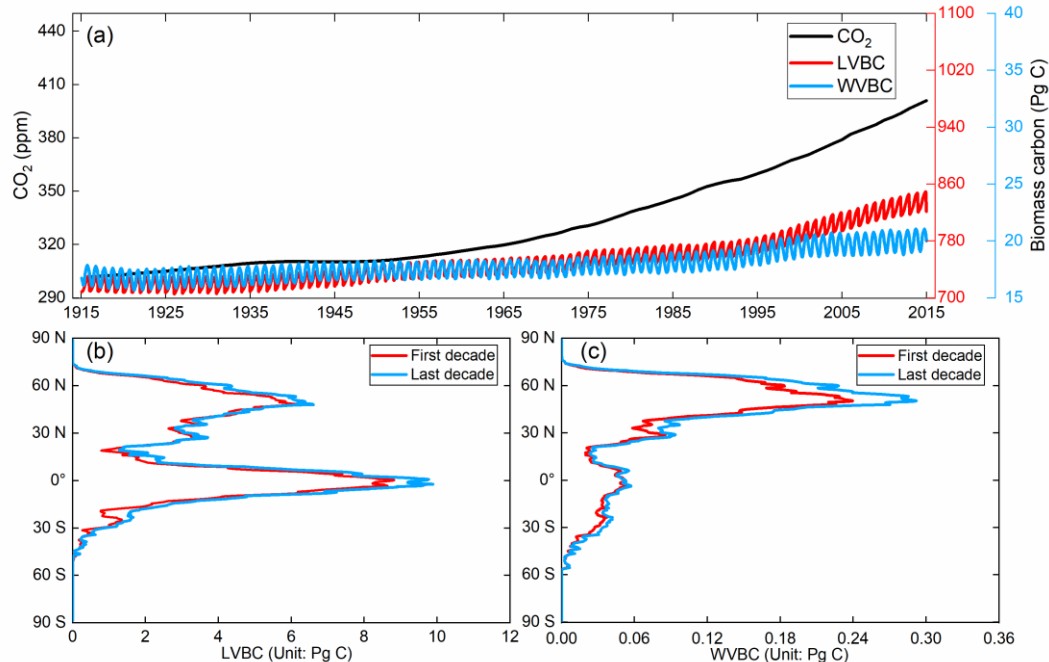

**Figure 6. Global potential biomass carbon stocks of vegetation during the past 100 years.** (a) The

evolution of global potential biomass stocks (LVBC+WVBC), along with changes in biomass stocks

that can be attributed to the variability and trend of LVBC and WVBC through the twentieth century.

The red line represents the monthly value of LVBC, the blue line represents the monthly value of

WVBC, and the black line represents the annual value of $CO_2$ concentration. (b, c) Zonal averaged

sums of the annual LVBC and WVBC for latitudinal bands during the first decade (1916–1925, red

line) and the last decade (2006–2015, blue line) shows the increased carbon stock capacity.

The global distributions of the decadal-average change in LVBC and WVBC are shown in Figures 6b
and 6c, respectively. The significant historical changes in climate and $CO_2$ enhance the carbon stock of
the terrestrial ecosystem, and their positive influences are broadly distributed across a latitudinal north–
south gradient. The latitudinal bands of increasing annual LVBC are mainly distributed in the tropical
and boreal latitudes, which is consistent with Figure 7b. The decadal and inter-annual variabilities of
LVBC are dominated by the tropical and boreal zones where large portions of the zones are highly
productive (Ahlstrom et al., 2015; Poulter et al., 2014). Tropical LVBC dominates the long-term trend
of global LVBC in the last hundred years. Compared with LVBC, the increase of tropical WVBC is light.
There is a single peak in the spatial variation of annual WVBC (Figure 6c and Figure 7c). WVBC exhibits
robust growth at most latitudes, and increases mainly in boreal latitudes.
**3.3 Spatial variability in estimated LVBC and WVBC trends**
In Figures 7(a) and 7(b), total carbon stock and LVBC exhibited a significantly increasing trend in eastern
South America, southern Africa, and northern Asia, while they declined in central North America,
northwest South America, and central Africa. WVBC showed a more widely increasing tendency in
North America, southeastern South America, and Europe, while had a decreasing trend in part zones of
Asian. We find that the total carbon stock as well as the light- and water-gathering vegetation biomass
carbon stocks over the period of 1916–2015 exhibited a remarkable spatial heterogeneity. Figure 7a
shows that an increase in vegetation carbon stocks occurred over zones and global aggregate levels during
the entire study period. About 57.39% of the terrestrial grid cells exhibited an increase with a noticeable
trend ($p<0.05$) in biomass carbon stock; 53.82% of global grid cells possessed increases that were
statistically significant at the $p=0.01$ level. To determine the contributions of each fraction (LVBC,
WVBC) to the total change in the potential vegetation carbon stock, we partitioned and present the
historical spatial and temporal patterns for each fraction separately (Figure 7b, 7c). LVBC contributes
97.33% to the incremental change of total carbon stock ($116.18 \pm 2.34$ Pg C), with about 51.32% of the
grid cells possessing a noticeable positive trend ($p=0.01$). Generally, spatial patterns of LVBC and the
total carbon stock are consistent (Figure 7a, 7b), which further supports the argument that LVBC
dominates the trend in carbon stocks in most zones. Although the proportion of the total change in carbon
stocks is small (2.58% of total carbon stock increase), about 61.00% of the land surface shows an increase
in WVBC; of these terrestrial grid cells, 55.81% was characterized by a significant $p=0.01$ increase.

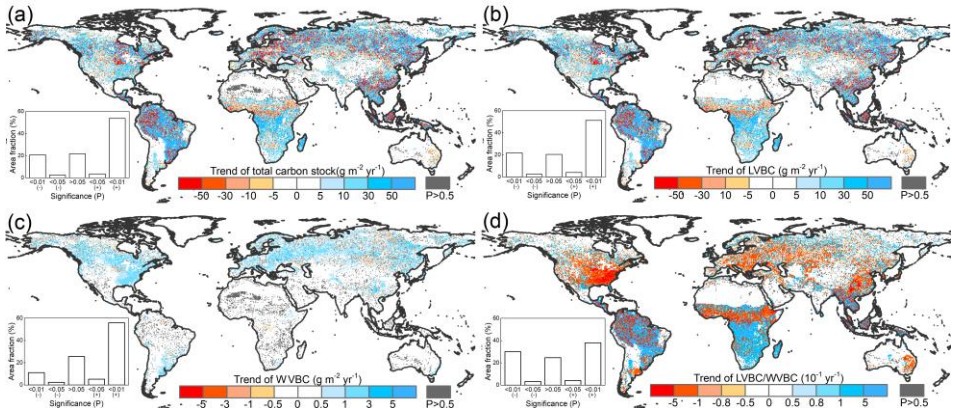

**Figure 7. Spatial patterns in the trends of potential vegetation carbon stocks and their fractions from 1916 to 2015.** Difference induced by changes in climate and $CO_2$ in terrestrial biomass carbon stock (a), LVBC (b), and WVBC (c) during the historic period 1916–2015. The blue bar indicates the significantly increasing trends and the red bar indicates the significantly decreasing trends in carbon stocks. (d) Trend in the LVBC/WVBC ratio from 1916 to 2015. The blue bar indicates significantly increasing trends in the ratio, and vice versa. The grey bar indicates the trend is statistically insignificant (P >0.05). The sub-graphs show the significant test results. A '+' symbol indicates a positive trend, and vice versa.

Under the influences of a changing climate and $CO_2$ concentrations, there is a slight increase in the ratio
of global LVBC/WVBC; the rate of increase is 0.0171 $yr^{-1}$ in the last hundred years, which is significant
at the 0.01 level (Figure 7d). About 42.08% of the terrestrial grid cells exhibits an increase with a
noticeable trend (p<0.05) in the ratio of LVBC and WVBC; 37.95% of global grid cells possessed
increases that are statistically significant at the p=0.01 level. Meanwhile, 33.32% of the land surface
shows a significant decrease in LVBC/WVBC; of these terrestrial grid cells, 30.06% is characterized by
a significant p=0.01 decrease. Grid cells with noticeable increases in the ratio of LVBC to WVBC are
mainly located in southern Africa, central South America, and northern Eurasia. Negative trends in
LVBC/WVBC ratios are found in northern America, southern Europe, and tropical Africa.
**3.4 Responses of LVBC and WVBC to environmental drivers**
The responses of LVBC and WVBC to changes in climate and $CO_2$ are both positive at the global level
(Figure 8a, 8c), although zonally, they exhibit both negative and positive responses (Figure 8b, 8d).
Based on the results of factorial simulations and Mann-Kendall+Sen tests, $CO_2$ fertilization explains the
largest proportion of the change in the carbon stock; about 82.45% change in LVBC was positive (Figure
8a), whereas 89.28% of the change in WVBC was positive (Figure 8c). In factorial simulation S2, the
long-term trend of LVBC was 15.521 g C m$^{-2}$ yr$^{-1}$ and that of WVBC was 0.435 g C m$^{-2}$ yr$^{-1}$ at the
period from 1916 to 2015 (Figure A2a and Figure A3a). The separately simulated LVBC and WVBC
increased by 80.98 Pg C and 2.66 Pg C with increasing atmospheric $CO_2$ concentrations (from 301.73
ppm in 1916 to 400.83 ppm in 2015). The other climatic drivers (precipitation, temperature, radiation,
humidity, and wind speed) remained at baseline values. While the increase or decrease in the carbon
stock may be attributed to more than one driving factor, within any specified grid cell, the one with the
highest positive or negative contribution is the dominant driver that consistently resulted in the highest
increase or decrease in the carbon stock for that grid cell. The spatial pattern illustrates that $CO_2$
dominates the variability in LVBC in 7.28% of the grid cells, including 1.21% of the grid cells that
exhibited a negative change and 6.07% that exhibited a positive change (Figure 8b). $CO_2$ dominates the
variability in WVBC in 27.60% of the grid cells, including 1.73% of the grid cells that exhibited a
negative change and 25.87% of grid cells with a positive change (Figure 8d). Under the effect of $CO_2$
fertilization, grid cells with increased trend in WVBC mainly distribute in boreal latitudes (Figure 6c).
These trends are consistent with previous studies (Tharammal et al., 2019; Zhu et al., 2016; Keenan et
al., 2017) in which positive trends occurred, especially for WVBC.

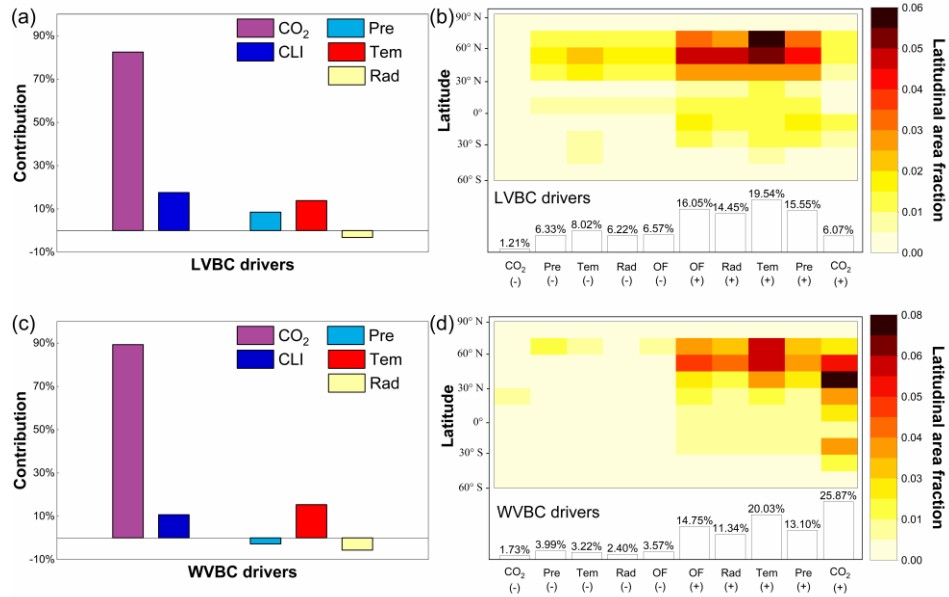

**Figure 8. The proportion of changes in vegetation biomass carbon stocks attributed to driving factors**. Ratios of the driving factors of $CO_2$ fertilization effects ($CO_2$), climate change effects (CLI), precipitation (Pre), temperature (Tem), radiation (Rad) for LVBC (a) and WVBC (c) are calculated

by the Mann-Kendall and Sen's slope estimator statistical tests. Attribution of LVBC (b) and WVBC

(d) dynamics to driving factors calculated as averages along 15° latitude bands. At the local scale,

the driving factors include $CO_2$, Pre, Tem, Rad, and other climate factors (OF). The fraction of global

grid cells (%) that is predominantly influenced by the driving factors is showed at the bottom of the

bar. The '-' symbol before fraction indicates a negative effect of the driving factor on carbon stock,

and vice versa.

Climate change induced by the greenhouse effect explains part of the increase in carbon stocks, but unlike
$CO_2$ fertilization, climate has dramatic negative effects on some vegetated zones. Figure 8a illustrates
that temperature is the largest climatic contributor to the change in LVBC (13.83%, 2.572 g m$^{-2}$ yr$^{-1}$),
followed by precipitation (8.51%, 1.572 g m$^{-2}$ yr$^{-1}$) and radiation (–3.19%, –0.649 g m$^{-2}$ yr$^{-1}$). The spatial
distribution shows that temperature predominantly influences the change in LVBC (Figure 8b),
influencing over 27.56% of the global vegetated grid cells, followed by precipitation (21.88%) and
radiation (20.67%). Figure 8c shows there are negative effects and contributions of precipitation on the
change in WVBC at the global level (–2.76%, –0.013 g m$^{-2}$ yr$^{-1}$). Temperature is the largest climatic
contributor to the change in WVBC (15.36%, 0.075 g m$^{-2}$ yr$^{-1}$), followed by radiation (-5.63%, -0.027 g
m$^{-2}$ yr$^{-1}$). Modelled WVBC trends based on the factorial simulations have similar spatiotemporal patterns
to LVBC (Figures A2 and A3), and the spatial patterns of light- and water-gathering carbon stocks show
a significantly increasing trend in the most of boreal zones. In the Southern Hemisphere, the trends of
WVBC are extensively statistically insignificant in all factorial simulations, and only a small proportion
of grid cells show a significantly increasing trend. There is a significantly increasing trend in LVBC in
south-central Africa and northern South America. The effects of temperature on WVBC are stronger than
LVBC, because temperature has a stronger effect on the metabolism process of root growth, dominating
the turnover rate and the costs of maintenance respiration in root growth process (Gill and Jackson, 2000).
It should be noted that trends in the global carbon stock can be largely attributed to the influences of $CO_2$,
precipitation, temperature, and radiation (Figure 8). Nonetheless, at the zonal scale, the contributions of
other factors should be considered, such as humidity and wind speed. The effects of these other factors
dominate trends in LVBC in over 16.05% of the grid cells that increased and 6.57% of the grid cells that
decreased. In the case of changes in WVBC, other factors were dominant drivers in over 14.75% of the
grid cells that increased and 3.57% of grid cells that decreased. Under the effect of climate, the variability
of LVBC and WVBC is positive in most grid cells, promoting the noticeable increase of carbon stocks
in boreal latitudes.
**3.5 Constraints imposed by water limitations**

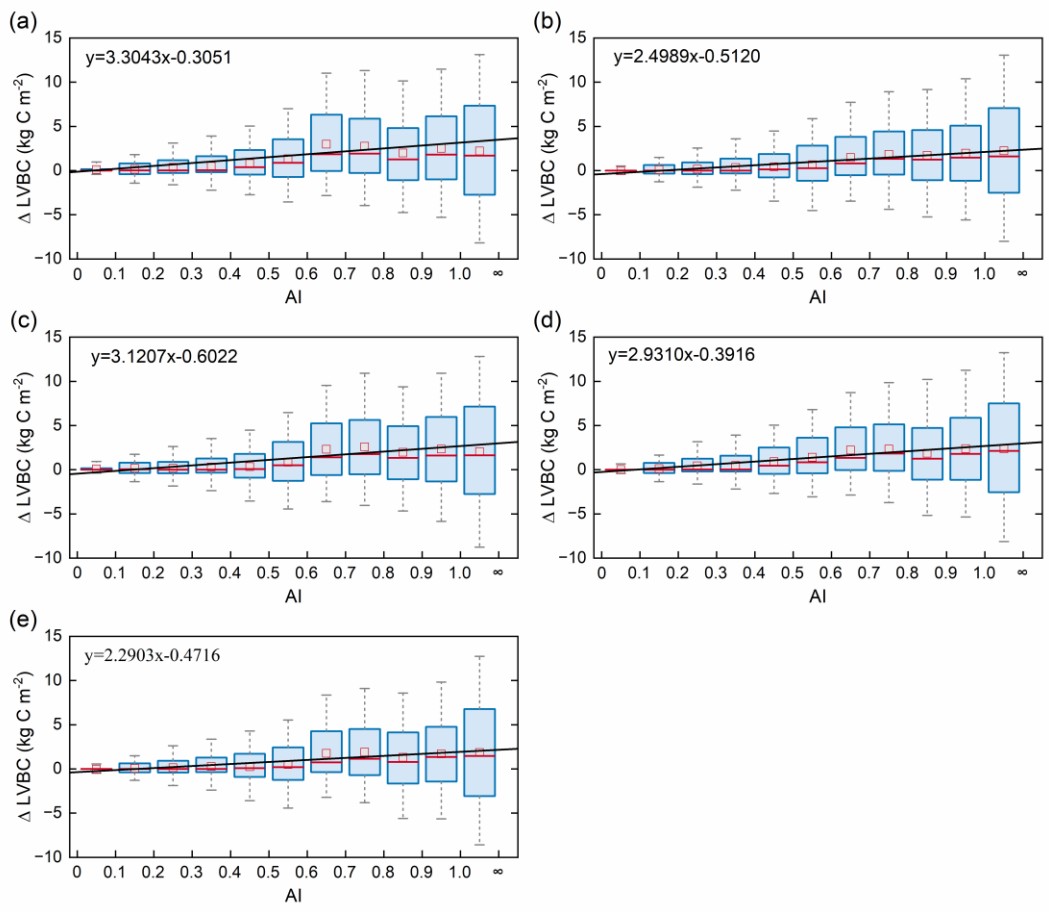

**Figure 9. Relationships of the incremental change between AI and LVBC.** Magnitude of change in LVBC in the historical scenario S1 (a), $CO_2$ in scenario S2 (b), $CO_2$ + precipitation in scenario S3 (c), $CO_2$ + temperature in scenario S4 (d), and $CO_2$ + radiation in scenario S5 (e). The range of the box is 25%-75% of values; the range of the whiskers is 10%-90% of values; the small red square is average value; the red line is the median line; and the black line is the fitted curve. Positive value of the Y axis represents the magnitude of increased LVBC from 1916 to 2015 under water-limitations conditions, and vice versa. AI of grid cells is calculated by multiyear average precipitation and multiyear average potential evapotranspiration in the period of 1916-2015. Categories of hydrological zones include: hyper-arid (AI $\leqslant$ 0.05), arid (0.05 < AI $\leqslant$ 0.2), semi-arid (0.2 < AI $\leqslant$ 0.5), sub-humid (0.5 < AI $\leqslant$ 0.65), and humid (AI > 0.65).

Terrestrial water availability emerged as a key regulator of terrestrial carbon storage, by affecting the
response mechanism of the vegetation carbon stock to changes in driving factors. As shown in Figures 9
and 10, with the accumulated change of LVBC and WVBC in the period of 1916 to 2015 across the
aridity index (i.e., an increase in available water), the magnitude and range in responses of LVBC density
and WVBC density gradually increase. Based on the results of the historical simulation (Figure 9), we
find a positive relationship between LVBC and aridity index. In extreme water stress, the increase of
LVBC tends to zero and plants stop increasing their carbon storage. There is no obvious difference in
the slopes of fitting curves between factorial simulations, which shows the robustness in the response of
LVBC to the change of water stress. The pattern of the enhanced magnitude and range of variation in the
WVBC density is unimodal with water stress gradient in all factorial simulations. With the increasing of
AI, the magnitude of change in WVBC increases at first and then decreases finally. The mitigation of
water stress promotes WVBC increase, while excess surface water limits the response of WVBC to
changes in climate and $CO_2$. These results reveal that the carbon stock increases stimulated by changes
in climate and $CO_2$ are constrained by water available. With increased warming, water limitations are
expected to increasingly limit the carbon stock increase, specially at arid regions. To further reveal the
controls of water limitation on the responses of inner carbon storages to each driver, we analyse the long-
term variability of potential vegetation carbon stocks by means of factorial simulations for each
hydrological region (Figure 1). Figure A7b shows that the fluctuation range (the difference between
maximum value and minimum value in each factorial simulation) of LVBC density across all factorial
simulation is 1.202 kg C $m^{-2}$ in the hyper-arid regions for the 1916-2015 period. As shown in Figure A7f,
the fluctuation range of LVBC density in humid regions is 6.068 kg C $m^{-2}$ during the same period. In
Figure A8b, the maximum change magnitude of WVBC density across all factorial simulation is 0.011
kg C $m^{-2}$ in the hyper-arid regions during the time of 1916-2015. In Figure A8f, the maximum change
magnitude of WVBC density is 0.046 kg C $m^{-2}$ in humid regions during the same period. Compared with
plants in arid regions, plants in humid regions show more dramatic responses to the stimulation from
drivers' change. With a lessening of water stress (from hyper-arid to humid region), the response
magnitudes of the carbon stock to the changes of climate and $CO_2$ gradually become more noticeable.
The robust pattern in the zonal average density of the carbon stock shows that terrestrial water limitations
strongly regulate the enhanced magnitude of the carbon stock.

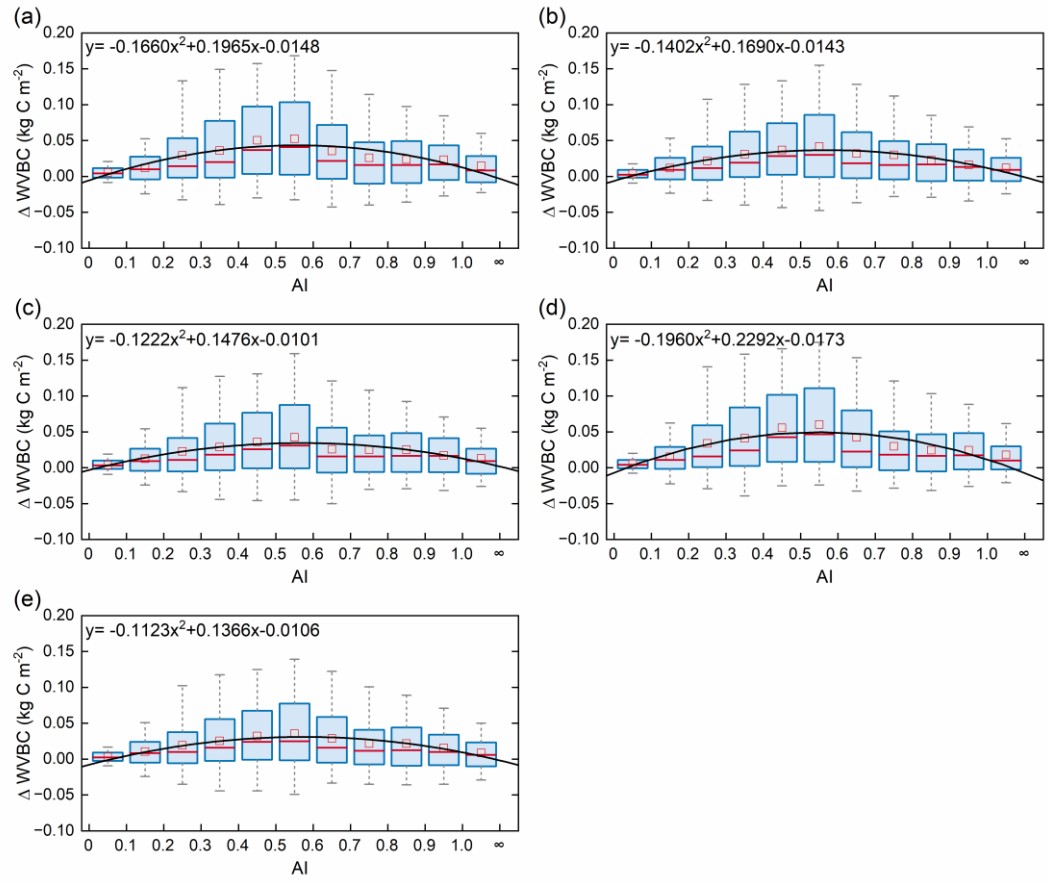

**Figure 10. Relationships of the incremental change in AI and WVBC.** Magnitude of change in WVBC in the historical scenario S1 (a), $CO_2$ in scenario S2 (b), $CO_2$ + precipitation in scenario S3 (c), $CO_2$ + temperature in scenario S4 (d), and $CO_2$ + radiation in scenario S5 (e). The range of the box is 25%-75% of values; the range of the whiskers is 10%-90% of values; the small red square is average value; the red line is the median line, and the black line is the fitted curve. Positive value of the Y axis represents the magnitude of increased WVBC from 1916 to 2015 under water-limitations conditions, and vice versa. AI of grid cells is calculated by multiyear average precipitation and multiyear average potential evapotranspiration in the period of 1916-2015. Categories of hydrological zones include: hyper-arid (AI ⩽ 0.05), arid (0.05 < AI ⩽ 0.2), semi-arid (0.2 < AI ⩽ 0.5), sub-humid (0.5 < AI ⩽ 0.65), and humid (AI > 0.65).

Water limitations not only directly reduced the magnitude of the response in the two fractions' carbon
stock (LVBC and WVBC) to changes in climate and $CO_2$, but also indirectly confined the response
direction of each fractions' carbon stock by transforming vegetation structure and function. Figure 11
illustrates temporal variations in the carbon stock ratio within and between hydrological regions. From
hyper-arid regions to humid regions, the fluctuation range of LVBC/WVBC ratio significantly changes.
The fluctuation magnitudes of LVBC/WVBC in humid and hyper-arid regions are greater than that in
other hydrological regions. Compared with plants in hyper-arid regions, plants in humid regions exhibit
more significant responses to changes in climate and $CO_2$. Meanwhile, the long-term effects of driver
changes have a remarkable influence on this carbon allocation pattern at global level (Figure 7d). Under
the synergistic effect of drivers and water stress, the trends of light- and water-gathering vegetation
carbon stock are upward in the past hundred years (Figure 6). However, there is a difference in the
increasing rate between LVBC and WVBC, resulting in a dramatic and complicated fluctuation in global
LVBC/WVBC ratio (Figure 11a). Whereas LVBC decreases and WVBC increases in hyper-arid and arid
regions (Fig. A7 and A8), causing a downward trend in LVBC:WVBC ratio, semiarid regions see an
increase in LVBC. So, the ratio of LVBC and WVBC shows a downward trend in these regions. LVBC
in semi-arid regions shows upward tendency in the past years (Figure A7d) because of the aridity
mitigation. There is an upward trend in WVBC in semi-arid regions (Figure A8d). Plants in semi-arid
regions still utilize a tolerance strategy and allocates more non-structural carbon to water-gathering
vegetation organ to resist water stress, resulting in the decline of LVBC/WVBC ratio. In humid regions,
light- and water-gathering biomass carbon stocks both increased (Figures A7 and A8). The proportion of
LVBC increases more than that of WVBC for capturing more resources like $CO_2$ and radiation energy,
leading to an increase in the LVBC/WVBC ratio. The value of LVBC/WVBC in S3 is higher than that
in S4 and S5, which represents that precipitation makes more contributions to the change of
LVBC/WVBC ratio among meteorological factors.

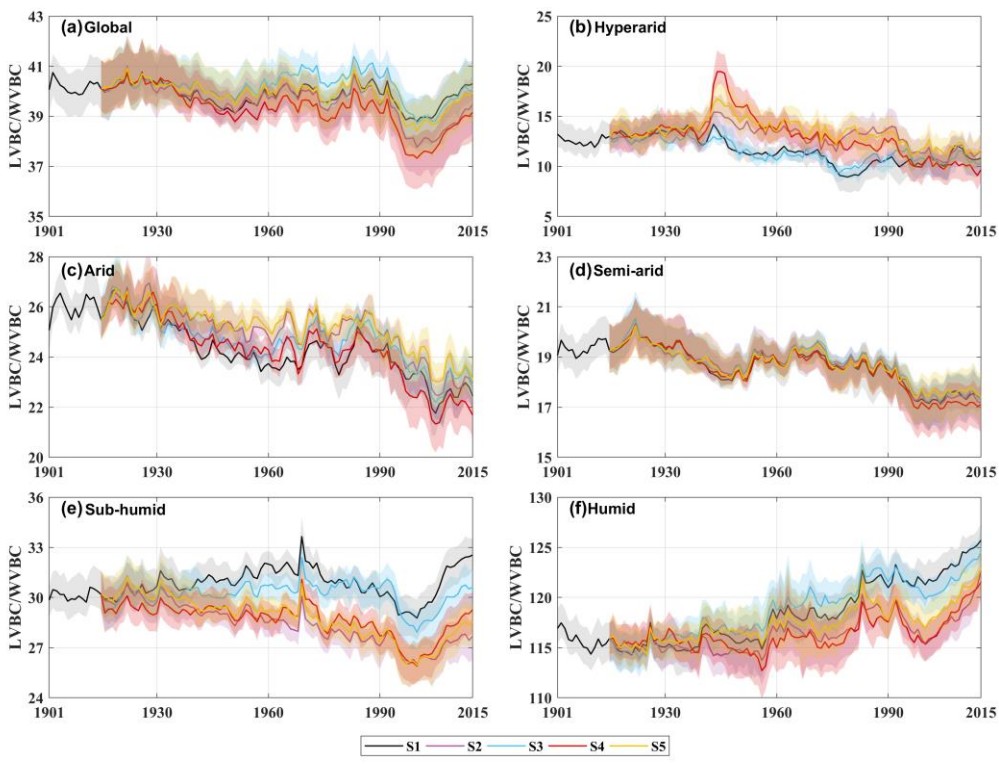

**Figure 11. Temporal fluctuations in carbon stock dynamics in vegetation biomass in different factorial simulations.** Black indicates historical factorial simulation from 1901-2015, green indicates the $CO_2$-driven factorial simulation, blue indicates the precipitation-driven factorial simulation, red indicates the temperature-driving factorial simulation and yellow indicates radiation driven factorial simulation. Uncertainty bounds are provided as shaded areas reflect the intra-annual fluctuation ($\pm$ 1 s.d.) (**a**) Modelled trend of LVBC/WVBC ratio in Global area. (**b-f**) Modelled trend of the LVBC/WVBC ratio in different hydrological regions (Figure 1).

**4 Discussions and conclusion**
To understand the response of carbon storage potential and its inner biomass carbon stocks to
environmental change, we conducted a series of factorial simulations using SEIB-DGVM V3.02. More
importantly, we investigated the extent of the responses of carbon stocks to water limitations.

Over the past 100 years, there has been an ongoing increase in the carbon storage capacity of the
terrestrial ecosystem from 735 Pg C in 1916 to 855 Pg C in 2015 (Figure 6), which has slowed the rate
at which atmospheric $CO_2$ has increased and may have mitigated global warming. These findings are
consistent with the conclusions of research conducted at the local scale. For example, based on carbon
flux data, Erb et al. (2008) suggested that the vegetation carbon stock in Austria increased from 1043 Mt
C to 1249 Mt C (aboveground carbon stocks growth was 1.059 Mt C $yr^{-1}$ and belowground carbon stocks
growth was 0.2 Mt C $yr^{-1}$) since industrialization. Le Noë et al. (2020) showed that increases in the
carbon stocks and carbon density were the predominant drivers in the forest terrestrial carbon
sequestration capacity in France from 1850 to 2015. Tong et al. (2020) also found a substantial increase
of aboveground carbon stocks in southern China (0.11 Pg C $yr^{-1}$) during the period 2002–2017. However,
these studies focused on zonal trends in total vegetation carbon stocks and did not investigate the extent
of the response in vegetation carbon stocks partitioned between light- and water-gathering biomass. Our
results show that the increase in carbon stock in light-gathering vegetation organs was much larger than
that in water-gathering vegetation organs, and light-gathering biomass carbon stock dominates the
historical trend of the terrestrial carbon stock. During the past decades, the global land surface has been
greening because of the flux and storage of more carbon into plant trunks and foliage (Zhu et al., 2016).
LVBC increases $116.18 \pm 2.34$ Pg C from 1916 to 2015, accounting for 97.42% of the total carbon stock
increase ($119.26 \pm 2.44$ Pg C). The long-term trends and spatial pattern of vegetation carbon stock
predominated the variability characteristic of LVBC. The latitudinal bands of increasing annual change
in LVBC are mainly distributed in tropical latitudes, a conclusion consistent with prior knowledge that
tropical zones dominate carbon uptake and storage (Erb et al., 2018; Schimel et al., 2015). Biomass
carbon allocation between light- and water-gathering vegetation organs reflect the changes in individual
growth, community structure and ecosystem function, which are important attributes in the investigation
of carbon stocks and carbon cycling within the terrestrial biosphere (Hovenden et al., 2014; Fang et al.,
2010; Ma et al., 2021). During the past hundred years, the ratio of LVBC/WVBC showed a slight upward
trend since LVBC increased relatively more than WVBC. The rate of increase is 0.0171 $yr^{-1}$, which is
significant at the 0.01 level. To better absorb $CO_2$ and sunlight required for photosynthesis, vegetated
regions are gradually covered by vegetation with higher plant height and wider leaf area, thereby
adjusting their characteristic ecosystem functions (Erb et al., 2008).

Based on our factorial simulations (Figure 8), the influences of $CO_2$ fertilization induce the most
significant variation of the vegetation carbon stock. In addition, the responses of carbon stocks to the
changes of climatic factors are obvious, particularly at the grid cell scale. Previous studies have pointed
out that the variation of the terrestrial carbon stock caused by releasing or sequestering carbon is sensitive
to anomalous changes in water availability and light use efficiency (Madani et al., 2020; Humphrey et
al., 2018). At the grid cell scale, as shown in Figures 8b and 8d, temperature, radiation, precipitation, and
other climate factors (humidity and wind speed) dominate the long-term trend of carbon stocks over two
thirds of global grid cells. At the global scale, climate factors explain 17.55% and 10.72% of long-term
trend in LVBC and WVBC, respectively (Figures 8a and 8c). LVBC and WVBC variations driven by
climate factors are ultimately offset by spatially compensatory effects, which dampens the response of
the carbon stock to these factors at the global scale (Jung et al., 2017). Thus, contributions of precipitation
and radiation to the variability of LVBC and WVBC are relatively low at the global scale, and the effects
of humidity and wind speed on global carbon stock are minor. This spatially compensatory effect of
climate changes is consistent with a previous analysis (Zhu et al. 2016) which found that climate changes
explain only 8% of the increasing trend in foliage carbon storage at the global level but that they dominate
the trend over 28.4% of global land area. Results show that trends in temperature drive historical long-
term trends in the potential carbon stocks, with faster increases and considerable variation occurring by
grid cell. Thus, our results reveal that temperature dominates the long-term trends of carbon stock among
climatic drivers, while a relatively strong compensatory effect exists in the global change in the carbon
stock induced by precipitation, radiation, humidity, and wind speed.

By partitioning the trends of LVBC and WVBC into five hydrological regions (Figure 1), we found that
the long-term change in carbon stocks is tightly coupled to terrestrial water availability. These results
indicate that vegetation in humid regions is responsible for most of the trend in global LVBC, while
plants in semi-arid regions play a dominate global role in controlling the long-term trend in WVBC
(Figures 9 and 10). As water stress decreases, the magnitude and range in variation of LVBC gradually
increase (Figure 9), which suggests that limited water availability constrains the response magnitude of
the changes in LVBC to changes in $CO_2$ and climate. The response pattern of WVBC growth to the
increasing water availability is different from that of LVBC. Drought mitigation promotes the growth of
WVBC. In sub-humid and humid regions, plants face low water limitations and intensified light-
competition and have to invest as much non-structural carbon as possible into leaf and trunk. This
allocation scheme leads to the decreased investment of ΔWVBC in wet regions. The result is consistent
with previous finding that plants reduce investment to roots in dense forests where aboveground
competition for light is high (Ma et al. 2021). Moreover, we found that indirect effects of water limitation
regulate increasing rate of each carbon pool. Although vegetation carbon stocks dramatically increase
under the effects of climate and $CO_2$ changes, the increasing rate of LVBC faster than WVBC in humid
regions. Vegetation stores more biomass in aboveground plant organs (trunk and foliage) to gather light.
Dryland plants decrease the LVBC/WVBC ratios and store more biomass below ground to enhance the
capture of water resources. Based on these results, we demonstrate that water limitations controlled the
variable response of terrestrial vegetation carbon stocks.

Our findings are consistent with other reports about the impact of increasing water limitations on
terrestrial ecosystem. Based on observation from satellite remote sensing, Madani et al. (2020) found
that the constraining impact of water limitation determines whether global ecosystem productivity
responds positively or negatively to the changes in climate factors. Humphrey et al. (2021) found that
increasing water stress limits the response magnitude of carbon uptake rates through a down-regulation
of stomatal conductance and suggested that land carbon uptake is driven by temperature and vapour
pressure deficit effects that are controlled by terrestrial water availability. Ma et al. (2021) found that
plants increase investment into building roots in arid region because the extent of water limitation there
is exacerbated by global warming. Terrestrial hydrological conditions significantly affect the carbon
cycle of terrestrial ecosystems, including carbon uptake, allocation, and stock. Terrestrial ecosystems
utilize sensitive strategies to allocate and store biomass to adjust to local hydrological conditions. A
significant conclusion is that water constraints not only confine the responses of vegetation carbon stocks
to drivers, but also constrain the proportion of biomass carbon stocks in gather- and water-gathering
fractions.

Distinguishing the response of carbon stock fractions estimated by SEIB-DGVM improves the
understanding of the interactive impacts of terrestrial carbon and water dynamics. However, uncertainty
still exists because of the limitations in the processes of modelling vegetation metabolism with SEIB-
DGVM. Trunk biomass contains tree branches and structural roots (coarse roots and tap roots) (Sato et

al., 2007), so the R/S ratio of potential vegetation in factorial simulations is smaller than the R/S of actual vegetation in observation stations. Root biomass only contains the fine root biomass, leading to an apparent underestimate in belowground organ biomass of trees and grasses compare with previous conclusion (Ma et al., 2021; Yang et al., 2009). Availability of nitrogen is a key limiting factor for vegetation growth, especially when higher $CO_2$ fertilization effects exist (Tharammal et al., 2019). The limitation could be alleviated by nitrogen deposition in most temperate and boreal ecosystems. The SEIB-DGVM experiments were conducted with a focus on documenting $CO_2$ fertilization and climate change interactions; these experiments did not consider the influences of nitrogen deposition, which should cause an underestimate of the contributions of $CO_2$ fertilization on biomass production.

In summary, we evaluated SEIB-DGVM V3.02 and used this model to offer new perspectives on the response of vegetation carbon storage potential to changes in climate and $CO_2$. Our simulation results show that changes in $CO_2$, rather than climate, dominate the light- and water-gathering partitioning of the carbon storage potential. More importantly, we suggest that the impact of $CO_2$ fertilization and temperature effects on vegetation carbon-sequestration potential depends on water availability and its impacts on plant stress. With increased global warming, water limitations are expected to increasingly confine global carbon sequestration and storage. Our findings highlight the need to account for terrestrial water limitation effects when estimating the response of the terrestrial carbon storage capacity to global climate change, and the need for stronger interactions between those involved in vegetation model development and those in between the hydrological and ecological research communities.

 **Appendices**

Table A1. MCD12C1 legend and class descriptions

| Name | Value | Description |
|---|---|---|
| Evergreen Needleleaf Forests | 1 | Dominated by evergreen conifer trees (canopy >2m). Tree cover >60%. |
| Evergreen Broadleaf Forests | 2 | Dominated by evergreen broadleaf and palmate trees (canopy >2m). Tree cover >60%. |
| Deciduous Needleleaf Forests | 3 | Dominated by deciduous needleleaf (larch) trees (canopy >2m). Tree cover >60%. |
| Deciduous Broadleaf Forests | 4 | Dominated by deciduous broadleaf trees (canopy >2m). Tree cover >60%. |
| Mixed Forests | 5 | Dominated by neither deciduous nor evergreen (40-60% of each) tree type (canopy >2m). Tree cover >60%. |
| Closed Shrublands | 6 | Dominated by woody perennials (1-2m height) >60% cover. |
| Open Shrublands | 7 | Dominated by woody perennials (1-2m height) 10-60% cover. |
| Woody Savannas | 8 | Tree cover 30-60% (canopy >2m). |
| Savannas | 9 | Tree cover 10-30% (canopy >2m). |
| Grasslands | 10 | Dominated by herbaceous annuals (<2m). |
| Permanent Wetlands | 11 | Permanently inundated lands with 30-60% water cover and >10% vegetated cover. |
| Croplands | 12 | At least 60% of area is cultivated cropland. |
| Urban and Built-up Lands | 13 | At least 30% impervious surface area including building materials, asphalt, and vehicles. |
| Cropland/Natural Vegetation Mosaics | 14 | Mosaics of small-scale cultivation 40-60% with natural tree, shrub, or herbaceous vegetation. |
| Permanent Snow and Ice | 15 | At least 60% of area is covered by snow and ice for at least 10 months of the year. |
| Barren | 16 | At least 60% of area is non-vegetated barren (sand, rock, soil) areas with less than 10% vegetation. |
| Water Bodies | 17 | At least 60% of area is covered by permanent water bodies. |
| Unclassified | 255 | Has not received a map label because of missing inputs |

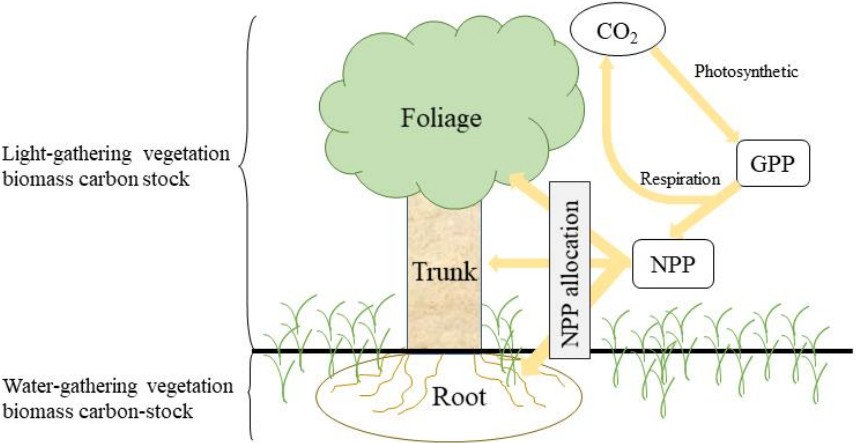

Figure A1. Schematic of ecosystem carbon cycle. Yellow arrow indicates carbon flux. Atmospheric $CO_2$ transitions into gross primary production (GPP) by photosynthesis. GPP is partitioned into respiration and net primary production (NPP). NPP is partitioned into three biomass carbon pools (foliage, trunk, and root).


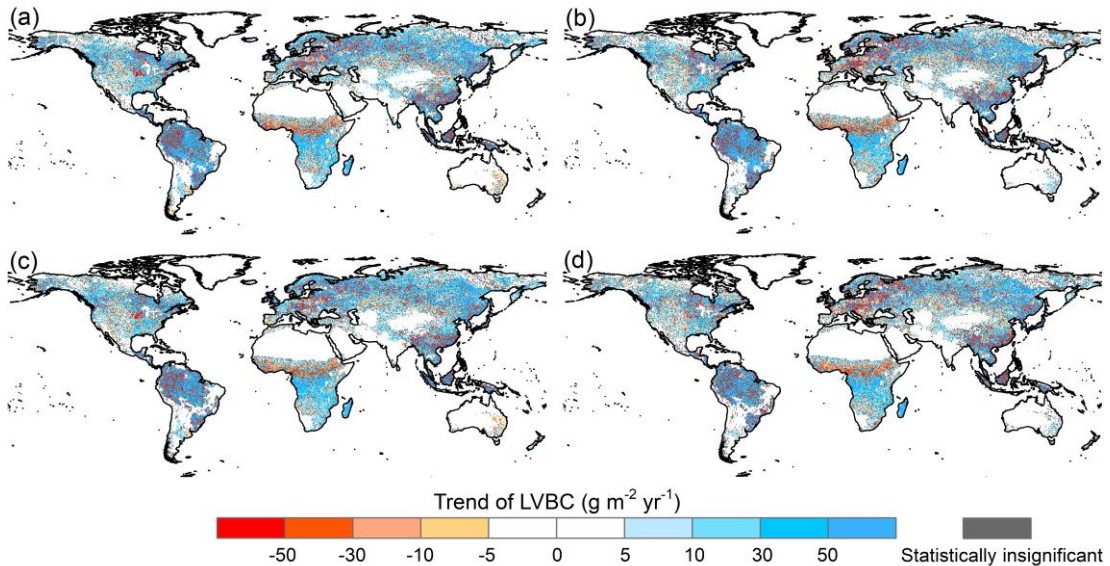

Figure A2. Potential LVBC trend maps during the period of 1916 to 2015 under different factorial simulations. (a) $CO_2$ driving factorial simulation (S2); (b) $CO_2$+precipitation driving factorial simulation (S3); (c) $CO_2$+temperature driving factorial simulation (S4); and (d) $CO_2$+radiation driving factorial simulation (S5). Positive values indicate increasing trends in the ratio, and vice versa. All results from Mann-Kendall and Sen's slope statistical tests correspond to the 95% confidence interval.


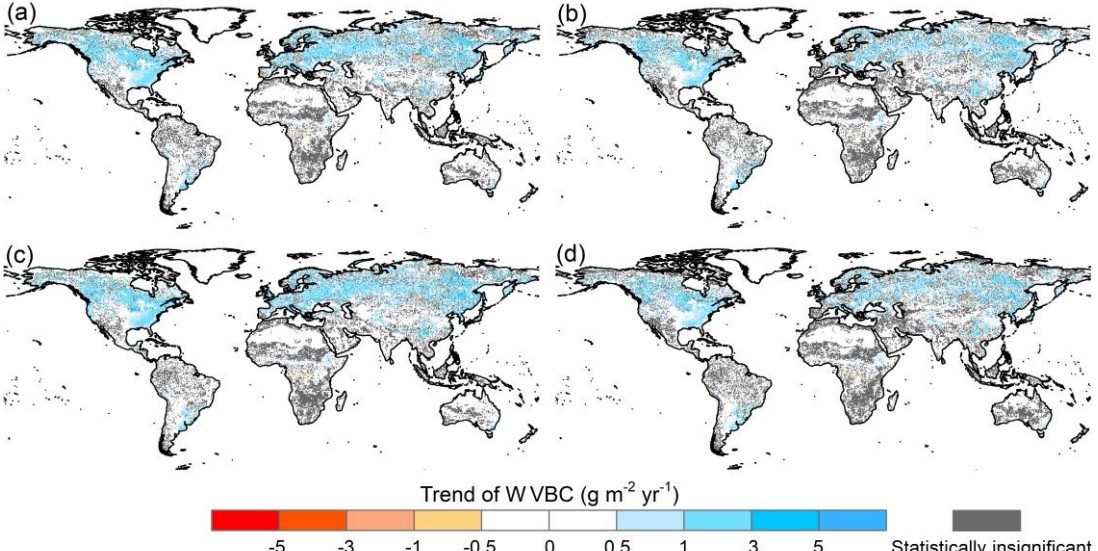

Figure A3. Potential WVBC variation trend maps during the period of 1916 to 2015 under different factorial simulations. (a) $CO_2$ driving factorial simulation (S2); (b) $CO_2$+precipitation driving factorial simulation (S3); (c) $CO_2$+temperature driving factorial simulation (S4); and (d) $CO_2$+radiation driving factorial simulation (S5). Positive values indicate increasing trends in the ratio, and vice versa. All results from Mann-Kendall and Sen's slope statistical tests correspond to the 95% confidence interval.


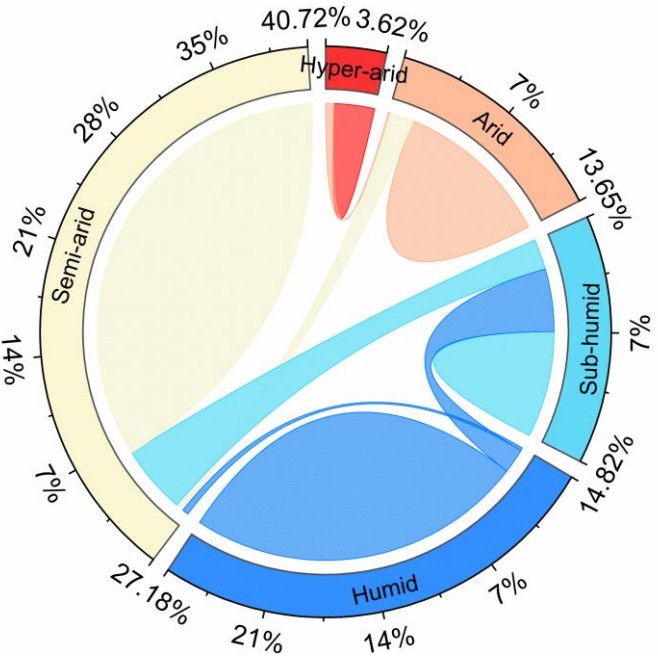

Figure A4. The shift of hydrological regions defined by the multiyear average AI index from the period of 1916-1945 to the period of 1986-2015. The outermost number represent the percentage of hydrological regions in 1916-1945.


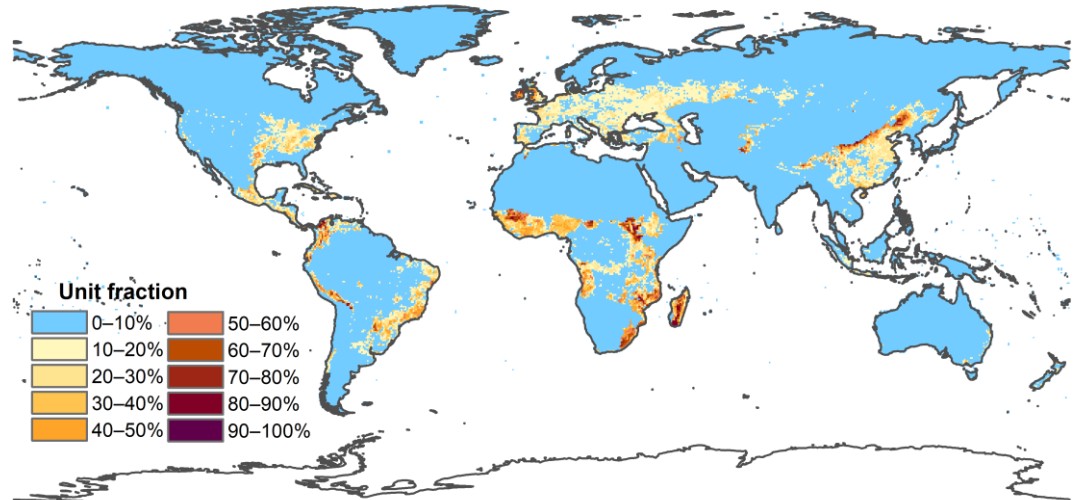

Figure A5. Spatial distribution of multi-year average fraction of managed pasture from 2001-2015 at $0.5 \times 0.5$ arc-degree resolution.


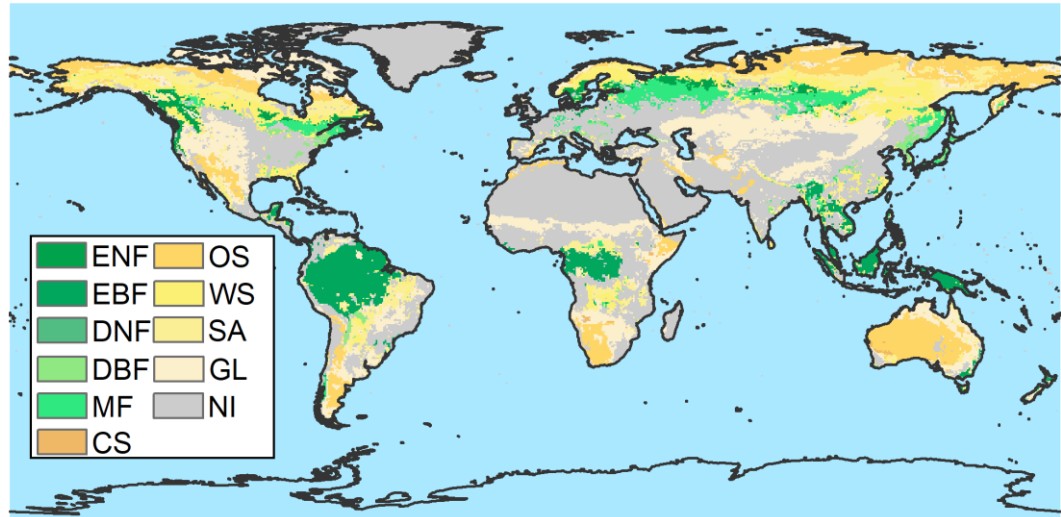

Figure A6. Map showing the largest vegetation component of each grid cell without anthropogenic disturbance from MCD12C1 and LUH2. ENF: Evergreen needleleaf forest, EBF: Evergreen broadleaf forest, DNF: Deciduous needleleaf forest, DBF: Deciduous broadleaf forest, MF: Mixed forest, CS: Closed shrublands, OS: Open shrublands, WS: Woody savannas, SA: Savannas, GL: Grasslands, NI: Not included, which means the zone is not covered by vegetation without anthropogenic disturbance.


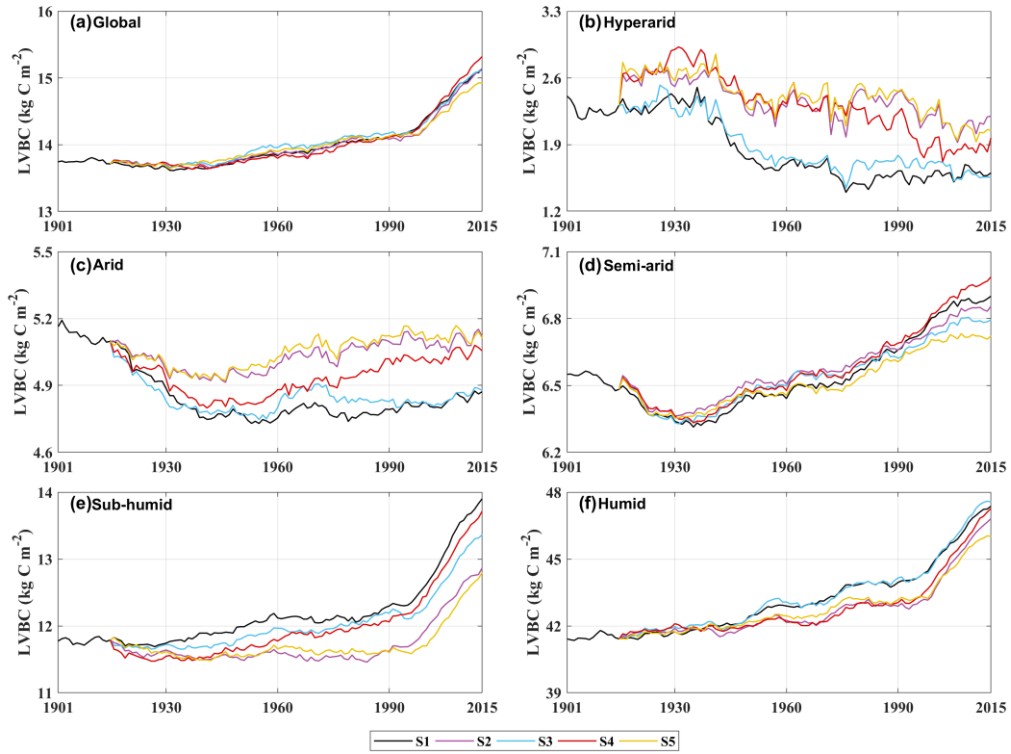

Figure A7. Trends in average density of potential LVBC. (a) Modelled trend of annual averaged LVBC globally. Modelled trends in annual averaged LVBC in hyper-arid region (b), arid region (c), semi-arid region (d), sub-humid region (e), and humid region (f).


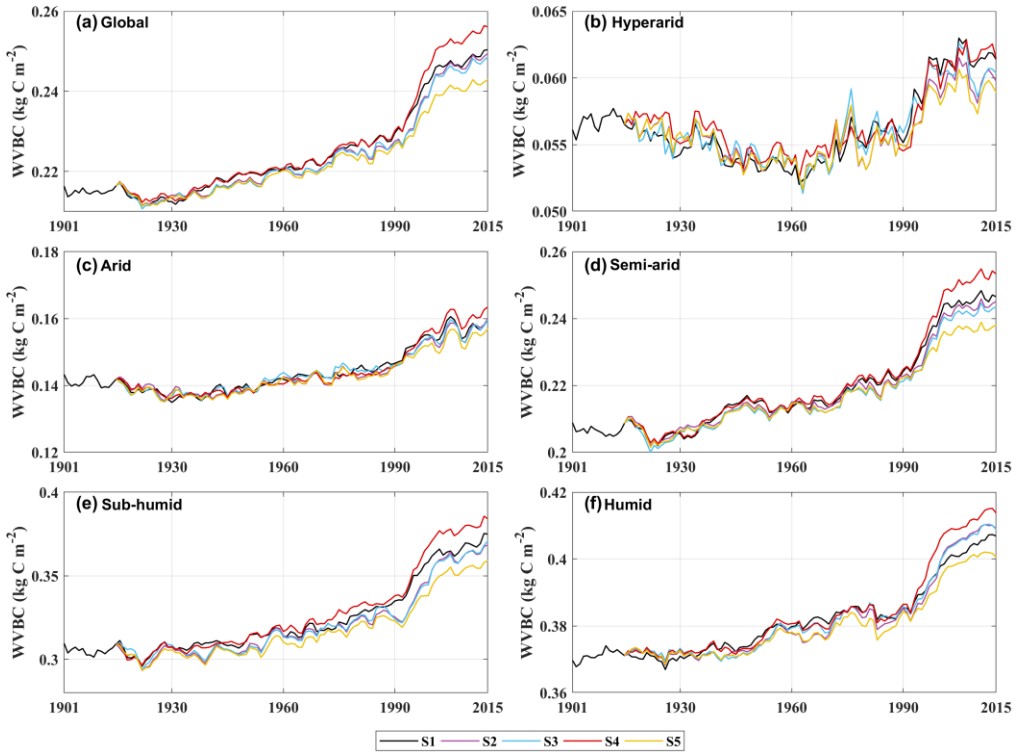

Figure A8. Trends in average density of potential WVBC. (a) Modelled trend of annual averaged WVBC globally. Modelled trends in annual averaged WVBC in hyper-arid region (b), arid region (c), semi-arid region (d), sub-humid region (e), and humid region (f).

**Code and data availability statement**

The code of SEIB-DGVM version 3.02 can be download from http://seib-dgvm.com/. Climatic Research Unit data can be downloaded from https://crudata.uea.ac.uk/cru/data/hrg/. The soil physical parameters can be downloaded from www.iges.org/gswp. The reconstructed $CO_2$ concentration dataset and SEIB code can be downloaded from http://seib-dgvm.com/. In model validation, Ecosystem Model-Data Intercomparison (multiyear average NPP product) data were collected from https://daac.ornl.gov/NPP/guides/NPP_EMDI.html. Remote sensing product MOD17A3 data were obtained from https://lpdaac.usgs.gov/products/mod17a3hgfv006/, MCD12C1 data were obtained from https://ladsweb.modaps.eosdis.nasa.gov/search/order, and LUH2 data were obtained from https://luh.umd.edu/.

**Authors contributions**


T.S. designed research. T.S., and S.H. performed research and developed the methodology. T.S. analyzed
data and produced the outputs. T.S., S.H., C.J., and X.C. wrote the first manuscript draft. W.W. and W.G.
supervised the study. All the authors discussed the methodology and commented on various versions of
the manuscript.
**Competing interests**
The authors declare that they have no conflict of interest.
**Acknowledgments**
This work was jointly supported by the National Natural Science Foundation of China (Grant Nos.
51979071, 51779073, 91547205, U2240218), the National Key Research and Development Program of
China (2018YFA0605402, 2021YFC3201100), the Distinguished Young Fund Project of Natural
Science Foundation of Jiangsu Province (BK20180021), the QingLan Project of Jiangsu Province, the
National "Ten Thousand Program" Youth Talent, and the "333 project" of Jiangsu Province. We thank
Zefeng Chen for technical support. We gratefully thank the following data providers and model
developers for their continuous efforts and for sharing their data: the University of East Anglia, the
National Centers for Environmental Prediction (NCEP), the National Oceanic and Atmospheric
Administration (NOAA), University of Maryland, and the Center for Ocean-Land-Atmosphere Studies
(COLA). Cordial thanks are extended to the editor, Dr. Hans Verbeeck, and two anonymous referees for
the valuable comments which greatly improve the quality of the paper.

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
