# Peer review of "storage potential of vegetation under limited water"

_Geoscientific Model Development, 2021_

## Referee Comment (RC2)

**Review: "Impact of changes in climate and CO$_2$ on the carbon-sequestration potential of vegetation under limited water availability using SEIB-DGVM version 3.02"**

**General comments**

In this manuscript, the authors perform simulations with the dynamic global vegetation model SEIB-DGVM to explore the impact of historical changes in climate and atmospheric CO$_2$ concentration on potential carbon sequestration in live vegetation. Intriguingly, they look not just at total biomass, but also "aboveground" vs. "belowground" biomass (although those terms are misleading; see below). This allows the authors to examine how plants have shifted their growth strategies over the last century to maintain a competitive edge under environmental change.

The results show that both biomass pools have increased, but with "belowground" increasing more than "aboveground" on a relative basis. Factorial experiments reveal that atmospheric CO$_2$ increase is unsurprisingly the dominant driver of potential biomass increase in most of the world, but temperature and other factors are more important at latitudes above 60°N. The results also show that "aboveground" and "belowground" responses to environmental change differ along an aridity gradient, as well as from each other.

The authors designed a suite of experiments well-suited to explore how plant individuals and communities have changed their growth strategies to deal with environmental change. However, the manuscript needs substantial rework. Most importantly, while the Introduction briefly mentions previous findings regarding shifts in above- and belowground allocation under environmental change, this should build up to a set of hypotheses that are then tested with the model experiments. It is also unclear why this was submitted to *Geoscientific Model Development*. Perhaps if it were more focused on comparing SEIB-DGVM biomass to observations it would fit as an evaluation paper, but the work performed is much more high-level than that. I thus think it would be more appropriate to move to *Biogeosciences*.

For these and other reasons that I will elaborate below, I think this manuscript should be ***reconsidered after major revisions*** and ***moved to a different journal***.

**Specific comments**

**Theoretical grounding and hypothesis testing**

The experiments and analyses in this paper are well designed to explore how changing climate and CO$_2$ concentrations, both individually and in aggregate, affect plant allocation strategies over time. However, the authors need to do a better job connecting the two. The Introduction should walk the reader through theory about *why* plants allocate photosynthesized C to different biomass pools. What's currently there (L 70–84) is insufficient; for example, the reader should understand *why* (or at least, a theory of why) Ma et al. (2021) found what they did. This introduction should then lead into specific hypotheses about what theory suggests the experiments will show. In the rest of the paper, the

experimental descriptions, results, and discussion should continually connect back to those hypotheses. As it is, the paper feels disjointed and directionless, with the results not sufficiently linked to any sort of theoretical framework.

This is worsened by the inclusion of seemingly extraneous information on some figures. Specifically, Figs. 10–12 and A3–4 include extra plots or lines showing the results of every factorial experiment, but I didn't see these discussed anywhere in the text. I can think of reasons why it would be useful to explore the effect of $CO_2$ and climate drivers in each of these—for the purpose of testing hypotheses, this might not require *every* factorial experiment on each figure—but the authors did not do so. I strongly recommend augmenting the results and discussion to explore the implications of what we see from the different factorial in these figures. If not, the not-discussed results should be removed from the figures (although perhaps the full figures could be put in a Supplement).

The authors should also consider breaking their analyses up based on plant growth form. The allocation strategies of grasses are constrained relative to what trees can do, since the former lack woody stems. Combining the two growth forms in analysis muddies the interpretation of the results.

Some miscellaneous notes I made that are related to my comments here:

- L 395–403: This belongs more in the Introduction than in Results. And it should be explained *why* these changes occur.
- L 412: *Why* does fine root mass correlate with temperature?

**"Aboveground" vs. "belowground"**

In the last section, the authors note that they consider it a "limitation" that SEIB-DGVM doesn't separate "trunk" (i.e., wood) biomass into stem and coarse root pools. I had actually thought the authors put all wood biomass into the "aboveground" pool intentionally, and that this was an interesting and novel idea! Yes, coarse roots are of course belowground, but these experiments and analyses are designed to explore how plants change their allocation strategies to improve competitiveness in terms of resource-gathering. For that reason, the coarse roots serve an "aboveground" *purpose* that is aligned with the overall goal of wood allocation (at least in theory)—to grow taller than one's neighbors in order to outcompete them for light. (See, e.g., Dybzinski et al. [2011, *Am. Naturalist*] and other work from those authors.) Coarse roots literally support this strategy as they anchor trees into the ground, so any investment in growing taller must also have a corresponding investment in coarse roots, lest the trees become vulnerable to uprooting.

Because all wood biomass—even coarse roots—is in the "aboveground" pool, I think that that and "belowground" are misleading as labels. The authors should rename them to something that better reflects the theoretical purpose of allocating to these different pools. Based on my background and understanding, I'd probably go with something like "light-gathering" and "soil-exploiting" (fine roots are the only belowground pool, and they enable uptake of water and N), respectively. However, the authors should choose labels that are appropriate to whatever theoretical framework and hypotheses they choose to lay out.

While I think this is a powerful and meaningful distinction, its novelty makes it difficult to compare SEIB-DGVM outputs to previous results from the literature. However, that's not necessarily a problem either. Some other DGVMs also don't really distinguish between aboveground and belowground wood in terms of allocation strategy: for example, LPJ-GUESS only makes the distinction (using a global constant for all trees) for the purposes of fire fuel calculations, wood harvest, and transfer of killed biomass to litter/soil pools. The authors should use the literature to classify wood into truly above- or belowground pools in post-processing using even something as simple as a global constant, then augment their evaluation results and experimental discussion to compare to previous findings.

Note that my thoughts here make less sense if SEIB-DGVM actually models tap roots and does so as part of the woody pool. If tap roots aren't modeled at all, this missing process should be mentioned at L 515.

**Methods (description) issues**

The MODIS NPP evaluation methods are complex and should be moved, along with the other NPP evaluation methods, to a new Methods subsection. (This would also resolve the current problem where the authors start describing the methods, then talk about the results, then finish talking about the methods.) There are a number of issues with the description here (and implications for discussion):

- The authors should clarify exactly what steps they used, in what order, to isolate "undisturbed" land (more on that in the next bullet point) in the MODIS data. Were cells excluded from the NPP dataset before or after aggregation to 0.5°? This should have occurred at the native 500-m MODIS resolution.
- What MODIS land cover classification scheme was used?
- Exactly what land cover types were considered "disturbed?" (The authors only list "undisturbed" types.) Clearly, urban and crop land should have been, but including all grasslands is going to include a lot that is grazed by livestock. To some extent and in some regions, livestock has simply replaced wild grazers, but that's not universally true qualitatively or quantitatively.
    - The authors should mention how including land grazed by livestock might affect their estimates of "potential" biomass. What kind of bias does this result in, and where might it be strongest? Maps of grazed area and grazer density can illuminate the latter.
    - Consider filtering the 0.5° gridcells in the analysis based on some threshold of pasture fraction, which can be obtained for example from the LUH2 land use data. That's actually at 0.25° resolution, so perhaps this filtering could happen at an intermediate step before aggregation to 0.5°, potentially enabling more 0.5° cells to be included.
- Minor notes:
    - L 304: This says MODIS NPP comparison period starts 2000, but Fig. 3 caption says 2001.
    - L 308–310: Simplify (this can be one sentence), and clarify why you're saying this (because MODIS data include used land).

In Fig. 4 and related text, it's not specified over what time period the SEIB output was averaged. This is critical to understanding how it compares to previous findings. SEIB value should *not* cover more than, say, 30 years; then you should exclude literature values outside that range. Or consider instead a scatter plot, with each literature value vs. SEIB *at the time the literature value refers to*.

Other comments on methods:

- It should be clarified somewhere what is meant by "potential" carbon stocks. Presumably this means "in the absence of human land use."
- L 153–155: Is tree growth daily or monthly? If both, I guess those are different growth processes? Please clarify.
- L 169–176: This paragraph purports to outline advantages of SEIB-DGVM compared to other models, but I think it should just be deleted.
  - L 169–171: I interpret this to mean that SEIB-DGVM includes size-mediated competition for light, but it's not the only DGVM that does this. See, for example, LPJ-GUESS (Smith et al., 2001, *Glob. Ecol. & Biogeog.*) and LM3-PPA/LM4 (Weng et al., 2015, *Biogeosciences*). See also Fisher et al. (2018, *Glob. Chg. Biol.*) for a review of such "vegetation demographic models."
  - L 171–172 made me think that the simulations would start with PFT composition and structure derived from observations, but later it seems that this is not the case ("SEIB-DGVM simulations begin with seeds of selected plant function types planted in bare ground. The plant functional types are favored for establishment by the environmental conditions in each grid cell."). Please edit this sentence to clarify that such inputs *can* be used.
  - L 172–174: Unclear what this is trying to say. Is it that SEIB-DGVM can't do land use? If so, that's not an advantage—in models that have land use, it can be disabled for potential vegetation runs if desired.
- Sect. 2.3 (description of relevant processes in SEIB-DGVM): It should be clarified what of this is new to SEIB-DGVM in this paper vs. what was already there.
- L 226: It's a bit surprising that plant demand doesn't actually enter into the calculation of water limitation status. The assumption seems to be that plants are always stressed, to some extent unless, the soil is fully saturated. I guess this is more of a comment about the Discussion: The authors should discuss the implications of this. It would seem to contribute to a bias of SEIB-DGVM towards greater fine root allocation.
- L 277–279: It's very unclear what this test of "detection trends" actually is.
- L 518–524: Unclear. Did you not include N deposition at all? Wouldn't that mean that you've *under*estimated $CO_2$ fertilization? Please elaborate the N deposition methods (or lack thereof) in Methods and clarify this text.

**Results interpretation issues**

There are a number of places in the Results where I thought the authors' interpretations were either incorrect, confusing, or insufficient:

- L 338: Slower? Slower than what? $CO_2$ change looks *faster* than biomass C change.

- L 415: Reference to Figs. 8–9 here is inappropriate, as those maps don't show attribution.
- L 419–426: Short-term variation is a completely different thing from long-term trend; it's unclear why they're being lumped together here (where it says the bit about temporal compensation).
- L 427–429:
  - If there's no long-term trend in precipitation and radiation (as asserted at L 422–423), how can they induce a long-term change?
  - Precipitation effect appears to *not* be compensatory for "above-ground" biomass, which is most biomass! (Again at L 489–490.)
- Figs. 10–11 and related text:
  - Regression tests for trends in mean and standard deviation across AI bins would be useful. I'd suggest trying a linear fit for Fig. 10 and a quadratic fit for Fig. 11.
  - L 438–439:
    - It seems to me like an increasing trend in this difference would indicate that more water-limited areas experience *more enhanced* C growth, not less.
    - …Although the trends are very small!
    - This sentence is very confusingly written. "Fluctuations" I think might be the reason. This connotes year-to-year changes rather than a trend, which is what the figures are actually looking at.
  - Discuss: Why is there a (slight) increasing trend for AVBC but a (slight) unimodal pattern for BVBC?

**Other specific comments**

- Throughout
  - Use of "integrated" and "integral" throughout is confusing. Do you mean "total," as in AVBC+BVBC?
  - "carbon-stocks" should be "carbon stocks" throughout (no hyphen).
  - "Regions" and "regional" to me imply land masses or geopolitical boundaries, but it is often used here to describe latitude bands. It would be better to use "latitude bands/zones" and "zonal" instead.
- L 49–51: It's unclear what the difference is between "direct" and "indirect" effects. This idea of "two mechanisms" is not ever returned to, so I suggest just simplifying this sentence to remove the distinction.
- L 73–75: Abrupt transition to talking about models was confusing.
- L 76: "negative response to climate" is vague.
- L 80–82: This sentence is vague (what is "oversensitivity"?) and seemingly unsupported.
- L 182: What is "stock" biomass? This is missing from Fig. A1 and its caption.
- L 215: How frequently? Annually?
- L 238: "adjusted"? What is the usual method?

- Table 1: Please replace the heading "$CO_2$ fertilization" with "$CO_2$ concentration" for consistency (referring to environmental conditions rather than plant processes).
- L 282: "We defined"… not really, right? Isn't this the same as used in Chen et al. (2019)?
- Fig. A2:
  - What is "no value"?
  - Please increase weight of font in legend.
- L 332–333: Is this just repeated from Methods? If so, delete. If not, elaborate (and move to Methods).
- L 337–338: How was 2.44 calculated? Mean annual range? Be more specific.
- L 340–341: Specify R-squared and p-values for AVBC and BVBC as well.
- Fig. 5
  - a:
    - Is the inset plot just the pink line? Clarify this in the caption.
    - Please use a color other than pink, as it's hard to tell from the red.
    - Please change "Dynamic of biomass carbon" to match the clearer label on the inset plot ("Biomass carbon")
    - Please add units to Y-axes in inset plot.
    - Consider just removing the inset plot. It doesn't really add much except potential for confusion. This would also allow zooming in on the biomass Y-axes to provide better visibility.
  - Caption: "during the first decade; the averaged value (1916–1925, red line) and the last decade averaged value"
- L 365: "further supports." Further? Where was this mentioned before? Should be mentioned in Sect. 3.2, where it becomes obvious that AVBC will dominate because it's so much higher.
- L 366–367: "the proportion of the total change in carbon-stocks is small (3.08 ± 0.14 Pg C)"—what does this mean? "Proportion" makes me think you're talking about a fraction, but the units are PgC.
- Fig. 6:
  - Colors on map are fine, but cells should be gray (or otherwise distinguished) if p-value is not significant. And then the inset bar graphs should not have colors, because that's confusing with the map colors; distinguish increasing vs. decreasing trends instead with text.
  - Note different color scale for (c) in caption.
  - Increase resolution so that pixels aren't blurred.
- L 386–393: What of this is coming from Fig. 7a/c? It's not referred to anywhere in the text.
- Fig. 7: Include labels on figure for above- vs below-ground.
- L 419: Start a new paragraph here at "Previous", to provide some separation between pure results and discussion.
- Figs. 8–9:
  - Gray out pixels without a significant trend.
  - Increase resolution so that pixels aren't blurred.

- o I don't think these are actually discussed anywhere except L 415, which I think is inappropriate because they don't actually deal with this aspect of attribution. Add some discussion of them in the Results.
  - o Might be more useful to replace these with mapped versions of Fig. 7, showing the fraction of the trend contributed by each factor in each gridcell.
- "Modelled AVBC enhanced magnitude" throughout is unclear.
- Figs. 10–11: Specify what a negative vs. a positive value means on the Y axis.
- L 432–435: Remove citations from this sentence. Add actual discussion (in a new paragraph after your results in this subsection) of how your results compare to the literature.
- L 435–438: Combining these sentences would increase readability.
- L 442–446: What does "drivers attributed to increase (A/B)VBC changed" mean?
- L 457–465: This should be in the Introduction.
- L 470: What is "terrestrial water"? Where did you do this?
- L 502: What are "indirect factors"?
- L 502–505: Second part of this sentence seems unrelated to the first.
- L 505: "lowers"? Relative to what?
- L 516: "in factorial simulations"? What does that have to do with anything?
- L 517–518: Why is this a limitation?

**Technical corrections**

- L 199: Second and third commas should be semicolons.
- L 200: Comma should be a semicolon.
- L 205: "are" should be "is"
- L 213: Tilde should be an en dash.
- L 253: Should be "functional"
- L 265: "trend" is there twice
- L 327: "form" should be "from"
- L 348: Tropical
- L 382: Comma should be a semicolon.
- L 405: Specify Fig. 7a
- L 423: "variant" should be "variation".
- L 453: Delete "that"; "spatial" should be "temporal"

---

## Author Comment (AC2)

**Response to reviewer**
* * *
**Reviewer # Questions and our responses**

**We extend our deep appreciation to Reviewer for the constructive comments and suggestions toward improving our paper. Acknowledgement was added in the revision.**

**Reviewer:**

This paper reports a set of factorial simulations of global vegetation biomass responses to changes in atmospheric $CO_2$, temperature, precipitation, and radiation based on a well-developed dynamic vegetation model, SEIB-DGVM. The purpose of this study is to "systematically determine the long-term variability of carbon-sequestration potential and understand its response mechanisms, and estimate trends in partitioning of potential biomass carbon-stocks of vegetation biomass".

However, after reading through this paper a couple of times, I do not think these questions are answered. The authors should keep it in mind that these results are simulations from a model. One cannot just run the model and tell us what they are. The simulations must be correctly evaluated before taken as conclusions. A detailed analysis of simulation results, model formulation, and uncertainty evaluation is necessary either in Results or in Discussion.

I also had a hard time in following the description of model description (Section 2.3 Carbon-stock of vegetation biomass partitioning). Please improve this section.

Response: We greatly appreciate your detailed summary and excellent comments which helped us to clarify our logic flow and presentation.

Feedbacks from terrestrial ecosystem to greenhouse effect noticeably strengthen carbon-sequestration potential. However, the enhanced trend and drivers of carbon-sequestration potential at the global scale in past hundred years is unclear. To answer the question that how carbon-sequestration potential of vegetation biomass partitioning would respond to the impact of changes in climate and carbon dioxide ($CO_2$), we used the spatially explicit individual-based dynamic global vegetation model (SEIB-DGVM) as research tool to simulate the historical trend of potential vegetation carbon-stock and verified modelled result. Then, we set factorial simulations to isolate and quantify the contributions of changes in climate and $CO_2$ to the variation of the carbon-sequestration. Based on the results of factorial simulations, we found that the interaction of terrestrial water availability and driving factors ($CO_2$, precipitation, temperature, radiation) adjusts the response magnitude of carbon-stock to changes in driving factors. We suggested that the long-term trend in increased vegetation biomass carbon-stocks is driven by $CO_2$ fertilization and temperature effects that are controlled by water limitations.

SEIB-DGVM is the first biogeochemical model with three-dimensional representation of forest structure (Sato et al. Ecological Modelling, 2007, 200(3-4): 279-307), and has been widely used in simulating carbon cycle and vegetation succession. In this research, we evaluated the SEIB-DGVM version 3.02 and used it to investigate the variation trend and drivers' contributions of vegetation carbon-stock partitioning. Due to the limitations in empirical formulas and coefficients, the artificial uncertainty still exists in SEIB-DGVM. So, we first verified the accuracy of SEIB-DGVM. The evaluation results shown that simulated value calculated by SEIB-DGVM has a high degree of consistency with observed value. Based on prior knowledge, we found that debate and uncertainty exist in the estimation of global potential vegetation carbon-stock. Compared with these results from the literature and the state-of-the-art dataset, the potential vegetation carbon-stock modelled by SEIB-DGVM was within the reasonable range. According to verification results, we thought that SEIB-DGVM is an available research tool, which could supply a way to investigate the change trend and drivers'

contributions of carbon-sequestration potential. We discussed uncertainty-induced result errors in simulation of vegetation carbon-stock (see Original, Page 14, Lines 513-524), while more detailed discussion and model description would be added into revision to help readers understand better.

Point-to-point responses to all the comments are given below.

1. line 89: "Large gaps in our knowledge of the effects of various drivers on the partitioning of carbon-stocks in vegetation biomass remain." Through this paper, the definition of "carbon-stock" is confusing. If it is referred to as the biomass, you do not have to use it. Just use "biomass".

Response: Thanks for your constructive suggestion. Yes, the definition of "carbon-stock" is the carbon content of biomass. With the increase of atmospheric $CO_2$ concentration, the capacity of vegetation carbon sequestration remarkably enhanced to maintain a balanced carbon cycle. Biomass increase is one of manifestation of the enhanced carbon sequestration capacity. To reveal the feedback from vegetation carbon sequestration to increase atmospheric $CO_2$ concentration, we thought carbon-stock is an appropriate proxy for the carbon-sequestration capacity.

2. There Lines 123~124 "Neither the CRU nor NCEP datasets included downward shortwave and longwave radiation." I used these data and I know they have downward shortwave and longwave radiation at 6-hourly time step. Go to TRENDY site, where you can find the links to these data.

Response: Thanks for your careful review and valuable suggestion. The climate forcing of TRENDY obtains from CRUNCEP, which is a combination of CRU monthly data and NCEP reanalysis data. Based on empirical functions (Sato et al, Ecological Modelling, 2007, 200(3-4): 279-307.), we employed historical data from CRU and NCEP to calculate the shortwave radiation and the longwave radiation at midday. We

would conduct a comparation about radiation from the CRUNCEP and empirical functions in revision.

3.  Line 177: "Carbon-stock of vegetation biomass partitioning" I think it does not have to say "carbon-stock" if it means carbon content of biomass. Biomass can be defined as unit of carbon (e.g., kg carbon per unit of land)

Response: Thanks for your careful review and constructive suggestion. As the $CO_2$ concentration rise, the vegetation carbon sequestration capacity increases for moderating $CO_2$ concentration buildup and stabilizing carbon cycle. Changes in vegetation biomass is an aspect of the variant of carbon sequestration capacity. To reveal the interaction among the components of carbon cycle, carbon-stock is an appropriate proxy for investigating the response of vegetation carbon sequestration capacity to other components' changes.

4.  Line 194: I am not clear about this equation. Does "(

$$max_1 = \left(crown_{area} + \pi crown_{diameter} crown_{depth}\right) \frac{LA_{max}}{SLA}$$

)" have any physical meaning? LAmax seems to be a maximum leaf area. However, it is said to be "maximum leaf area of PFTs per unit biomass ($m^2$ $m^{-2}$)," per unit biomass of what? Why is the unit $m^2$ $m^{-2}$?

Response: Thanks for your valuable comment. According to previous literature (Sato et al, Ecological Modelling, 2007, 200(3-4): 279-307., Page 287), terms in parenthesis indicate a surface area of tree crown (except basal plane), which is assumed to have a cylinder shape. By multiplying the crown-surface-area ($m^2$) by maximum-leaf-area per unit crown-surface-area ($LA_{max}$, $m^2$ $m^{-2}$), we have the maximum-leaf-area ($m^2$) that size of the crown of the tree allows. By dividing it with specific-leaf-area (SLA, $m^2$ $g^{-1}$), the maximum-leaf-area is converted into maximum-lead-biomass (g) of the crown. $LA_{max}$ is the plant functional type specific maximum leaf area per unit crown surface area excluding the bottom soffit ($m^2$ $m^{-2}$).

5. Line 204: "Grass leaf biomass is supplemented"? stop to grow?

Response: Thanks for your detailed comment. In plant growth phrase, the non-structural carbon of photosynthetic production is allocated to grass leaf consistently. When the leaf area index of grass equals the optimal leaf area index, it stops to allocate non-structural carbon to grass leaf. The grass leaf biomass stops to grow.

6. Line 206: Any scientific basis for this equation? Why is it like this?

Response: Thanks for your detailed comment. The equation (5) in Line 206 comes from previous literature (Sato et al, Ecological Modelling, 2007, 200(3-4): 279-307., Page 288, equation (36)). The $lai_{opt}$ represents the daily net primary production is the maximizes value under this leaf area index, derived from $gpp_g$ – cost $lai_g$/SLA.
The $gpp_g$ is the daily gross primary production (g day$^{-1}$ m$^{-2}$. Kuroiwa, Function and Productivity of Plant Population. Asakura-shoten, Tokyo, pp. 84-141 (in Japanese)). The cost is the maintenance respiration rate per unit biomass (dimensionless). The $lai_g$ is the leaf area index of the grass layer (m$^2$ m$^{-2}$). SLA is the PFT-specific leaf area per unit biomass (m$^2$ g$^{-1}$)
The equation (36) is derived from equation (19) and (34) according to Sato *et al.* (2007), calculated as follows:

$$gpp_g = 0.090936 \int_{y=0}^{lai_g} \int_{t=0}^{dlen} p_{single}\, dt\, dy$$

$$= 0.090936 \frac{2 dlen\, p_{sat}}{eK} \ln\left(\frac{1+\sqrt{1+(par_{grass}eK\, lue/p_{sat})}}{1+\sqrt{1+(par_{grass}eK\, lue/p_{sat})e^{-eK\, lai_g}}}\right) \quad (19)$$

$$stat_{leaf} = gpp_l - cost\, \frac{la}{SLA} \frac{1}{10\, crown_{depth}} \quad (34)$$

$$lai_{opt} = \frac{\ln par_{grass} - \ln\{p_{sat}/lue[(1-(cost/SLA)/0.09093 dlen\, p_{sat})^{-2}-1]\}}{eK} \quad (36)$$

where $gpp_g$ is gross primary production of grass layer (g day$^{-1}$ m$^{-2}$), $stat_{leaf}$ is benefit per cost of maintaining leaf mass (g g$^{-1}$ day$^{-1}$), $gpp_l$ is gross primary production of each crown layer (g day$^{-1}$), $cost$ is the cost of maintaining leaves per unit leaf mass per day (g DM g DM$^{-1}$ day$^{-1}$), $la$ is leaf area (m$^2$), $crown_{depth}$ is

crown depth (m), $lai_{opt}$ is optimal leaf area index (m² m⁻²), $par_{grass}$ is the grass photosynthetically active radiation ( μ mol photon m⁻² s⁻¹), $p_{sat}$ is the light-saturated photosynthetic rate ( μ CO₂ m⁻² s⁻¹), $lue$ is the light-use efficiency of photosynthesis (mol CO₂ mol photon⁻¹), $dlen$ is day length (hour), and $eK$ is light attenuation coefficient at midday.

7. Lines 214~215: This sentence is funny "When total woody biomass is more than 10 kg DM, which defines the minimum tree size for reproduction, 10% of non-structural carbon is transformed into litter." The authors are talking about "reproduction" limit of biomass, and then they tell you if this requirement is met, some non-structure carbon (NSC) will be converted to litter. Then, what is reproduction? Is it "10% of non-structural carbon is transformed into seeds"?

Response: Thanks for your detailed comment. The SEIB-DGVM assumes that each tree consumes 10% of NSC for reproduction. This 10% NSC is used for every process of reproduction, including having flowers, pollen, nectar, fruits, and seeds. As reproduction is a very diverse process for plant species, this kind of assumption is required for simulating vegetation at large geographic scales. This simple assumption of the SEIB-DGVM was actually taken from the LPJ-DGVM, and many other DGVMs also employ this assumption.

8. Lines 216: "the remaining structural carbon is allocated to sapwood biomass" What is "structural carbon"?

Response: Thanks for your detailed comment. We reworded the sentence "the remaining non-structural carbon is allocated to sapwood biomass" in the revision.

9. Lines 222 "Terrestrial water availability represents a significant source of variability in the ecosystem carbon cycle" This sentence is not necessary.

Response: Thanks for your detailed suggestion. We deleted the sentence in the revision.

10. Lines 232 "According to the flexible allocation scheme, SEIB-DGVM allocates and stores the biomass carbon …" the phrase "According to the flexible allocation scheme," is not needed.

Response: Thanks for your detailed suggestion. We deleted the phrase in the revision.

11. Lines 236~238 This sentence disrupts the description of model formulation. Reword it.

Response: Thanks for your detailed suggestion. We reworded the sentence in the revision as below:

"The root-shoot ratio (R/S) has been used to distinguish and investigate the variation trend of inner biomass partitioning of carbon-sequestration potential (Zhang et al., 2016)."

12. Lines 253~254: "The plant functional types are favored for establishment by the environmental conditions in each grid cell." Reword this sentence. I could not understand what it wants to say. Does it mean the environmental conditions will select out PFT(s) in each grid cell?

Response: Thanks for your detailed comment and constructive suggestion. Yes, the establishment of PFT(s) is determined by environmental conditions. For example, the SEIB-DGVM assumes that boreal broad-leaved summer-green trees can only establish when the midday photosynthetically active radiation that averaged for the previous year exceeded 700 $\mu$ mol photon m$^{-2}$ s$^{-1}$ at the surface of the grass layer.

We reworded the sentence in the revision as below:

"The establishment of plant functional types are determined by the climatic conditions in each grid cell."

13. Lines 260~268: section "Factorial simulation scheme" Clarify this section please. It is really difficult to understand it.

Response: Thanks for your detailed comment and constructive suggestion. We added more information about "Factorial simulation scheme" in order to help readers understand better.

"In order to further quantify the relative contributions of varying atmospheric $CO_2$ concentrations, precipitation, temperature, radiation, and other factors, we performed six factorial simulations. Other factors included wind velocity and relative humidity, which had remarkable effects on change in vegetation carbon-stock at regional scale. In simulation S1, atmospheric $CO_2$ concentration and all of climate variables were varied. In simulation S2, only atmospheric $CO_2$ concentration was varied, and climate variables were held constant (Climate variables of the transient period (1901-1915) were repeatedly inputted). In simulation S3 (or S4, S5), atmospheric $CO_2$ and precipitation (or temperature, radiation) were varied, and other climate variables were held constant. In simulation S6, atmospheric $CO_2$, wind velocity, and relative humidity were varied, and other climate variables were held constant. Finally, S2 was used to evaluate the effects of $CO_2$ fertilization on carbon-stock variation. The differences of S2-S3, S2-S4, S2-S5, and S2-S6 were used to evaluate the response of carbon-stock growth to precipitation, temperature, radiation, and other drivers, respectively."

14. Line 260: What are "Other drivers" in Table 1? You only listed "atmosphere $CO_2$, precipitation, temperature, and radiation". Specify them please.

Response: Thanks for your detailed suggestion. Other drivers included wind velocity and relative humidity (see Original, Page 10, Lines 263-264).

Follow your suggestion, I added more introduction of other factors in Table 1. We rewrote the sentence as below:

"In the last simulation S6, historical atmospheric $CO_2$ concentrations and other climate variables (wind velocity and relative humidity) were input, excluding precipitation, temperature, and radiation."

15. Line 265: What is "carbon-stocks trend"?

Response: Thanks for your detailed comment. We rewrote the sentence, "the contribution of $CO_2$ to the trend in carbon-stocks trend" was changed to "the contribution of $CO_2$ to the trend in carbon-stocks".

16. Line 295: I don't understand "In terrestrial vegetation biomes, there is a high correlation between biomass carbon-stock density and NPP per unit (Erb et al., 2016; Kindermann et al., 2008)". Why does it need "In terrestrial vegetation biomes"? per unit of what?

This is supposed to be the results. Why does it have citation here?

Response: We greatly appreciate your insightful comment. To evaluate the accuracy of potential vegetation carbon-stock modelled by SEIB-DGVM, we tried to find a dataset including global potential vegetation carbon-stock from 1916 to 2015. To our best effort, we only collected the spatial pattern of potential vegetation carbon-stock (see Original, Page 14, Lines 326-329). According to previous conclusions (Erb et al., 2016; Kindermann et al., 2008), we knew that there is a high correlation between net primary production (NPP) and carbon-stock in regions covered by vegetation. We cited literatures to reveal this correlation of NPP and carbon-stock, and used NPP as a proxy of the carbon-stock to assess model accuracy. We collected long-term series observations of NPP to verify the modelled NPP of SEIB-DGVM from 1916 to 2015. The general agreement of observed NPP and modelled NPP suggested that it is possible

to use the SEIB-DGVM model to evaluate the long-term trend of potential vegetation carbon-stock.

17. Lines 295~302: If this paragraph is to describe another dataset, it should be Method and Data section.

Response: Thanks for your constructive suggestion, the section was moved to Method and Data.

18. Lines 303~314: Same for this section. Move it to the data analysis method section.

Response: Thanks for your constructive suggestion, the section was moved to Method and Data.

19. Lines 331~332: Move to Method section.

Response: Thanks for your constructive suggestion, the section was moved to Method and Data.

20. Lines 335~336: I am confused by the definition of "carbon-stock". Is it new growth of biomass or the biomass a plant has?

Response: Thanks for your constructive suggestion. Vegetation carbon-stock is the content of vegetation. The turnover time of carbon-stock in trunk organ is more than one year. The year-to-year assimilative carbon is stored at vegetation organs until it is transformed into litter. We changed "the year-to-year accumulation of carbon in the terrestrial plant without external interference" to "plant grow in the assumed absence of external interference under current climate".

21. Line 349 "a conclusion consistent with prior knowledge (Erb et al., 2018; Schimel et al., 2015)" should be in discussion.

Response: Thanks for your constructive suggestion, the section was moved to Discussion and Conclusion.

22. Lines 355~357 "Based on the carbon-stock partitioning method, we found that the integrated carbon-stock as well as the above- and belowground carbon-stocks over the period of 1916–2015 exhibited a remarkable spatial heterogeneity." This sentence does not have information. Say it directly: What the spatial pattern is.

Response: Thanks for your constructive suggestion, the detailed information of spatial pattern was added in the revision as follow:

"Integrated carbon-stock and AVBC exhibited an increasing trend in eastern South America, southern Africa, and northern Asia, while declined in central North America, northwest South America, and central Africa. BVBC showed a more widespread increase in North America, southeastern South America, and Europe, while had a decrease trend in part region of Asian."

23. Lines 369~372: "Biomass carbon allocation between above- and belowground vegetation organs reflect the changes in individual growth, community structure and ecosystem function, which are important attributes in the investigation of carbon-stocks and carbon cycling within the terrestrial biosphere (Hovenden et al., 2014; Fang et al., 2010; Ma et al., 2021)" this sentence should be in discussion. Present your own results. Throughout the results section, this type of evaluations to their own results should go to discussion.

Response: Thanks for your constructive suggestion, this section and other similarity sections were moved to Discussion and Conclusion in the revision.

24. Line 466 "4 Conclusions and discussions" Change it to "4 Discussion and conclusion".

Response: Thanks, we changed "4 Conclusions and discussions" to "4 Discussion and conclusion".

25. For a modeling paper, the uncertainty of simulations should be evaluated. One cannot pretend these simulations to be the sure thing and "offer perspectives" based on them directly. Many patterns are just artifacts from model assumptions and model response equations, which are highly uncertainty.
For example, in line 495, the authors found "the long-term change in carbon-stocks is tightly coupled to terrestrial water availability". Then, it should be talked about that how the model simulates water effects on vegetation and to what extent this formulation can be trusted.

Response: We greatly appreciate the reviewer's insightful comment. Yes, there is uncertainty in the simulation results of SEIB-DGVM because of model assumptions and empirical equations. To assess the effects of uncertainty on model, we evaluated the simulation accuracy of NPP and potential vegetation carbon-stock. Based on the result of verification, we thought that SEIB-DGVM is an available research tool, which could supply a way to investigate the change trend and drivers' contributions of vegetation carbon-stock.

Based on photosynthesis, plant assimilated carbon and allocated non-structural carbon to plant organs. In SEIB-DGVM, terrestrial water availability affected vegetation carbon-stock by controlling leaf phenology and the rate of photosynthesis. We added detailed information about the effect of water limitation on vegetation in SEIB-DGVM, equations as follow:

$$p_{sat} = PMAX ce_{tmp} ce_{co_2} ce_{water}$$

$$ce_{water} = \sqrt{stat_{water}}$$

$$stat_{water} = \frac{max\left(pool_{w(1)}/Depth_{(1)}, \ pool_{w(2)}/Depth_{(2)}\right) - W_{wilt}}{W_{fi} - W_{wilt}}$$

where $p_{sat}$ is the single-leaf photosynthetic rate ( μ mol $CO_2$ m$^{-2}$ s$^{-1}$); $PMAX$ is the potential maximum of photosynthetic rate ( μ mol mol$^{-1}$ $CO_2$ m$^{-2}$ s$^{-1}$); $ce_{tmp}$ and $ce_{co_2}$ are the temperature and $CO_2$ concentration effect coefficient (dimensionless), separately (Raich et al, Ecological Applications, 1991, 1(4), 399–429. Brooks and Farquhar, Planta, 1985, 165(3), 397–406.); $ce_{water}$ is the water effect coefficient (dimensionless); $stat_{water}$ is the physiological status of water availability (dimensionless).

Recent version of the SEIB-DGVM appropriately reproduce geographical distributions of GPP (gross primary production) and biomass in the African continent, where plant productivity and structures are mainly controlled by aridity (Sato and Ise 2012, Sato et al. 2015). These results demonstrate that the model appropriately treats water effects on vegetation.

Sato, H. and T. Ise (2012). "Effect of plant dynamic processes on African vegetation responses to climate change: Analysis using the spatially explicit individual-based dynamic global vegetation model (SEIB-DGVM)." Journal of Geophysical Research-Biogeosciences 117(G3): 202-215.

Sato, H., et al. (2015). "Effects of different representations of stomatal conductance response to humidity across the African continent under warmer $CO_2$-enriched climate conditions." Journal of Geophysical Research-Biogeosciences 120(5): 979-988.

**Thanks again for your time and efforts put on this manuscript, which is acknowledged in the paper**

---

## Author Response (AR1)

**Response to reviewers**
* * *
**Reviewers #1 Questions and our responses**

**We extend our deep appreciation to Reviewer #1 for the constructive comments and suggestions toward improving our paper. Acknowledgement is added in the revision.**

**Reviewer #1:**

This paper reports a set of factorial simulations of global vegetation biomass responses to changes in atmospheric $CO_2$, temperature, precipitation, and radiation based on a well-developed dynamic vegetation model, SEIB-DGVM. The purpose of this study is to "systematically determine the long-term variability of carbon-sequestration potential and understand its response mechanisms, and estimate trends in partitioning of potential biomass carbon-stocks of vegetation biomass".

However, after reading through this paper a couple of times, I do not think these questions are answered. The authors should keep it in mind that these results are simulations from a model. One cannot just run the model and tell us what they are. The simulations must be correctly evaluated before taken as conclusions. A detailed analysis of simulation results, model formulation, and uncertainty evaluation is necessary either in Results or in Discussion.

I also had a hard time in following the description of model description (Section 2.3 Carbon-stock of vegetation biomass partitioning). Please improve this section.

Response: We greatly appreciate your detailed summary and excellent comments which helped us to clarify our logic flow and presentation.

Feedbacks from terrestrial ecosystem to greenhouse effect noticeably strengthen carbon storage potential. However, the enhanced trend and drivers of inner carbon storages potential at the global scale in past hundred years is unclear. To answer the question that how partitioning components of vegetation carbon stock would respond to the impact of changes in climate and carbon dioxide ($CO_2$), we used the spatially explicit individual-based dynamic global vegetation model (SEIB-DGVM) as research tool to simulate the historical trend of potential vegetation carbon-stock and verified modelled result. Then, we set factorial simulations to isolate and quantify the contributions of changes in climate and $CO_2$ to the variation of the carbon stocks. Based on the results of factorial simulations, we found that the interaction of terrestrial water availability and driving factors ($CO_2$, precipitation, temperature, radiation) adjusts the response magnitude of carbon stocks to changes in driving factors. We suggested that the long-term trend in increased vegetation biomass carbon stocks is driven by $CO_2$ fertilization and temperature effects that are controlled by water limitations.

SEIB-DGVM is the first biogeochemical model with three-dimensional representation of forest structure (Sato et al. Ecological Modelling, 2007, 200(3-4): 279-307), and has been widely used in simulating carbon cycle and vegetation succession. In this research, we evaluated the SEIB-DGVM version 3.02 and used it to investigate the variation trend and drivers' contributions of vegetation carbon stocks partitioning. Due to the limitations in empirical formulas and coefficients, the artificial uncertainty still exists in SEIB-DGVM. So, we first verified the accuracy of SEIB-DGVM. The evaluation results shown that simulated value calculated by SEIB-DGVM has a high degree of consistency with observed value. Yes, errors and uncertainties exist in the estimation of global potential vegetation carbon stocks simulated by SEIB-DGVM. Compared with these results from the literature and the state-of-the-art dataset, the potential vegetation carbon stock modelled by SEIB-DGVM was within the reasonable range. According to verification results, we thought that SEIB-DGVM is an available research tool, which could supply a way to investigate the change trend and drivers' contributions of carbon storage potential. We added more detailed model description

and discussion about uncertainty-induced result errors in simulation of vegetation carbon stocks to help readers understand better.

Point-to-point responses to all the comments are given below.

1.  line 89: "Large gaps in our knowledge of the effects of various drivers on the partitioning of carbon-stocks in vegetation biomass remain." Through this paper, the definition of "carbon-stock" is confusing. If it is referred to as the biomass, you do not have to use it. Just use "biomass".

Response: Thanks for your constructive suggestion. Yes, the definition of "carbon-stock" is the carbon content of biomass. With the increase of atmospheric $CO_2$ concentration, the capacity of vegetation carbon sequestration remarkably enhanced to maintain a balanced carbon cycle. Biomass increase is one of manifestation of the enhanced carbon sequestration capacity. To reveal the feedback from vegetation carbon sequestration to increase atmospheric $CO_2$ concentration, we thought carbon-stock is an appropriate proxy for the carbon storage and sequestration capacity.

2.  There Lines 123~124 "Neither the CRU nor NCEP datasets included downward shortwave and longwave radiation." I used these data and I know they have downward shortwave and longwave radiation at 6-hourly time step. Go to TRENDY site, where you can find the links to these data.

Response: Thanks for your careful review and valuable suggestion. The forcing data of SEIB-DGVM is the shortwave radiation and the longwave radiation at midday, and I added explanation in section 2.1. Based on empirical functions (Sato et al, Ecological Modelling, 2007, 200(3-4): 279-307.), we employed historical data from CRU and NCEP to calculate the shortwave radiation and the longwave radiation at midday. The climate forcing of TRENDY obtains from CRUNCEP, which is a combination of CRU monthly data and NCEP reanalysis data. After the comparison, the magnitudes of the shortwave radiation and the longwave radiation at midday are smaller than that of daily

shortwave radiation and daily longwave radiation. So, the radiation of TRENDY is unable to run SEIB-DGVM.

We added more detailed illustration of the radiation data in the section 2.1 to help readers understand better.

(1) "Long-term daily meteorological time-series data are required to run model simulations, including precipitation, daily range of air temperature, mean daily air temperature, downward shortwave radiation at midday, downward longwave radiation at midday, wind velocity and relative humidity." (see Revision, Page 5, Lines 120-122)

(2) "Neither the CRU nor NCEP datasets included downward shortwave and longwave radiation at midday. Thus, daily cloudiness values in the NCEP were used to calculate radiation values using empirical functions (Sato et al., 2007)." (see Revision, Page 5, Lines 130-132)

3. Line 177: "Carbon-stock of vegetation biomass partitioning" I think it does not have to say "carbon-stock" if it means carbon content of biomass. Biomass can be defined as unit of carbon (e.g., kg carbon per unit of land)

Response: Thanks for your careful review and constructive suggestion. Yes, the biomass can represent the change in carbon. Changes in vegetation biomass is an aspect of the variant of carbon sequestration and storage capacity. Carbon stock is the dry-mass carbon content. To reveal the effect of water limitations on vegetation carbon, carbon-stock is an appropriate proxy for investigating the response of vegetation carbon storage and sequestration capacity to changes in climate and $CO_2$, especially in arid regions.

4. Line 194: I am not clear about this equation. Does "(

$$max_1 = \left(crown_{area} + \pi crown_{diameter} crown_{depth}\right) \frac{LA_{max}}{SLA}$$

)" have any physical meaning? LAmax seems to be a maximum leaf area. However, it is said to be "maximum leaf area of PFTs per unit biomass ($m^2$ $m^{-2}$)," per unit biomass of what? Why is the unit $m^2$ $m^{-2}$?

Response: Thanks for your valuable comment. According to previous literature (Sato et al, Ecological Modelling, 2007, 200(3-4): 279-307., Page 287), terms in parenthesis indicate a surface area of tree crown (except basal plane), which is assumed to have a cylinder shape. By multiplying the crown-surface-area ($m^2$) by maximum-leaf-area per unit crown-surface-area ($LA_{max}$, $m^2$ $m^{-2}$), we have the maximum-leaf-area ($m^2$) that size of the crown of the tree allows. By dividing it with specific-leaf-area (SLA, $m^2$ $g^{-1}$), the maximum-leaf-area is converted into maximum-lead-biomass (g) of the crown. We are sorry about making mistake here, and we changed "$LA_{max}$ is maximum leaf area of PFTs per unit biomass ($m^2$ $m^{-2}$)" (see Original, Page 8, Line 200) to "$LA_{max}$ is the plant functional type specific maximum leaf area per unit crown surface area excluding the bottom soffit ($m^2$ $m^{-2}$);" (see Revision, Page 8, Lines 206-207)

5.  Line 204: "Grass leaf biomass is supplemented"? stop to grow?

Response: Thanks for your detailed comment. In plant growth phrase, the non-structural carbon of photosynthetic production is allocated to grass leaf consistently. When the leaf area index of grass equals the optimal leaf area index, it stops to allocate non-structural carbon to grass leaf. The grass leaf biomass stops to grow. We added more detailed illustration as below:
"When the leaf area index of grass equals the optimal leaf area index, it stops to allocate non-structural carbon to grass leaf, which is calculated as:" (see Revision, Page 8, Lines 210-211)

6.  Line 206: Any scientific basis for this equation? Why is it like this?

Response: Thanks for your detailed comment. The equation (5) in Line 206 comes from previous literature (Sato et al, Ecological Modelling, 2007, 200(3-4): 279-307., Page 288, equation (36)). The $lai_{opt}$ represents the daily net primary production is the maximizes value under this leaf area index, derived from $gpp_g$ – cost $lai_g$/SLA.

The $gpp_g$ is the daily gross primary production (g day$^{-1}$ m$^{-2}$. Kuroiwa, Function and Productivity of Plant Population. Asakura-shoten, Tokyo, pp. 84-141 (in Japanese)). The cost is the maintenance respiration rate per unit biomass (dimensionless). The $lai_g$ is the leaf area index of the grass layer (m$^2$ m$^{-2}$). SLA is the PFT-specific leaf area per unit biomass (m$^2$ g$^{-1}$)

The equation (36) (equation (5) in original manuscript) is derived from equation (19) and (34) according to Sato $et$ $al.$ (2007), calculated as follows:

$$gpp_g = 0.090936 \int_{y=0}^{lai_g} \int_{t=0}^{dlen} p_{single}\, dt\, dy$$

$$= 0.090936 \frac{2\, dlen\, p_{sat}}{eK} \ln \left( \frac{1+\sqrt{1+(par_{grass} eK\, lue/p_{sat})}}{1+\sqrt{1+(par_{grass} eK\, lue/p_{sat})e^{-eK\, lai_g}}} \right) \quad (19)$$

$$stat_{leaf} = gpp_l - cost\, \frac{la}{SLA}\, \frac{1}{10\, crown_{depth}} \quad (34)$$

$$lai_{opt} = \frac{\ln par_{grass} - \ln\{p_{sat}/lue[(1-(cost/SLA)/0.09093 dlen\, p_{sat})^{-2}-1]\}}{eK} \quad (36)$$

where $gpp_g$ is gross primary production of grass layer (g day$^{-1}$ m$^{-2}$); $stat_{leaf}$ is benefit per cost of maintaining leaf mass, g g$^{-1}$ day$^{-1}$; $gpp_l$ is gross primary production of each crown layer, g day$^{-1}$; $cost$ is the cost of maintaining leaves per unit leaf mass per day, g DM g DM$^{-1}$ day$^{-1}$; $la$ is leaf area, m$^2$; $crown_{depth}$ is crown depth, m; $lai_{opt}$ is optimal leaf area index, m$^2$ m$^{-2}$; $par_{grass}$ is the grass photosynthetically active radiation, μ mol photon m$^{-2}$ s$^{-1}$; $p_{sat}$ is the light-saturated photosynthetic rate, μ CO$_2$ m$^{-2}$ s$^{-1}$; $lue$ is the light-use efficiency of photosynthesis, mol CO$_2$ mol photon$^{-1}$; $dlen$ is day length, hour; and $eK$ is light attenuation coefficient at midday.

7.  Lines 214~215: This sentence is funny "When total woody biomass is more than 10 kg DM, which defines the minimum tree size for reproduction, 10% of non-structural carbon is transformed into litter." The authors are talking about "reproduction" limit of biomass, and then they tell you if this requirement is met, some non-structure carbon (NSC) will be converted to litter. Then, what is reproduction? Is it "10% of non-structural carbon is transformed into seeds"?

Response: Thanks for your detailed comment. The SEIB-DGVM assumes that each tree consumes 10% of NSC for reproduction. This 10% NSC is used for every process of reproduction, including having flowers, pollen, nectar, fruits, and seeds. As reproduction is a very diverse process for plant species, this kind of assumption is required for simulating vegetation at large geographic scales. This simple assumption of the SEIB-DGVM was actually taken from the LPJ-DGVM, and many other DGVMs also employ this assumption. We added more detailed illustration as below:

"When total woody biomass is more than 10 kg DM, which defines the minimum tree size for reproduction. This 10% NSC is used for every daily process of reproduction, including having flowers, pollen, nectar, fruits, and seeds." (see Revision, Page 8, Lines 220-222)

8. Lines 216: "the remaining structural carbon is allocated to sapwood biomass" What is "structural carbon"?

Response: Thanks for your detailed comment. We are very sorry for our incorrect writing. "the remaining structural carbon" was changed into "the remaining non-structural carbon" (see Revision, Page 8, Line 223)

9. Lines 222 "Terrestrial water availability represents a significant source of variability in the ecosystem carbon cycle" This sentence is not necessary.

Response: Thanks for your detailed suggestion. We removed the sentence in the revision.

10. Lines 232 "According to the flexible allocation scheme, SEIB-DGVM allocates and stores the biomass carbon …" the phrase "According to the flexible allocation scheme," is not needed.

Response: Thanks for your detailed suggestion. We removed the phrase in the revision.

11. Lines 236~238 This sentence disrupts the description of model formulation. Reword it.

Response: Thanks for your detailed suggestion. We reworded the sentence in the revision as below:

"The root-shoot ratio (R/S) has been used to distinguish and investigate the ratio of below-ground biomass (root biomass) and above-ground biomass (shoot biomass) (Zhang et al., 2016)." (see Revision, Page 9, Lines 246-248)

12. Lines 253~254: "The plant functional types are favored for establishment by the environmental conditions in each grid cell." Reword this sentence. I could not understand what it wants to say. Does it mean the environmental conditions will select out PFT(s) in each grid cell?

Response: Thanks for your detailed comment and constructive suggestion. Yes, the establishment of PFT(s) is determined by environmental conditions. For example, the SEIB-DGVM assumes that boreal broad-leaved summer-green trees can only establish when the midday photosynthetically active radiation that averaged for the previous year exceeded 700 $\mu$ mol photon $m^{-2}$ $s^{-1}$ at the surface of the grass layer.

"The establishment of PFTs seeds are determined by the climatic conditions in each grid cell." (see Revision, Page 10, Lines 267-268)

13. Lines 260~268: section "Factorial simulation scheme" Clarify this section please. It is really difficult to understand it.

Response: Thanks for your detailed comment and constructive suggestion. We added more detailed illustration about "Factorial simulation scheme" in order to help readers understand better as below:

"In order to further quantify the relative contributions of varying atmospheric $CO_2$ concentrations, precipitation, temperature, radiation, and other factors, we performed six factorial simulations. Other factors included wind velocity and relative humidity, which had remarkable effects on the change in vegetation carbon stock at zonal scale. In simulation S1, atmospheric $CO_2$ concentration and all of climate variables were varied. In simulation S2, only atmospheric $CO_2$ concentration was varied, and climate variables were held constant (Climate variables of the transient period (1901-1915) were repeatedly inputted). In simulation S3 (or S4, S5), atmospheric $CO_2$ and precipitation (or temperature, radiation) were varied, and other climate variables were held constant. In simulation S6, atmospheric $CO_2$, wind velocity, and relative humidity were varied, and other climate variables were held constant. Finally, S2 was used to evaluate the effects of $CO_2$ fertilization on carbon stock variation. The differences of S2-S3, S2-S4, S2-S5, and S2-S6 were used to evaluate the response of carbon stock growth to precipitation, temperature, radiation, and other drivers, respectively." (see Revision, Pages 10-11, Lines 274-285)

14. Line 260: What are "Other drivers" in Table 1? You only listed "atmosphere $CO_2$, precipitation, temperature, and radiation". Specify them please.

Response: Thanks for your detailed suggestion. Other drivers included wind velocity and relative humidity. Follow your suggestion, we added the explanation about other factors in Table 1 as below:

"In the last simulation S6, historical atmospheric $CO_2$ concentrations and other climate variables were input, including wind velocity and relative humidity." (see Revision, Page 10, Table 1)

15. Line 265: What is "carbon-stocks trend"?

Response: Thanks for your detailed comment. We are sorry about making mistake here,

and we changed the sentence to "on the change in vegetation carbon stock". (see Revision, Page 11, Lines 276-277)

16. Line 295: I don't understand "In terrestrial vegetation biomes, there is a high correlation between biomass carbon-stock density and NPP per unit (Erb et al., 2016; Kindermann et al., 2008)". Why does it need "In terrestrial vegetation biomes"? per unit of what?

This is supposed to be the results. Why does it have citation here?

Response: We greatly appreciate your insightful comment. To evaluate the accuracy of potential vegetation carbon-stock modelled by SEIB-DGVM, we tried to find observation dataset including global potential vegetation carbon stock from 1916 to 2015. To our best effort, we only collected the spatial pattern of potential vegetation carbon-stock (see Original, Page 14, Lines 326-329). According to previous conclusions (Erb et al., 2016; Kindermann et al., 2008), we knew that there is a high correlation between net primary production (NPP) and carbon-stock in regions covered by natural vegetation. We cited literatures to reveal this correlation of NPP and carbon stock, and used NPP as a proxy of the carbon stock to assess model accuracy. In this study, we collected a long-term series monitoring data from EMDI and MODIS to verify the modelled NPP from SEIB-DGVM between 1916 and 2015. The general agreement of observed NPP and modelled NPP in vegetation regions without anthropogenic disturbance indicated that it is possible to use the SEIB-DGVM model to evaluate the long-term trend of potential vegetation carbon stock.

17. Lines 295~302: If this paragraph is to describe another dataset, it should be Method and Data section.

Response: Thanks for your constructive suggestion, the section was moved to Method and Data as below:

"A global time series of potential vegetation carbon was modelled by the SEIB-DGVM between 1916-2015. In terrestrial vegetation biomes, there is a high correlation between biomass carbon stock density and NPP per unit (Erb et al., 2016; Kindermann et al., 2008) (Figure A1). Thus, we collected NPP observation dataset and used NPP as a proxy of the carbon stock to assess model accuracy. Ecosystem Model-Data Intercomparison (EMDI) builds upon the accomplishments of the original worldwide synthesis of NPP measurements and associated model driver data prepared by Global Primary Production Data Initiative. We obtained the monitoring station data from the EMDI working group, and then compared their data with modelled multiyear average NPP in the period of 1916-1999 (Figure 2)." (see Revision, Pages 12-13, Lines 311-318)

18. Lines 303~314: Same for this section. Move it to the data analysis method section.

Response: Thanks for your constructive suggestion, the section was moved to Method and Data, and we added more illustration in the revision as below:

"However, *in-situ* observations are sparse for global spatial-temporal validation. Therefore, we used the MOD17A3 products to further verify the simulated potential NPP in the twenty first century. These data were collected by the Moderate Resolution Imaging Spectroradiometer and are some of the most widely used data to assess the accuracy of global model simulations (Gulbeyaz et al., 2018). The natural vegetation zones refer to the hypothetical condition that would prevail in an assumed absence of anthropogenic activity, but under historical climate fields (Erb et al., 2018; Haberl et al., 2014). The potential NPP is defined as that assimilated carbon stored in natural vegetation without the disturbance of anthropogenic activities (Erb et al., 2018).

In order to distinguish the distribution of vegetation zones without anthropogenic disturbance, we obtained global land cover types in the period 2001-2015 from MCD12C1 (Table A1). It was defined as vegetation grid that the land cover type of this grid is evergreen needleleaf forest, evergreen broadleaf forest, deciduous needleleaf

forest, deciduous broadleaf forest, mixed forest, closed shrublands, open shrublands, woody savannas, savannas or grasslands. Grid covered by other 7 land types was defined as non-vegetation grid. Then, we calculated the proportion of each land cover types in corresponding 0.5° grid unit. The land cover type of grid unit was determined by the max proportion among 17 land cover types. Part of grids covered by grassland were grazed by livestock, leading to the decrease of NPP of grass PFTs. We obtained land-use forcing data from Land-Use Harmonization (LUH2) to map the distribution of managed pasture data from 2001 to 2015 (Hurtt et al., 2020). As shown in Figure A4, grassland in eastern Asia, western Europe, south central Africa, and western South America were severely affected by grazing. To exhibit the disturbance of managed pasture, we calculated the mean fraction of managed pasture within the corresponding 0.5° grid unit. When the fraction of managed pasture over 0.01, the grid covered by grassland was considered to be affected by managed pasture. We filtered grassland affected by pasture to map the distribution of natural vegetation zones without anthropogenic disturbance (Figure A5)." (see Revision, Pages 13-14, Lines 319-342)

19.  Lines 331~332: Move to Method section.

Response: Thanks for your constructive suggestion, the section was moved to Method and Data as below.

"A global time series of potential vegetation carbon was modelled by the SEIB-DGVM between 1916-2015." (see Revision, Page 12, Lines 311-312)

20.  Lines 335~336: I am confused by the definition of "carbon-stock". Is it new growth of biomass or the biomass a plant has?

Response: Thanks for your constructive suggestion. Vegetation carbon-stock is the content of vegetation. The turnover time of carbon-stock in trunk organ is more than one year. The year-to-year assimilative carbon is stored at vegetation organs until it is

transformed into litter. We added more detailed illustration about the definition of "potential carbon stock" as below:

"Throughout this study, the potential biomass carbon stock, biomass carbon stored in vegetation without anthropogenic disturbance, is recognized as a proxy for the potential of carbon storage by natural vegetation." (see Revision, Page 4, Lines 104-106).

21. Line 349 "a conclusion consistent with prior knowledge (Erb et al., 2018; Schimel et al., 2015)" should be in discussion.

Response: Thanks for your constructive suggestion, the section was moved to Discussion and Conclusion.

"Compared with WVBC, LVBC increase 116.18 ± 2.34 Pg C and dominates the long-term trends of vegetation carbon stock. The latitudinal bands of increasing annual LVBC are mainly distributed in tropical latitudes, a conclusion consistent with prior knowledge that tropical zones dominate carbon uptake and storage (Erb et al., 2018; Schimel et al., 2015)." (see Revision, Page 26, Lines 519-523)

22. Lines 355~357 "Based on the carbon-stock partitioning method, we found that the integrated carbon-stock as well as the above- and belowground carbon-stocks over the period of 1916–2015 exhibited a remarkable spatial heterogeneity." This sentence does not have information. Say it directly: What the spatial pattern is.

Response: Thanks for your constructive suggestion, the detailed information of spatial pattern was added in the revision as follow:

"In Figures 7(a) and 7(b), total carbon stock and LVBC exhibit a significantly increasing trend in eastern South America, southern Africa, and northern Asia, while declined in central North America, northwest South America, and central Africa. WVBC showed a more widely increasing tendency in North America, southeastern South America, and Europe, while had a decrease trend in part zones of Asian. We find

that the total carbon stock as well as the light- and water-gathering vegetation biomass carbon stocks over the period of 1916–2015 exhibited a remarkable spatial heterogeneity." (see Revision, Page 18, Lines 384-389)

23. Lines 369~372: "Biomass carbon allocation between above- and belowground vegetation organs reflect the changes in individual growth, community structure and ecosystem function, which are important attributes in the investigation of carbon-stocks and carbon cycling within the terrestrial biosphere (Hovenden et al., 2014; Fang et al., 2010; Ma et al., 2021)" this sentence should be in discussion. Present your own results. Throughout the results section, this type of evaluations to their own results should go to discussion.

Response: Thanks for your constructive suggestion, this section and other similarity sections were moved to Discussion and Conclusion in the revision.

"Biomass carbon allocation between light- and water-gathering vegetation organs reflect the changes in individual growth, community structure and ecosystem function, which are important attributes in the investigation of carbon stocks and carbon cycling within the terrestrial biosphere (Hovenden et al., 2014; Fang et al., 2010; Ma et al., 2021). During the past hundred years, the ratio of LVBC/WVBC shown a slight upward trend. The rate of increase is 0.0171 $yr^{-1}$, which is significant at the 0.01 level. To better absorb $CO_2$ and sunlight required for photosynthesis, vegetated regions are gradually covered by vegetation with higher plant height and wider leaf area, thereby adjusting their characteristic ecosystem functions (Anderson et al., 2010)." (see Revision, Pages 26-27, Lines 523-530)

24. Line 466 "4 Conclusions and discussions" Change it to "4 Discussion and conclusion".

Response: Thanks, we changed "4 Conclusions and discussions" to "4 Discussion and conclusion". (see Revision, Page 25, Line 499)

25. For a modeling paper, the uncertainty of simulations should be evaluated. One cannot pretend these simulations to be the sure thing and "offer perspectives" based on them directly. Many patterns are just artifacts from model assumptions and model response equations, which are highly uncertainty.

For example, in line 495, the authors found "the long-term change in carbon-stocks is tightly coupled to terrestrial water availability". Then, it should be talked about that how the model simulates water effects on vegetation and to what extent this formulation can be trusted.

Response: We greatly appreciate the reviewer's insightful comment. Yes, there is uncertainty in the simulation results of SEIB-DGVM because of model assumptions and empirical equations. To assess the effects of uncertainty on model, we evaluated the simulation accuracy of NPP and potential vegetation carbon-stock. Based on the result of verification, we thought that SEIB-DGVM is an available research tool, which could supply a way to investigate the change trend and drivers' contributions of vegetation carbon-stock.

Based on photosynthesis, plant assimilated carbon and allocated non-structural carbon to plant organs. In SEIB-DGVM, terrestrial water availability affected vegetation carbon-stock by controlling leaf phenology and the rate of photosynthesis. We added detailed information about the effect of water limitation on vegetation in SEIB-DGVM, equations as follow:

$$p_{sat} = PMAX \, ce_{tmp} \, ce_{co_2} \, ce_{water}$$

$$ce_{water} = \sqrt{stat_{water}}$$

$$stat_{water} = \frac{max\left(pool_{w(1)}/Depth_{(1)}, \quad pool_{w(2)}/Depth_{(2)}\right) - W_{wilt}}{W_{fi} - W_{wilt}}$$

where $p_{sat}$ is the the single-leaf photosynthetic rate of tree PFTs and grass PFTs ($\mu$ mol $CO_2$ m$^{-2}$ s$^{-1}$); $PMAX$ is the potential maximum of photosynthetic rate ($\mu$ mol mol$^{-1}$ $CO_2$ m$^{-2}$ s$^{-1}$); $ce_{tmp}$ and $ce_{co_2}$ are the temperature and $CO_2$ concentration effect coefficient (dimensionless), separately (Raich et al, Ecological Applications, 1991, 1(4),

399–429. Brooks and Farquhar, Planta, 1985, 165(3), 397–406.); $ce_{water}$ is the water effect coefficient (dimensionless); $stat_{water}$ is the physiological status of water availability (dimensionless); $pool_{w(i)}$ is the water content at soil layer $i$; $Depth_{(i)}$ is the depth of soil layer $i$ (mm),$Depth_{(1)}$ and $Depth_{(2)}$ are 500 mm and 1000 mm, separately; $W_{wilt}$ is the soil moisture at wilting point (m m$^{-1}$); $W_{fi}$ is the soil moisture at field capacity (m m$^{-1}$).

Recent version of the SEIB-DGVM appropriately reproduce geographical distributions of GPP (gross primary production) and biomass in the African continent, where plant productivity and structures are mainly controlled by aridity (Sato and Ise 2012, Sato et al. 2015). These results demonstrate that the model appropriately treats water effects on vegetation.

Sato, H. and T. Ise (2012). "Effect of plant dynamic processes on African vegetation responses to climate change: Analysis using the spatially explicit individual-based dynamic global vegetation model (SEIB-DGVM)." Journal of Geophysical Research-Biogeosciences 117(G3): 202-215.

Sato, H., et al. (2015). "Effects of different representations of stomatal conductance response to humidity across the African continent under warmer $CO_2$-enriched climate conditions." Journal of Geophysical Research-Biogeosciences 120(5): 979-988.

**Thanks again for your time and efforts put on this manuscript, which is acknowledged in the paper**

**Reviewers # 2 Questions and our responses**

We extend our deep appreciation to Reviewer #2 for constructive comments and suggestions toward improving our paper. Acknowledgement was added in the revision.

Reviewer #2:

**General comments**

In this manuscript, the authors perform simulations with the dynamic global vegetation model SEIB-DGVM to explore the impact of historical changes in climate and atmospheric $CO_2$ concentration on potential carbon sequestration in live vegetation. Intriguingly, they look not just at total biomass, but also "aboveground" vs. "belowground" biomass (although those terms are misleading; see below). This allows the authors to examine how plants have shifted their growth strategies over the last century to maintain a competitive edge under environmental change.

The results show that both biomass pools have increased, but with "belowground" increasing more than "aboveground" on a relative basis. Factorial experiments reveal that atmospheric $CO_2$ increase is unsurprisingly the dominant driver of potential biomass increase in most of the world, but temperature and other factors are more important at latitudes above 60°N. The results also show that "aboveground" and "belowground" responses to environmental change differ along an aridity gradient, as well as from each other.

The authors designed a suite of experiments well-suited to explore how plant individuals and communities have changed their growth strategies to deal with environmental change. However, the manuscript needs substantial rework. Most importantly, while the Introduction briefly mentions previous findings regarding shifts in above- and belowground allocation under environmental change, this should build up to a set of hypotheses that are then tested with the model experiments. It is also unclear why this was submitted to Geoscientific Model Development. Perhaps if it were more focused on comparing SEIB-DGVM biomass to observations it would fit as an evaluation paper, but the work performed is much more high-level than that. I thus think it would be more appropriate to move to Biogeosciences.

For these and other reasons that I will elaborate below, I think this manuscript should be reconsidered after major revisions and moved to a different journal.

Response: Thanks for your time and effort to review our manuscript. We greatly appreciate your excellent comments and insightful summary. We made great efforts to revise the manuscript to further improve the quality of the paper.

With substantial changes in this round, we believe our contributions to this topic are more clearly highlighted.

Point-to-point responses to all the comments are given below.

**Specific comments**

1.  Theoretical grounding and hypothesis testing

The experiments and analyses in this paper are well designed to explore how changing climate and $CO_2$ concentrations, both individually and in aggregate, affect plant allocation strategies over time. However, the authors need to do a better job connecting the two. The Introduction should walk the reader through theory about why plants allocate photosynthesized C to different biomass pools. What's currently there (L 70–84) is insufficient; for example, the reader should understand why (or at least, a theory of why) Ma et al. (2021) found what they did. This introduction should then lead into specific hypotheses about what theory suggests the experiments will show. In the rest of the paper, the experimental descriptions, results, and discussion should continually connect back to those hypotheses. As it is, the paper feels disjointed and directionless, with the results not sufficiently linked to any sort of theoretical framework

This is worsened by the inclusion of seemingly extraneous information on some figures. Specifically, Figs. 10–12 and A3–4 include extra plots or lines showing the results of every factorial experiment, but I didn't see these discussed anywhere in the text. I can think of reasons why it would be useful to explore the effect of $CO_2$ and climate drivers in each of these—for the purpose of testing hypotheses, this might not require every factorial experiment on each figure—but the authors did not do so. I strongly recommend augmenting the results and discussion to explore the implications of what

we see from the different factorial in these figures. If not, the not-discussed results should be removed from the figures (although perhaps the full figures could be put in a Supplement).

The authors should also consider breaking their analyses up based on plant growth form. The allocation strategies of grasses are constrained relative to what trees can do, since the former lack woody stems. Combining the two growth forms in analysis muddies the interpretation of the results.

Response: We greatly appreciate your detailed summary and excellent comments which helped us to clarify our logic flow and presentation. According to reviewer's comments in theoretical grounding and hypothesis testing, the modifications we did as follow:

(1) In section 1, we added the explanation about the importance of allocate strategy. According to the optimal partitioning hypothesis, vegetation adjusts the allocation scheme of carbon stocks in organs to improve competitiveness for obtain more resources. The change of allocation scheme effects the response pattern of carbon stocks to changes in climate and $CO_2$. In regions with water stress, plants allocate and storage more carbon at root. Ma et al. (2021) shown that water limitation dominates the proportion of biomass invested in aboveground organs and belowground organs, which is consistent with the optimal partitioning hypothesis. Previous investigations explained that water limitation changes carbon stocks by adjusting allocation scheme, while the effects of limited water resource on the response pattern of carbon stocks to the changes of climate and $CO_2$ is unclear. So, this research focuses on the response patterns of carbon stocks to changes in climate and $CO_2$ in different dry and wet regions. We added more illustration in the revision as below:

"The change of carbon storages in vegetation inner components is not only affected by environmental factors, but also controlled by allocation scheme of assimilated carbon. Fractional dynamics of the carbon stock are widely used as a key indicator to investigate the responses of vegetation to environmental drivers, which also reflect the response strategies of vegetation in environments with different water limitations (Yang et al., 2010). In arid region, vegetation utilizes a tolerance strategy to allocate biomass, storing more biomass carbon in roots to resist enhanced water stress (Chen et al., 2013). Conforming to the optimal partitioning hypothesis, plants store more carbon in shoots and leaves in environments where water is more available and shift more carbon to roots when water is more limited (Yang et al., 2010; Mcconnaughay and Coleman,

1999). Water availability controls both carbon allocation and storage and can potentially transform zones characterized by a positive response to changes in climate and $CO_2$ to zones exhibiting a negative response. For example, global warming stimulates plant productively, Madani et al. (2020) found that there is a dramatically downward trend in the tropical productivity. With increased warming, water limitations are predictable to increasingly reduce the proportion of leaves' biomass, and decrease plant photosynthesis (Ma et al., 2021). Water limitations have a strong regulating effect on the spatial pattern of change in vegetation carbon storage, demonstrating the effects of the changes in climate and $CO_2$ on the dynamics of the plant organs are affected by the terrestrial water gradient. Thus, it is important to systematically investigate the distinct responses of carbon storage potential to changes in climate and $CO_2$ under differing conditions of water stress." (see Revision, pages 3-4, Lines 72-89)

(2) We added more detailed explanation of Figures 9-11 (Figures 10-12 in original manuscript) and Figures A6-7 (Figures 3-4 in original manuscript) in section 3.5 and section 4 as below:

"As shown in Figures 9 and 10, with an increase in the aridity index (i.e., an increase in available water), the magnitude and range in variations of LVBC density and WVBC density gradually enhance. Based on the results of factorial simulations, we find a positive relationship between LVBC and water pressure. In extreme water stress, the increase of LVBC tends to zero and plants stop growing. There is no obvious different in the slopes of fitting curves between factorial simulations. The pattern of the enhanced magnitude and range of variation in the WVBC density is unimodal with water stress gradient in all factorial simulations. With the increasing of AI, the magnitude of change in WVBC increases at first and then decreases finally. The mitigation of water stress promotes WVBC increase, while excess surface water limits the response of WVBC to changes in climate and $CO_2$." (see Revision, Pages 22-23, Lines 459-468)

"Figure 11 illustrates temporal variations in the carbon stock ratio within and between hydrological regions. From hyper-arid region to humid region, the variation range of ratio between LVBC and WVBC significantly increases. Plants store more assimilated carbon in shoots and leaves in humid regions. The long-term effects of driver changes have a positive influence on this carbon allocate pattern." (see Revision, Page 24, Lines 487-491)

"The response pattern of WVBC growth to the increasing water availability is different from that of LVBC. Drought mitigation promotes the growth of WVBC, while humid

region with high light competition limits root growth. The result is consistent with previous finding that plants reduce allocation to roots in dense forests where aboveground competition for light is high (Ma et al. 2021)." (see Revision, Pages 27-28, Lines 556-560)

(3) Thanks for your constructive suggestion that investigating the inner carbon stocks of trees and grasses, separately. This study focuses on the effects of water limitations on the response of inner components of carbon stocks. Only the fine root biomass determines the capacity of vegetation to absorb water. SEIB-DGVM uses the same empirical equations to simulate the effect of water limitation on the growth process of trees and grasses, and the same model assumptions to allocate non-structure carbon to plant organs. Equations of photosynthesis considering water limitations about as follow:

$$p_{sat} = PMAX ce_{tmp} ce_{co_2} ce_{water}$$

$$ce_{water} = \sqrt{stat_{water}}$$

$$stat_{water} = \frac{max(pool_{w(1)}/Depth_{(1)}, \ pool_{w(2)}/Depth_{(2)}) - W_{wilt}}{W_{fi} - W_{wilt}}$$

where $p_{sat}$ is the single-leaf photosynthetic rate of tree PFTs and grass PFTs ($\mu$ mol $CO_2$ m$^{-2}$ s$^{-1}$); $PMAX$ is the potential maximum of photosynthetic rate ($\mu$ mol mol$^{-1}$ $CO_2$ m$^{-2}$ s$^{-1}$); $ce_{tmp}$ and $ce_{co_2}$ are the temperature and $CO_2$ concentration effect coefficient (dimensionless), separately (Raich et al, Ecological Applications, 1991, 1(4), 399–429. Brooks and Farquhar, Planta, 1985, 165(3), 397–406.); $ce_{water}$ is the water effect coefficient (dimensionless); $stat_{water}$ is the physiological status of water availability (dimensionless); $pool_{w(i)}$ is the water content at soil layer $i$; $Depth_{(i)}$ is the depth of soil layer $i$ (mm),$Depth_{(1)}$ and $Depth_{(2)}$ are 500 mm and 1000 mm, separately; $W_{wilt}$ is the soil moisture at wilting point (m m$^{-1}$); $W_{fi}$ is the soil moisture at field capacity (m m$^{-1}$).

So, the combination of trees and grasses forms does not disturb the interpretation of the results.

2.   L 395–403: This belongs more in the Introduction than in Results. And it should be explained why these changes occur.

Response: Thanks for your detailed suggestion. Following your suggestion, we moved

the sentences to the Introduction as below:

"The atmospheric $CO_2$ concentration are affected by the vegetation carbon stock, while the long-term trend of vegetation carbon storage capacity is also affected by the changes in climate and $CO_2$. Since the beginning of industrialization, there has been a noticeable enhancement in the capacity of storing and sequestering carbon, which is needed for stabilizing greenhouse gas concentrations and mitigating global warming (Chen et al., 2019; Pan et al., 2011; Le Noë et al., 2020; Magerl et al., 2019; Bayer et al., 2015; Harper et al., 2018). Due to the interaction between terrestrial vegetation and a changing environment, both photosynthesis and respiration of the vegetation also changed. To better absorb $CO_2$ and sunlight required for photosynthesis, vegetated zones are gradually covered by vegetation with higher plant height and wider leaf area. This change has coincided with a widespread change in other vegetation features, including a positive increase in annual gross primary productivity and a greening of the biosphere (Madani et al., 2020; Zhu et al., 2016)." (see Revision, Pages 2-3, Lines 50-60)

"The change of carbon storages in vegetation inner components is not only affected by environmental factors, but also controlled by allocation scheme of assimilated carbon. Fractional dynamics of the carbon stock are widely used as a key indicator to investigate the responses of vegetation to environmental drivers, which also reflect the response strategies of vegetation in environments with different water limitations (Yang et al., 2010)." (see Revision, Page 3, Lines 72-76)

3. L 412: Why does fine root mass correlate with temperature?

Response: Thanks for your detailed comment. Gill and Jackson (2008) introduced that soil temperature affects the timing and duration of root growth. The onset of production is often keyed by temperature in spring and the maintenance respiration of root increase exponentially with temperature. Root turnover rate of fine roots increases exponentially with mean annual temperature. Above all, there is a correlation between fine root mass and temperature. Meanwhile, we added more detailed illustration in order to help readers understand better.

"The effects of temperature on WVBC are stronger than LVBC, because temperature has a stronger effect on the metabolism process of root growth, dominating the turnover

rate and the costs of maintenance respiration in root growth process (Gill and Jackson, 2000)." (see Revision, Page 21, Lines 445-447)

4.    "Aboveground" vs. "belowground"

In the last section, the authors note that they consider it a "limitation" that SEIB-DGVM doesn't separate "trunk" (i.e., wood) biomass into stem and coarse root pools. I had actually thought the authors put all wood biomass into the "aboveground" pool intentionally, and that this was an interesting and novel idea! Yes, coarse roots are of course belowground, but these experiments and analyses are designed to explore how plants change their allocation strategies to improve competitiveness in terms of resource-gathering. For that reason, the coarse roots serve an "aboveground" purpose that is aligned with the overall goal of wood allocation (at least in theory)—to grow taller than one's neighbors in order to outcompete them for light. (See, e.g., Dybzinski et al. [2011, Am. Naturalist] and other work from those authors.) Coarse roots literally support this strategy as they anchor trees into the ground, so any investment in growing taller must also have a corresponding investment in coarse roots, lest the trees become vulnerable to uprooting.

Because all wood biomass—even coarse roots—is in the "aboveground" pool, I think that that and "belowground" are misleading as labels. The authors should rename them to something that better reflects the theoretical purpose of allocating to these different pools. Based on my background and understanding, I'd probably go with something like "light-gathering" and "soil-exploiting" (fine roots are the only belowground pool, and they enable uptake of water and N), respectively. However, the authors should choose labels that are appropriate to whatever theoretical framework and hypotheses they choose to lay out.

While I think this is a powerful and meaningful distinction, its novelty makes it difficult to compare SEIB-DGVM outputs to previous results from the literature. However, that's not necessarily a problem either. Some other DGVMs also don't really distinguish between aboveground and belowground wood in terms of allocation strategy: for example, LPJGUESS only makes the distinction (using a global constant

for all trees) for the purposes of fire fuel calculations, wood harvest, and transfer of killed biomass to litter/soil pools. The authors should use the literature to classify wood into truly above- or belowground pools in post-processing using even something as simple as a global constant, then augment their evaluation results and experimental discussion to compare to previous findings.

Note that my thoughts here make less sense if SEIB-DGVM actually models tap roots and does so as part of the woody pool. If tap roots aren't modeled at all, this missing process should be mentioned at L 515.

Response: Thanks for your encouraging comments and constructive suggestion. SEIB-DVGM simulate the coarse/tap roots and regards it as the part of the trunk pool. At the beginning of the study, it was a puzzle to us about how to distinct and define the inner components of vegetation carbon stocks. Considering the construction of vegetation, we used "aboveground" and "belowground" to name these inner carbon pools of vegetation. Yes, it is difficult for readers to accurately understand the effects of water limitations on the change of inner carbon stocks based on these terms. Follow your suggestion, we renamed these different carbon pools according to the types of resource-gathering. "Aboveground vegetation biomass carbon-stock (AVBC)" was replaced by "light-gathering vegetation biomass carbon stock (GVBC)", and "belowground vegetation biomass carbon-stock (BVBC)" was changed to "water-gathering vegetation biomass carbon stock (WVBC)" throughout.

Yes, it difficult to evaluate LVBC and WVBC simulated by SEIB-DGVM according to previous results from the literature. So, we compared root biomass carbon and R/S ratio to previous result as below:

"Trunk biomass contains tree branches and structural roots (coarse roots and tap roots) (Sato et al., 2007), so the R/S ratio of potential vegetation in factorial simulations is smaller than the R/S of actual vegetation in observation stations. Root biomass only contains the fine root biomass, leading to an underestimate in belowground organ biomass of trees and grasses compare with previous conclusion (Ma et al., 2021; Yang et al., 2009)." (see Revision, Page 28, Lines 582-586)

5. Methods (description) issues

The MODIS NPP evaluation methods are complex and should be moved, along with the other NPP evaluation methods, to a new Methods subsection. (This would also resolve the current problem where the authors start describing the methods, then talk about the results, then finish talking about the methods.) There are a number of issues with the description here (and implications for discussion):

Response: Thanks for your constructive comment. Following your suggest, we moved the NPP dataset introduction to section 2.5, and added more detailed illustration of the pre-processing process in order to help readers understand better.

Point-to-point responses to all the comments of Methods issues are given below.

6. The authors should clarify exactly what steps they used, in what order, to isolate "undisturbed" land (more on that in the next bullet point) in the MODIS data. Were cells excluded from the NPP dataset before or after aggregation to 0.5°? This should have occurred at the native 500-m MODIS resolution.

Response: Thanks for your detailed comment. The spatial resolution of NPP simulated by SEIB-DGVM is 0.5°. So, we aggregated land cover data from 0.05 to 0.5, and obtained the observed NPP of grids covered by natural vegetation to evaluate the accuracy of simulated NPP. We added this reference in the introduction as below:

"In order to distinguish the distribution of vegetation zones without anthropogenic disturbance, we obtained global land cover types in the period 2001-2015 from MCD12C1 (Table A1). It was defined as vegetation grid that the land cover type of this grid is evergreen needleleaf forest, evergreen broadleaf forest, deciduous needleleaf forest, deciduous broadleaf forest, mixed forest, closed shrublands, open shrublands, woody savannas, savannas or grasslands. Grid covered by other 7 land types was defined as non-vegetation grid. Then, we calculated the proportion of each land cover types in corresponding 0.5° grid unit. The land cover type of grid unit was determined by the max proportion among 17 land cover types." (see Revision, Pages 13-14, Lines 327-334)

7. What MODIS land cover classification scheme was used?

Response: Thanks for your detailed suggestion. We added MCD12C1 classification scheme in Table A1. (see Revision, Pages 30, Table A1)

Table A1. MCD12C1 legend and class descriptions

| Name | Value | Description |
| --- | --- | --- |
| Evergreen Needleleaf Forests | 1 | Dominated by evergreen conifer trees (canopy >2m). Tree cover >60%. |
| Evergreen Broadleaf Forests | 2 | Dominated by evergreen broadleaf and palmate trees (canopy >2m). Tree cover >60%. |
| Deciduous Needleleaf Forests | 3 | Dominated by deciduous needleleaf (larch) trees (canopy >2m). Tree cover >60%. |
| Deciduous Broadleaf Forests | 4 | Dominated by deciduous broadleaf trees (canopy >2m). Tree cover >60%. |
| Mixed Forests | 5 | Dominated by neither deciduous nor evergreen (40-60% of each) tree type (canopy >2m). Tree cover >60%. |
| Closed Shrublands | 6 | Dominated by woody perennials (1-2m height) >60% cover. |
| Open Shrublands | 7 | Dominated by woody perennials (1-2m height) 10-60% cover. |
| Woody Savannas | 8 | Tree cover 30-60% (canopy >2m). |
| Savannas | 9 | Tree cover 10-30% (canopy >2m). |
| Grasslands | 10 | Dominated by herbaceous annuals (<2m). |
| Permanent Wetlands | 11 | Permanently inundated lands with 30-60% water cover and >10% vegetated cover. |
| Croplands | 12 | At least 60% of area is cultivated cropland. |
| Urban and Built-up Lands | 13 | At least 30% impervious surface area including building materials, asphalt, and vehicles. |
| Cropland/Natural Vegetation Mosaics | 14 | Mosaics of small-scale cultivation 40-60% with natural tree, shrub, or herbaceous vegetation. |
| Permanent Snow and Ice | 15 | At least 60% of area is covered by snow and ice for at least 10 months of the year. |
| Barren | 16 | At least 60% of area is non-vegetated barren (sand, rock, soil) areas with less than 10% vegetation. |
| Water Bodies | 17 | At least 60% of area is covered by permanent water bodies. |
| Unclassified | 255 | Has not received a map label because of missing inputs |

8. Exactly what land cover types were considered "disturbed?" (The authors only list "undisturbed" types.) Clearly, urban and crop land should have been, but including all grasslands is going to include a lot that is grazed by livestock. To some extent and in some regions, livestock has simply replaced wild grazers, but that's not universally true qualitatively or quantitatively.

The authors should mention how including land grazed by livestock might affect their estimates of "potential" biomass. What kind of bias does this result in, and where might it be strongest? Maps of grazed area and grazer density can illuminate the latter.

Consider filtering the 0.5° grid cells in the analysis based on some threshold of pasture fraction, which can be obtained for example from the LUH2 land use data. That's actually at 0.25° resolution, so perhaps this filtering could happen at an intermediate step before aggregation to 0.5°, potentially enabling more 0.5° cells to be included.

Response: Thanks for your constructive suggestion. Like urban and cropland, pasture disturbs calculations of vegetation biomass, because livestock eats up part of the grass. Typically, the expansion of pasture area decreases the actual vegetation biomass, NPP and potential vegetation biomass. According to the spatial distribution of managed pasture from Land-Use Harmonization 2 (LUH2), we found managed pasture gathers in eastern Asia, western Europe, south central Africa, and western South America. We filtered the 0.5° grid with pasture fraction greater than 0.01 to reduce the disturbance of managed pasture on evaluation accuracy of simulated potential NPP.

Follow your suggestion, we remap the distribution of vegetation without anthropogenic disturbance using MCD12C1 and LUH2. Grassland affected by grazing were removed. We added more explanation and figure in the revision as below:

"Part of grids covered by grassland were grazed by livestock, leading to the decrease of NPP of grass PFTs. We obtained land-use forcing data from Land-Use Harmonization (LUH2) to map the distribution of managed pasture data from 2001 to 2015 (Hurtt et al., 2020). As shown in Figure A4, grassland in eastern Asia, western Europe, south central Africa, and western South America were severely affected by

grazing. To exhibit the disturbance of managed pasture, we calculated the mean fraction of managed pasture within the corresponding 0.5° grid unit. When the fraction of managed pasture over 0.01, the grid covered by grassland was considered to be affected by managed pasture. We filtered grassland affected by pasture to map the distribution of natural vegetation zones without anthropogenic disturbance (Figure A5)." (see Revision, Page 14, Lines 334-342)

[Figure]

Figure A4. Spatial distribution of multi-year average fraction of managed pasture from 2001-2015 at 0.5 × 0.5 arc-degree resolution.

[Figure]

Figure A5. Map of land vegetation without anthropogenic disturbance from MCD12C1 and LUH2. END: Evergreen needleleaf forest, EBF: Evergreen broadleaf forest, DNF: Deciduous needleleaf forest, DBF: Deciduous broadleaf forest, MF:

Mixed forest, CS: Closed shrublands, OS: Open shrublands, WS: Woody savannas, SA: Savannas, GL: Grasslands, NNG: No natural vegetation, which means the zone is not covered by vegetation without anthropogenic disturbance.

(see Revision, Page 33)

9.  L 304: This says MODIS NPP comparison period starts 2000, but Fig. 3 caption says 2001.

Response: Thanks for your detailed comments. The time series of MOD17A3 is 2000-2015, while the time series of MCD12C1 is 2001-2015. In the evaluation of potential NPP, we used MCD12C1 to distinguish regions covered by natural vegetation. Therefore, the calculation of correlation coefficient began in 2001. We added more detailed explanations of NPP comparison period in the section 3.1 in order to help readers understand better.

"Based on land cover types dataset from 2001 to 2015, we obtained NPP-MOD17A3 data in natural vegetation zones without anthropogenic disturbance at the same period. Figure 4 shows that the modelled NPP from the SEIB-DGVM exhibited a high degree of consistency with the NPP-MOD17A3 data in natural vegetation zones over the period ($R^2$=0.63, p<0.05)." (see Revision, Page 15, Lines 349-352)

[Figure]

NPP correlation coefficient of MODIS and SEIB-DGVM

-0.8  -0.6  -0.4  -0.2  -0.1   0   0.1  0.2  0.4  0.6  0.8

Figure 3. Spatial patterns in the potential NPP correlation coefficients between SEIB-DGVM and MODIS between 2001–2015 (P<0.05). These data were used to validate SEIB-DGVM.

10. L 308–310: Simplify (this can be one sentence), and clarify why you're saying this (because MODIS data include used land).

Response: Thanks for your detailed comments. We added more clarify illustration of the definition about potential NPP as below:

"The potential NPP is defined as that assimilated carbon stored in natural vegetation without the disturbance of anthropogenic activities (Erb et al., 2018)." (see Revision, Page 13, Line 324-326)

11. In Fig. 4 and related text, it's not specified over what time period the SEIB output was averaged. This is critical to understanding how it compares to previous findings. SEIB value should not cover more than, say, 30 years; then you should exclude literature values outside that range. Or consider instead a scatter plot, with each literature value vs. SEIB at the time the literature value refers to.

Response: Thanks for your constructive suggestion. Following your suggestion, we remap the Figure 5 (Figure 4 in original manuscript) as below (see Revision, Page 16):

[Figure]

Figure 5. **Estimates of the potential vegetation biomass carbon stock from the literature (blue plot), state-of-the-art datasets (red plot) and this study (black line)**. Datasets are from the following studies: (1)(Erb et al., 2018; Erb et al., 2007), (2)(Bazilevich et al., 1971), (3)(Saugier et al., 2001), (4)(Erb et al., 2018; Bartholome and Belward, 2005), (5)(Olson et al., 1983), (6)(Erb et al., 2018; Pan et al., 2011), (7)(Ajtay et al., 1979), (8)(Erb et al., 2018; Ruesch and Gibbs, 2008), (9)(Kaplan et al., 2011), (10)(Shevliakova et al., 2009), (11)(Kaplan et al., 2011), (12)(Pan et al., 2013), (13)(Prentice et al., 2011), (14)(Erb et al., 2018; Erb et al., 2007), (15)(Erb et al., 2018; West et al., 2010), (16)(Hurtt et al., 2011).

12. It should be clarified somewhere what is meant by "potential" carbon stocks. Presumably this means "in the absence of human land use."

Response: Thanks for your constructive comments. We added the definition of "potential carbon stock" as below

"Throughout this study, the potential biomass carbon stock, biomass carbon stored in vegetation without anthropogenic disturbance, is recognized as a proxy for the potential of carbon storage by natural vegetation." (see Revision, Page 4, Lines 104-106)

13. L 153–155: Is tree growth daily or monthly? If both, I guess those are different growth processes? Please clarify.

Response: Thanks for your detail comments. The growth process of tree consists of three procedures with daily, monthly, and annual time step. We added more detailed illustration of the growth processes in order to help readers understand better.

"SEIB-DGVM utilizes three computational time steps: (1) During the growth phase, the metabolic procedures including photosynthesis, respiration, and carbon allocation are executed for each individual tree every simulation day. (2) The monthly process of tree growth including reproduction, trunk growth, and expansion of a cross-sectional area of the crown are executed. (3) On the last day of each year, the height of the lowest branch increases as a result of purging crown disks, or self pruning of branches, at the bottom of the crown layer." (see Revision, Page 6, Lines 160-165)

14. L 169–176: This paragraph purports to outline advantages of SEIB-DGVM compared to other models, but I think it should just be deleted. L 169–171: I interpret this to mean that SEIB-DGVM includes size-mediated competition for light, but it's not the only DGVM that does this. See, for example, LPJ-GUESS (Smith et al., 2001, Glob. Ecol. & Biogeog.) and LM3- PPA/LM4 (Weng et al., 2015, Biogeosciences). See also Fisher et al. (2018, Glob. Chg. Biol.) for a review of such "vegetation demographic models."

Response: Thanks for your constructive comments. Following your suggestion, we removed this paragraph.

15.  L 171–172 made me think that the simulations would start with PFT composition and structure derived from observations, but later it seems that this is not the case ("SEIB-DGVM simulations begin with seeds of selected plant function types planted in bare ground. The plant functional types are favored for establishment by the environmental conditions in each grid cell."). Please edit this sentence to clarify that such inputs can be used.

Response: Thanks for your constructive comments. In the model, new individual PFTs establish on the last day of each simulation year. The establishment of PFTs is determined by the climatic conditions. We modified this sentence as below:

"SEIB-DGVM simulations begin with seeds of selected PFTs planted in bare ground. The establishment of PFTs seeds are determined by the climatic conditions in each grid cell." (see Revision, Page 10, Lines 267-268)

16.  L 172–174: Unclear what this is trying to say. Is it that SEIB-DGVM can't do land use? If so, that's not an advantage—in models that have land use, it can be disabled for potential vegetation runs if desired.

Response: Thanks for your constructive comments. SEIB-DGVM is able to simulate the land cover types, while can't simulate the land use types. Following your suggestion, we removed Lines 172-174 in original manuscript.

17.  Sect. 2.3 (description of relevant processes in SEIB-DGVM): It should be clarified what of this is new to SEIB-DGVM in this paper vs. what was already there.

Response: Thanks for your constructive comments. In section 2.3, we added the more detailed illustration about the improvement of SEIB-DGVM V3.2 as below:

 "Based on the updated observation data, the allocation schemes of Boreal Needle-leaved summer-green trees and Tropical Broad-leaved evergreen trees were improved at SEIB-DGVM V3.02. Allocation schemes of other PFTs are the same as the original version." (see Revision, Page 7, Lines 183-185)

18. L 226: It's a bit surprising that plant demand doesn't actually enter into the calculation of water limitation status. The assumption seems to be that plants are always stressed, to some extent unless, the soil is fully saturated. I guess this is more of a comment about the Discussion: The authors should discuss the implications of this. It would seem to contribute to a bias of SEIB-DGVM towards greater fine root allocation.

Response: Thanks for your constructive comments. The model equation (Original, Page 8, Line 226) explains the constraining impact of water limitations on photosynthesis of tree and grass. Then, assimilated carbon is allocated to, and stored in, light- and water-gathering biomass carbon stocks among woody PFTs and grass PFTs. So, this model procedure does not induce a bias of carbon stock towards greater fine root allocation. We added more model equations about the water stress in order to help reader understand better as below:

"To control plant phenology and the rate of photosynthesis as a function of the limitation in terrestrial water, the physiological status of the limitation of terrestrial water is calculated as:

$$p_{sat} = PMAX ce_{tmp} ce_{co_2} ce_{water} \tag{6}$$

$$ce_{water} = \sqrt{stat_{water}} \tag{7}$$

$$stat_{water} = \frac{max(pool_{w(1)}/Depth_{(1)}, \ pool_{w(2)}/Depth_{(2)}) - W_{wilt}}{W_{fi} - W_{wilt}} \tag{8}$$

where $p_{sat}$ is the single-leaf photosynthetic rate of tree PFTs and grass PFTs ($\mu$ mol $CO_2$ m$^{-2}$ s$^{-1}$); $PMAX$ is the potential maximum of photosynthetic rate ($\mu$ mol mol$^{-1}$ $CO_2$ m$^{-2}$ s$^{-1}$); $ce_{tmp}$ and $ce_{co_2}$ are the temperature and $CO_2$ concentration effect coefficient (dimensionless), separately; $ce_{water}$ is the water effect coefficient (dimensionless); $stat_{water}$ is the physiological status of the terrestrial water limitation, which ranges between 0.0 – 1.0, dimensionless; $pool_{w(n)}$ is the water content in soil layer n, mm; $Depth_{(n)}$ is the depth of the soil layer n, mm; $W_{wilt}$ is soil moisture at the wilting point, m m$^{-1}$; and $W_{fi}$ is soil moisture at field capacity, m m$^{-1}$. When the temperature of all soil layers is less than 0° C, $stat_{water}$ is equal to 0." (see Revision, Pages 8-9, Lines 229-241)

19. L 277–279: It's very unclear what this test of "detection trends" actually is.

Response: Thanks for your detailed comments. We used Mann-Kendall and Sen's slope estimator statistical tests to determine whether there was a positive or negative trend in factorial simulation with statistical significance. The distribution of trends stimulated by each driver were shown at Figures A2, 3 in Revision.

"As shown in Figures A2, 3, detection trends of LVBC and WVBC for all driving factors performed statistically well (in agreement at the 95% confidence intervals), indicating this analytical method was suitable for trend attribution at the global scale."
(see Revision, Page 11, Lines 294-296)

[Figure]

**Figure A2. Potential LVBC trend maps during the period of 1916 to 2015 under different factorial simulations.** (a) $CO_2$ driving factorial simulation; (b) $CO_2$+precipitation driving factorial simulation. (c) $CO_2$+temperature driving factorial simulation; and (d) $CO_2$+radiation driving factorial simulation. Positive values indicate increasing trends in the ratio and vice versa. All results from Mann-Kendall and Sen's slope statistical tests correspond to the 95% confidence interval.

(see Revision, Page 31)

[Figure]

**Figure A3. Potential WVBC variation trend maps during the period of 1916 to 2015 under different factorial simulations.** (a) $CO_2$ driving factorial simulation; (b) $CO_2$+precipitation driving factorial simulation. (c) $CO_2$+temperature driving factorial simulation; and (d) $CO_2$+radiation driving factorial simulation. Positive values indicate increasing trends in the ratio and vice versa. All results from Mann-Kendall and Sen's slope statistical tests correspond to the 95% confidence interval.

20. L 518–524: Unclear. Did you not include N deposition at all? Wouldn't that mean that you've underestimated $CO_2$ fertilization? Please elaborate the N deposition methods (or lack thereof) in Methods and clarify this text.

Response: Thanks for your detailed comment. Yes, SEIB-DGVM does not include N deposition. Nitrogen is a limiting factor for vegetation growth. Consistent with $CO_2$ fertilization, N-deposition increases the land carbon sink and vegetation carbon storage. This study focused on the contributions of changes in $CO_2$ and climate to the long-term trend of carbon stocks, and did not evaluate the effect of N-deposition on carbon stock. Therefore, our study overestimated the contributions of driving factors, especially at $CO_2$ fertilization. We added explanation in Revision as below:

"For a wide variety of plant organs, the maintenance respiration rate is linearly related to the nitrogen content of living tissue. The relative proportions of nitrogen in each

organ for any PFT are linearly correlated. N-deposition doesn't include in SEIB-DGVM." (see Revision, Pages 6-7, Lines 173-176)

21. Results interpretation issues

L 338: Slower? Slower than what? $CO_2$ change looks faster than biomass C change.

Response: Thanks for your detailed comments. Yes, the increase of atmospheric $CO_2$ concentration is faster than that of global potential carbon stock. We changed "slower" to "dramatic". (see Revision, Page 16, Line 366)

22. L 415: Reference to Figs. 8–9 here is inappropriate, as those maps don't show attribution.

Response: Thanks for your detailed comments. We changed the "Figures A2, A3" (Figures 8, 9 in original manuscript) to "Figure 8" (Figure 7 in original manuscript). (see Revision, Page 21, Line 449)

23. L 419–426: Short-term variation is a completely different thing from long-term trend; it's unclear why they're being lumped together here (where it says the bit about temporal compensation).

Response: Thanks for your constructive comments. At local scale, radiation dominated the long-term trend of LVBC in 20.67% of global regions and that of WVBC in 13.74%. In contrast, radiation induced -3.19% variation of LVBC and -5.62% variation of WVBC at global scale. We suggest that this apparent paradox can be explained by spatially compensatory effects (Jung et al. 2017. Nature, 541, 516-520). We added more explanation in order to help reader understand better.

"Previous studies have pointed out that the variation of the terrestrial carbon stock caused by releasing or sequestering carbon is sensitive to anomalous changes in water availability and light use efficiency (Madani et al., 2020; Humphrey et al., 2018). At local scale, radiation dominated the long-term trend of LVBC in 20.67% of global zones and that of WVBC in 13.74%, while precipitation dominated the long-term trend of LVBC in 21.88% of global zones and that of WVBC in 17.09% of global zones.

However, radiation induced light variation in LVBC (-3.19%) and WVBC (-5.62%) at global scale. Precipitation explain 8.51% of LVBC trend and -2.76% of WVBC trend at global scale. LVBC and WVBC variations driven by precipitation and radiation were ultimately offset by spatially compensatory effects, which dampened the response of the carbon stock to these factors at global scale (Jung et al. 2017)." (see Revision, Page 27, Lines 534-543)

24. L 427–429: If there's no long-term trend in precipitation and radiation (as asserted at L 422–423), how can they induce a long-term change?

Response:Thanks for your constructive comments. Compared with $CO_2$ concentration and temperature, precipitation and radiation didn't show a dramatic and consistent trend based on multidecade observational. So, radiation and precipitation explain a small part of the variation in carbon stock, which is induced by the unobvious long-term changes in precipitation and radiation at global scale.

25. Precipitation effect appears to not be compensatory for "above-ground" biomass, which is most biomass! (Again at L 489–490.)

Response: Thanks for your detailed comment. Jung et al. (2017, Nature, 541, 516-520) pointed out that the contribution of water availability to global carbon sink was offset by compensatory effects with the increase of spatial grid-call resolution, which is consistent with our result that the light contribution of precipitation to carbon stocks at global scale.

26. Figs. 10–11 and related text: Regression tests for trends in mean and standard deviation across AI bins would be useful. I'd suggest trying a linear fit for Fig. 10 and a quadratic fit for Fig. 11.

Response: Thanks for your detailed comment. Following your suggestions, we added regression tests and plotted fitted curves in Figures 9 and 10 (Figures 10, 11 in original manuscipt) as below: (see Revision, Page 22 and Page 24)

[Figure]

**Figure 9. Relationships in the incremental change between AI and LVBC over the hydrological zones.** Magnitude of change in LVBC in the historical scenario S1 (a), $CO_2$ in scenario S2 (b), $CO_2$ + precipitation in scenario S3 (c), $CO_2$ + temperature in scenario S4 (d), and $CO_2$ + radiation in scenario S5 (e). Range of the box is 25%-75% of values; range of the whiskers is 10%-90% of values; the small red square is average value; the red line is the median line; and the black line is the fitted curve. Positive value of the Y axis represents the magnitude of increased LVBC from 1916 to 2015 under water-limitations conditions, and vice verse.

[Figure]

**Figure 10. Relationships in the incremental change in AI and WVBC over the hydrological regions.** Modelled WVBC enhanced magnitude in the historical scenario S1 (a), $CO_2$ in scenario S2 (b), $CO_2$ + precipitation in scenario S3 (c), $CO_2$ + temperature in scenario S4 (d), and $CO_2$ + radiation in scenario S5 (e). Range of the box is 25%-75% of values; range of the whiskers is 10%-90% of values; the small red square is average value; the red line is the median line, and the black line is the fitted curve. Positive value of the Y axis represents the magnitude of increased WVBC from 1916 to 2015 under water-limitations conditions, and vice verse.

27. L 438–439: It seems to me like an increasing trend in this difference would indicate that more water-limited areas experience more enhanced C growth, not less. ⋯ Although the trends are very small! This sentence is very confusingly written. "Fluctuations" I think might be the reason. This connotes year-to-year changes rather than a trend, which is what the figures are actually looking at.

Response: Thanks for your constructive comments. With mitigating water stress, plants can gather more water for photosynthesis and productivity. Figures 9 and 10 shown a

consistent phenomenon that the increase of carbon stock in zones without water stress is greater than that in zones with water stress. Decreased AI (increased water stress) constrained the carbon stock increase stimulated by drivers. The capacity of carbon storage is limited in arid regions.

Yes, we focused on the response of carbon stock density to changes in drivers within different water limitations conditions. So, the long-term trends were slight.

We added more detailed explanation as below:

"These results reveal that the carbon stock increases stimulated by changes in climate and $CO_2$ are constrained by water available. With increased warming, water limitations are expected to increasingly limit the carbon stock increase, specially at arid regions." (see Revision, Page 23, Lines 468-470)

28. Discuss: Why is there a (slight) increasing trend for AVBC but a (slight) unimodal pattern for BVBC?

Response: Thanks for your constructive suggestion. We added more explanation in section 4 as below:

"Drought mitigation promotes the growth of WVBC, while humid region with high light competition limits root growth. The result is consistent with previous finding that plants reduce allocation to roots in dense forests where aboveground competition for light is high (Ma et al. 2021)." (see Revision, Pages 27-28, Lines 557-560)

29. Use of "integrated" and "integral" throughout is confusing. Do you mean "total," as in AVBC+BVBC?

Response: Thanks, we changed "integrated" and "integral" to "total".

30. "carbon-stocks" should be "carbon stocks" throughout (no hyphen)

Response: Thanks, we changed "carbon-stocks" to "carbon stocks".

31. "Regions" and "regional" to me imply land masses or geopolitical boundaries, but it is often used here to describe latitude bands. It would be better to use "latitude bands/zones" and "zonal" instead.

Response: Thanks, we changed "regions" and "regional" to "latitude bands/zones" and "zonal".

32. L 49–51: It's unclear what the difference is between "direct" and "indirect" effects. This idea of "two mechanisms" is not ever returned to, so I suggest just simplifying this sentence to remove the distinction.

Response: Thanks for your detailed suggestion. We simplified this sentence as below: "The atmospheric $CO_2$ concentration are affected by the vegetation carbon stock, while the long-term trend of vegetation carbon storage capacity is also affected by the changes in climate and $CO_2$." (see Revision, Page 2, Lines 50-51)

33. L 73–75: Abrupt transition to talking about models was confusing.

Response: Thanks, we removed this sentence for maintaining coherence.

34. L 76: "negative response to climate" is vague.

Response: Thanks for your detailed comment. We added more detailed explanation as below:

"Water availability controls both carbon allocation and storage and can potentially transform zones characterized by a positive response to changes in climate and $CO_2$ to zones exhibiting a negative response. For example, global warming stimulates plant productively, Madani et al. (2020) found that there is a dramatically downward trend in the tropical productivity. With increased warming, water limitations are predictable to increasingly reduce the proportion of leaves' biomass, and decrease plant photosynthesis (Ma et al., 2021)." (see Revision, Page 3, Lines 80-85)

35. L 80–82: This sentence is vague (what is "oversensitivity"?) and seemingly unsupported.

Response: Thanks for your detailed comment. We added more detailed explanation as below:

"Water limitations have a strong regulating effect on the spatial pattern of change in vegetation carbon storage, demonstrating the effects of the changes in climate and $CO_2$ on the dynamics of the plant organs are affected by the terrestrial water gradient." (see Revision, Page 3, Lines 85-88)

36. L 182: What is "stock" biomass? This is missing from Fig. A1 and its caption.

Response: Thanks for your detailed comments. We added the explanation of stock as below:

"Stock biomass is used for foliation after dormant phase and after fires in PFTs, which is reserve resource in each individual tree." (see Revision, Page 9, Lines 253-254)

37. L 215: How frequently? Annually?

Response: Thanks for your detailed comments. This procedure is executed every simulation day.

"This 10% NSC is used for every daily process of reproduction, including having flowers, pollen, nectar, fruits, and seeds." (see Revision, Page 8, Lines 221-222)

38. L 238: "adjusted"? What is the usual method?

Response: Thanks. The usual root-shoot ratio (R/S) is the ratio of below-ground biomass (root biomass) and above-ground biomass (shoot biomass), and we added explanation as below:

"The root-shoot ratio (R/S) has been used to distinguish and investigate the ratio of below-ground biomass (root biomass) and above-ground biomass (shoot biomass) (Zhang et al., 2016)." (see Revision, Page 9, Lines 246-248)

39. Table 1: Please replace the heading "$CO_2$ fertilization" with "$CO_2$ concentration" for consistency (referring to environmental conditions rather than plant processes).

Response: Thanks. We changed "$CO_2$ fertilization" to "$CO_2$ concentration". (see Revision, Page 10, Table 1)

40. L 282: "We defined"… not really, right? Isn't this the same as used in Chen et al. (2019)?

Response: Thanks for your detailed comment. The aridity index is consistent with Chen et al. (2019). We are very sorry about making mistake here, and have re-written this sentence.

"We used aridity index (AI) to distinguish between the global hydrological regions for comparing the long-term trend in carbon stocks over different hydrological environments, and for quantifying the influences of each hydrological environment on the variations in the trends." (see Revision, Page 12, Lines 299-301)

41. Fig. A2: What is "no value"? Please increase weight of font in legend.

Response: Thanks for your detailed comment. Following your suggestions, we increased weight of font in legend and added the explanation of "No natural vegetation" ("no value" in original manuscript) as below:

"NNG: No natural vegetation, which means the zone is not covered by vegetation without anthropogenic disturbance" (see Revision, Page 33, Figure A5)

42. L 332–333: Is this just repeated from Methods? If so, delete. If not, elaborate (and move to Methods).

Response: Thanks. We removed this sentence.

43. L 337–338: How was 2.44 calculated? Mean annual range? Be more specific.

Response: Thanks for your detailed comment. We added more detailed explanation in order to help readers understand better.

"($\pm$ 2.44 represents intra-annual fluctuation in carbon stock, which is the difference between maximum value and a minimum value of carbon stock within the year" (see Revision, Page 16, Lines 363-365)

44. L 340–341: Specify R-squared and p-values for AVBC and BVBC as well.

Response: Thanks for your constructive suggestion. We added more detailed explanation about calculation of determined coefficient.

"Based on Pearson correlation analysis, this increasing trend of annual average carbon stock exhibits a robust agreement with the dramatic increase in atmospheric $CO_2$ concentration ($R^2$=0.9677, p<0.001), suggesting that the carbon stock is strongly affected by $CO_2$ fertilization. Meanwhile, the positive correlation between the carbon stock and $CO_2$ generally extends across LVBC ($R^2$=0.9669) and WVBC ($R^2$=0.9622)." (see Revision, Page 16, Lines 365-369)

45. Fig. 5 a:

Is the inset plot just the pink line? Clarify this in the caption.

Please use a color other than pink, as it's hard to tell from the red.

Please change "Dynamic of biomass carbon" to match the clearer label on the inset plot ("Biomass carbon")

Please add units to Y-axes in inset plot.

Consider just removing the inset plot. It doesn't really add much except potential for confusion. This would also allow zooming in on the biomass Y-axes to provide better visibility.

Caption: "during the first decade; the averaged value (1916–1925, red line)

and the last decade averaged value"

Response: Thanks for your detailed comment. Following your suggestion, we removed the inset plot form Figure 6(a) (Figure 5 in original manuscript), and deleted "averaged value" in the caption as below (see Revision, Page 17):

[Figure]

**Figure 6. Global potential biomass carbon stocks of vegetation during the past 100 years.** (**a**) The evolution of global potential biomass stocks (LVBC+WVBC), along with changes in biomass stocks that can be attributed to the variability and trend of LVBC and WVBC through the twentieth century. The red line represents the monthly value of LVBC, the blue line represents the monthly value of WVBC, and the black line represents the annual value of $CO_2$ concentration. (**b, c**) Zonal averaged sums of the annual LVBC and WVBC for latitudinal bands during the first decade (1916–1925, red line) and the last decade (2006–2015, blue line) shows the increased carbon stock capacity.

46. L 365: "further supports." Further? Where was this mentioned before? Should be mentioned in Sect. 3.2, where it becomes obvious that AVBC will dominate because it's so much higher.

Response: Thanks for your detailed comment. In section 3.2, we added more detailed illustration about the contribution of the change in LVBC to the change in total carbon stock as below:

"we see that LVBC increases $116.18 \pm 2.34$ Pg C (or ~15.60%), which explains 97.42% of total carbon stock increasing trend and dominates the positive global carbon stock trend;" (see Revision, Pages 16-17, Lines 370-372)

47. L 366–367: "the proportion of the total change in carbon-stocks is small (3.08 ± 0.14 Pg C)"—what does this mean? "Proportion" makes me think you're talking about a fraction, but the units are PgC.

Response: Thanks for your detailed comment. We are very sorry about making mistake here, and have re-written this sentence.

"Although the proportion of the total change in carbon stocks is small (2.58% of total carbon stock increase), about 61.00% of the land surface shows an increase in WVBC; of these terrestrial grids, 55.81% was characterized by a significant p=0.01 increase." (see Revision, Page 18, Lines 399-401)

48. Fig. 6: Colors on map are fine, but cells should be gray (or otherwise distinguished) if p-value is not significant. And then the inset bar graphs should not have colors, because that's confusing with the map colors; distinguish increasing vs. decreasing trends instead with text. Note different color scale for (c) in caption. Increase resolution so that pixels aren't blurred.

Response: Thanks you for detail comments. Following your suggestion, the color of the grid that was not statistically significant was changed to gray. The color of inset bar graphs was removed. We added the symbols of (+) and (-) to indicate the increasing and decreasing trend, separately. (see Revision, Pages 18-19)

[Figure]

**Figure 7. Spatial patterns in the trends of potential vegetation carbon stocks and their fractions from 1916 to 2015.** Difference induced by changes in climate and $CO_2$ in terrestrial biomass carbon stock (a), LVBC (b), and WVBC (c) during the

historic period 1916–2015. The blue bar indicates the significantly increasing trends and the red bar indicates the significantly decreasing trends in carbon stocks. (d) Trend in the LVBC/WVBC ratio from 1916 to 2015. The blue bar indicates significantly increasing trends in the ratio, and vice versa. The grey bar indicates the trend is statistically insignificant (P >0.05). The sub-graphs show the significant test results. A '+' symbol indicates a positive trend, and vice versa.

49. L 386–393: What of this is coming from Fig. 7a/c? It's not referred to anywhere in the text.

Response: Thanks for your detailed comment. We added the explanation of Figure 8a, 8c (Figure 7a,7c in original manuscript) as below:

"$CO_2$ fertilization explains the largest proportion of the change in the carbon stock; about 82.45% change in LVBC was positive (Figure 8a), whereas 89.28% of the change in WVBC was positive (Figure 8c)." (see Revision, Page 19, Lines 414-416)

"Figure 8a illustrates that temperature is the largest climatic contributor to the change in LVBC (13.83%, 2.572 g m$^{-2}$ yr$^{-1}$), followed by precipitation (8.51%, 1.572 g m$^{-2}$ yr$^{-1}$) and radiation (–3.19%, –0.649 g m$^{-2}$ yr$^{-1}$)." (see Revision, Page 21, Lines 432-434)

"Figure 8c shows there is a difference in the negative contribution of precipitation to the change in WVBC at the global level (–2.76%, –0.013 g m$^{-2}$ yr$^{-1}$). Temperature is the largest climatic contributor to the change in WVBC (15.36%, 0.075 g m$^{-2}$ yr$^{-1}$), followed by radiation (-5.63%, -0.027 g m$^{-2}$ yr$^{-1}$)." (see Revision, Page 21, Lines 437-439)

50. Fig. 7: Include labels on figure for above- vs below-ground.

Response: Thanks. Following your suggestion, we added labels on Figure 8 (Figure 7 in original manuscript, see Revision, Page 20).

[Figure]

**Figure 8. The proportion of change in the vegetation biomass carbon stocks attributed to driving factors**. Ratios of the driving factors of $CO_2$ fertilization effects ($CO_2$), climate change effects (CLI), precipitation (Pre), temperature (Tem), radiation (Rad) for LVBC (a) and WVBC (c) under the five scenarios using the Mann-Kendall and Sen's slope estimator statistical tests. Attribution of LVBC (b) and WVBC (d) dynamics to driving factors calculated as averages along 15° latitude bands. At local scales, the driving factors include $CO_2$, Pre, Tem, Rad, and other climate factors (OF). A '+' symbol indicates a positive effect of the driving factor on carbon stock, and vice versa. The fraction of global area (%) that is predominantly influenced by the driving factors is shown at the top of the bar.

51. L 419: Start a new paragraph here at "Previous", to provide some separation between pure results and discussion.

Response: Thanks your constructive suggestion. We moved Lines 419-429 in original manuscript to section 4 as below:

"Previous studies have pointed out that the variation of the terrestrial carbon stock caused by releasing or sequestering carbon is sensitive to anomalous changes in water availability and light use efficiency (Madani et al., 2020; Humphrey et al., 2018). At local scale, radiation dominated the long-term trend of LVBC in 20.67% of global zones and that of WVBC in 13.74%, while precipitation dominated the long-term trend of

LVBC in 21.88% of global zones and that of WVBC in 17.09% of global zones. However, radiation induced light variation in LVBC (-3.19%) and WVBC (-5.62%) at global scale. Precipitation explain 8.51% of LVBC trend and -2.76% of WVBC trend at global scale. LVBC and WVBC variations driven by precipitation and radiation were ultimately offset by spatially compensatory effects, which dampened the response of the carbon stock to these factors at global scale (Jung et al. 2017)." (see Revision, Page 27, Lines 534-543)

52. Figs. 8–9: Gray out pixels without a significant trend. Increase resolution so that pixels aren't blurred. I don't think these are actually discussed anywhere except L 415, which I think is inappropriate because they don't actually deal with this aspect of attribution. Add some discussion of them in the Results. Might be more useful to replace these with mapped versions of Fig. 7, showing the fraction of the trend contributed by each factor in each grid cell.

Response: Thanks for your detailed comments. Figures 8 and 9 in original manuscript were put in the Supplement as Figures A2 and A3. Following your suggestion, we used gray to indicate grids without statistical significance in Figures A2 and A3. Yes, revised Figure 8 (Figure 7 in original manuscript) is more appropriate to explain the contribution of drivers to the trend of carbon stocks. We added more explanation of Figures A2 and A3 as below:

"Modelled WVBC trends based on the factorial simulations have similar spatiotemporal patterns to LVBC (Figures A2 and A3), the spatial patterns of light- and water-gathering carbon stocks show a significant increasing trend in the most of boreal zones. In the Southern Hemisphere, the trends of WVBC are extensively statistically insignificant in all factorial simulations, and only a small proportion of grids show a significantly increasing trend. There is a significantly increasing trend in LVBC in south-central Africa and northern South America." (see Revision, Page 21, Lines 440-445)

[Figure]

**Figure A2. Potential LVBC trend maps during the period of 1916 to 2015 under different factorial simulations.** (a) $CO_2$ driving factorial simulation; (b) $CO_2$+precipitation driving factorial simulation. (c) $CO_2$+temperature driving factorial simulation; and (d) $CO_2$+radiation driving factorial simulation. Positive values indicate increasing trends in the ratio and vice versa. All results from Mann-Kendall and Sen's slope statistical tests correspond to the 95% confidence interval.

[Figure]

**Figure A3. Potential WVBC variation trend maps during the period of 1916 to 2015 under different factorial simulations.** (a) $CO_2$ driving factorial simulation; (b) $CO_2$+precipitation driving factorial simulation. (c) $CO_2$+temperature driving factorial simulation; and (d) $CO_2$+radiation driving factorial simulation. Positive values indicate increasing trends in the ratio and vice versa. All results from Mann-

Kendall and Sen's slope statistical tests correspond to the 95% confidence interval.

53. "Modelled AVBC enhanced magnitude" throughout is unclear.

Response: Thanks you for your detailed comment. We changed "Modelled AVBC enhanced magnitude" to "Magnitude of change in LVBC". (see Revision, Page 22, Figure 9)

54. Figs. 10–11: Specify what a negative vs. a positive value means on the Y axis.

Response: Thanks you for your detailed comment. We added explanation in the captions of Figure 9 and 10 as below:

"Positive value of the Y axis represents the magnitude of increased LVBC from 1916 to 2015 under water-limitations conditions, and vice verse." (see Revision, Page 22, Figure 9)

"Positive value of the Y axis represents the magnitude of increased WVBC from 1916 to 2015 under water-limitations conditions, and vice verse." (see Revision, Page 24, Figure 10)

55. L 432–435: Remove citations from this sentence. Add actual discussion (in a new paragraph after your results in this subsection) of how your results compare to the literature.

Response: Thanks for your suggestion. We removed citations and added more discussion in section 4 as below:

"Our findings are consistent with other reports about the impact of increasing water limitations on terrestrial ecosystem. Based on satellite remote sensing observations, Madani et al. (2020) found that changes in water constraints can lead to variable responses in ecosystem productivity and net carbon exchange. Humphrey et al. (2021) found that increasing water stress limits the response magnitude of carbon uptake rates through a down-regulation of stomatal conductance and suggested that land carbon uptake is driven by temperature and vapour pressure deficit effects that are controlled by terrestrial water availability. Ma et al. (2021) found that plants increase investment

into building roots in arid region because the extent of water limitation there is exacerbated by global warming. Terrestrial ecosystems utilize sensitive strategies to allocate and store biomass to adjust to local hydrological conditions." (see Revision, Page 28, Lines 566-575)

56. L 435–438: Combining these sentences would increase readability.

Response: Thanks for your suggestion. We combined these sentences in order to help readers understand better as below:

"As shown in Figures 9 and 10, with an increase in the aridity index (i.e., an increase in available water), the magnitude and range in variations of LVBC density and WVBC density gradually enhance. Based on the results of factorial simulations, we find a positive relationship between LVBC and water pressure. In extreme water stress, the increase of LVBC tends to zero and plants stop growing. There is no obvious different in the slopes of fitting curves between factorial simulations. The pattern of the enhanced magnitude and range of variation in the WVBC density is unimodal with water stress gradient in all factorial simulations. With the increasing of AI, the magnitude of change in WVBC increases at first and then decreases finally. The mitigation of water stress promotes WVBC increase, while excess surface water limits the response of WVBC to changes in climate and $CO_2$." (see Revision, Pages 22-23, Lines 459-468)

57. L 442–446: What does "drivers attributed to increase (A/B)VBC changed" mean?

Response: Thanks for your detailed comment. We added more detailed explanation as below:

"drivers attributed to increase AVBC density" (see Original, Page 21, Line 442) was changed to " increased LVBC density induced by drivers" (see Revision, Page 23, Lines 473-474)

"Drivers attributed to increase BVBC density" (see Original, Page 22, Line 444) was changed to "Increased WVBC density induced by drivers" (see Revision, Page 23, Lines 478)

58. L 457–465: This should be in the Introduction.

Response: Thanks for your detailed suggestion. We moved these sentences to the Introduction as below:

"In arid region, vegetation utilizes a tolerance strategy to allocate biomass, storing more biomass carbon in roots to resist enhanced water stress (Chen et al., 2013). Conforming to the optimal partitioning hypothesis, plants store more carbon in shoots and leaves in environments where water is more available and shift more carbon to roots when water is more limited (Yang et al., 2010; Mcconnaughay and Coleman, 1999)." (see Revision, Page 3, Lines 76-80)

59. L 470: What is "terrestrial water"? Where did you do this?

Response: Thanks for your detailed comment. We are very sorry about making mistake here, and removed this sentence as below.

"More importantly, we investigated the extent of the responses of carbon stocks to water limitations." (see Revision, Page 26, Lines 501-502)

60. L 502: What are "indirect factors"?

Response: Thanks for your detailed comment. We are very sorry about making mistake here, and have and have re-written this sentence.

"Moreover, we found that indirect effects of water limitation regulate increasing rate of each carbon pool." (see Revision, Page 28, Lines 560-561)

61. L 502–505: Second part of this sentence seems unrelated to the first.

Response: Thanks for your detailed comment. We are very sorry about making mistake here, and have and have re-written this sentence.

"Although vegetation carbon stocks dramatically increase under the effects of climate and $CO_2$ changes, the increasing rate of LVBC faster than WVBC in humid region. Vegetation stores more biomass in aboveground plant organs (trunk and foliage) to gather light." (see Revision, Page 28, Lines 561-563)

62. L 505: "lowers"? Relative to what?

Response: Thanks for your detailed comment. We are very sorry about making mistake here, and have and have re-written this sentence.

"Dryland vegetation decrease the LVBC/WVBC ratios and stores more biomass below ground to enhance the capture of water resources." (see Revision, Page 28, Lines 563-565)

63. L 516: "in factorial simulations"? What does that have to do with anything?

Thanks for your detailed comment. We explained this in a more detailed way by changing the sentence to the following:

"Trunk biomass contains tree branches and structural roots (coarse roots and tap roots) (Sato et al., 2007), so the R/S ratio of potential vegetation in factorial simulations is smaller than the R/S of actual vegetation in observation stations." (see Revision, Page 28, Lines 582-584)

64. L 517–518: Why is this a limitation?

Response: Thanks for your detailed comment. We added more detailed explanation of limitation as below:

"Root biomass only contains the fine root biomass, leading to an underestimate in belowground organ biomass of trees and grasses compare with previous conclusion (Ma et al., 2021; Yang et al., 2009)." (see Revision, Page 28, Lines 584-586)

65. L 199: Second and third commas should be semicolons.

Response: Thanks. Commas were changed to semicolons. (see Revision, Page 8, Lines 205-206)

66. L 200: Comma should be a semicolon.

Response: Thanks. Comma was changed to semicolons. (see Revision, Page 8, Line 207)

67. L 205: "are" should be "is"

Response: Thanks. "are" was changed to "is". (see Revision, Page 8, Line 211)

68. L 213: Tilde should be an en dash.

Response: Thanks. "Tilde" was changed to "en dash". (see Revision, Page 8, Line 219)

69. L 253: Should be "functional".

Response: Thanks. "plant function types" was changed to "PFTs". (see Revision, Page 10, Line 267)

70. L 265: "trend" is there twice.

Response: Thanks. We removed the second "trend".

71. L 327: "form" should be "from".

Response: Thanks. We changed "form" to "from". (see Revision, Page 15, Line 357)

72. L 348: Tropical.

Response: Thanks. We changed "tropic" to "tropical". (see Revision, Page 17, Line 376)

73. L 382: Comma should be a semicolon.

Response: Thanks. We changed comma to semicolon. (see Revision, Page 19, Line 415)

74. L 405: Specify Fig. 7a.

Response: Thanks. We changed "Figure 7" to "Figure 8a" (Figure 7a in original manuscript). (see Revision, Page 21, Line 432)

75. L 423: "variant" should be "variation".

Response: Thanks. We removed "variant".

76. L 453: Delete "that"; "spatial" should be "temporal".

Response: Thanks for your detailed suggestion. We are very sorry about making mistake here, and have re-written this sentence as below:

"Figure 11 illustrates temporal variations in the carbon stock ratio within and between hydrological regions." (see Revision, Page 24, Lines 487-488)

**Thanks again for your time and efforts put on this manuscript, which is acknowledged in the paper.**

---

## Referee Report (RR1)

**Review of revision: "Impact of changes in climate and CO$_2$ on the carbon-sequestration potential of vegetation under limited water availability using SEIB-DGVM version 3.02"**

**General comments**

General comments from my previous review:

> In this manuscript, the authors perform simulations with the dynamic global vegetation model SEIB-DGVM to explore the impact of historical changes in climate and atmospheric CO$_2$ concentration on potential carbon sequestration in live vegetation. Intriguingly, they look not just at total biomass, but also "aboveground" vs. "belowground" biomass (although those terms are misleading; see below). This allows the authors to examine how plants have shifted their growth strategies over the last century to maintain a competitive edge under environmental change.

> The results show that both biomass pools have increased, but with "belowground" increasing more than "aboveground" on a relative basis. Factorial experiments reveal that atmospheric CO$_2$ increase is unsurprisingly the dominant driver of potential biomass increase in most of the world, but temperature and other factors are more important at latitudes above 60°N. The results also show that "aboveground" and "belowground" responses to environmental change differ along an aridity gradient, as well as from each other.

> The authors designed a suite of experiments well-suited to explore how plant individuals and communities have changed their growth strategies to deal with environmental change. However, the manuscript needs substantial rework. Most importantly, while the Introduction briefly mentions previous findings regarding shifts in above- and belowground allocation under environmental change, this should build up to a set of hypotheses that are then tested with the model experiments. It is also unclear why this was submitted to *Geoscientific Model Development*. Perhaps if it were more focused on comparing SEIB-DGVM biomass to observations it would fit as an evaluation paper, but the work performed is much more high-level than that. I thus think it would be more appropriate to move to *Biogeosciences*.

The authors decided not to move journals, which is fine. They have made significant improvements in terms of explaining how water stress theoretically affects allocation, as well as tying their results back to this theory. However, I still have some significant questions about the methodology and confusion about the interpretation of results. As such, I again suggest this manuscript be ***reconsidered after major revisions***.

**Specific comments**
**Methods: Hydrological regions**

The aridity zones were determined based on a 115-year average, but it's possible that they could see long-term trends. For example, a gridcell classified as "semi-arid" on average might have been arid at the beginning of the simulation and sub-humid at the end. This is potentially a very important confounding factor, and might explain the sometimes-large variation around Δ=0 and the resulting weak trends in Figs. 9–10. If a lot of gridcells see shifts like this, it might be necessary to restrict analysis to grid cells that didn't change in terms of aridity, or didn't change much.

Another option might be to let gridcells shift their classification over time. This would require the classification to be based on a rolling mean of the previous, say, 15 or 30 years of climate. So, e.g., a bin in Figs. 9–10 would be "[mean 2015 L/WVBC value for all cells that were in this bin in 2015; i.e., whose 2001–2014 climate fit into this bin] minus [mean 1916 L/WVBC value for all cells that were in this bin in 1916; i.e., whose 1901–1914 climate fit into this bin]." Fig. 11 would show, at each year Y in each subfigure F, "mean Y L/W ratio for all cells that were in class F in Y; i.e., that qualify as class F based on climate in years Y–15 to Y–1." This would be necessary in order to avoid spurious interannual switches in classification, as well as to minimize the effects of a lag in vegetation community response to changing conditions.

**Methods: Pasture exclusion**

I appreciate the authors having made this change to their analyses, as I suggested. However, I'm a bit unclear as to how they actually did it. L338-42 are confusing, but I interpret them as saying: "To minimize confounding effects of livestock grazing, we excluded grassland grid cells with pasture fraction greater than 1%."

If my interpretation is correct, it's hypothetically possible for a "forest" gridcell that's 51% forest and 49% grassland, with LUH2 saying "49% of the area in this gridcell is pasture," and it wouldn't be excluded. This doesn't seem right. Instead, the authors should exclude *any* gridcell with a significant pasture fraction, whether it's classified as "grassland" or not. 1% might be too strict, though—maybe something like 10% instead.

Also, make it clear that this exclusion only applies for the NPP comparison.

**Unimodal pattern in Fig. 10**

In my initial review, I raised the question of why $\Delta$WVBC decreases along the aridity axis from semi-arid to humid regions. The authors have not actually addressed this, despite an attempt at L556-9. "Drought mitigation promotes the growth of WVBC"—okay, sure, that explains why it initially increases from left to right. "humid region with high light competition limits root growth"—but this figure is about WVBC, not LVBC. Why would high light competition in humid regions lead to decreased $\Delta WVBC$ relative to semi-arid regions? (My thinking is it's because vegetation in these regions is light-limited: They're not seeing any alleviation of that limitation with climate change.)

**Other issues with interpretation: Results**

Some of these suggestions might seem like they're better suited for a Discussion section than a Results section, but I think it would not be good writing to simply list a set of observations with no context, then explain them somewhere later in the paper. The current Discussion section is well structured in the sense that it provides a general overview of the results and then compares to previous literature.

- Figs. 9-10, A6-7

- o L461-2: It's not that plants stop growing; they're still alive! It's that they don't end up increasing their carbon stocks—i.e., it's so dry that they can't take advantage of higher $CO_2$.
- o L462-3: What does it *mean* that there's no obvious difference among the slopes?
- o L472-483: What do all these factorial results *mean*? It is not sufficient to just saying how big the difference is between the minimum and maximum experiment. What do they imply regarding the drivers of partitioning according to your hypotheses? You approach this for LVBC at L475-7, but you don't actually connect the results back to the hypotheses. (And you don't do this at all for WVBC.) Guide the reader!
  - ▪ L472-4: I'm not sure where these numbers come from.
  - ▪ L473: "changed" should be "ranged", I think. You're not comparing a *change*, you're comparing across a *range* of aridity classes. Unless I'm misunderstanding—as I said, I don't know where the numbers came from.
- Fig. 11 (L484-97)
  - o Again, guide the reader. Here you've done a good job of phrasing the results in a way that connects back to hypotheses ("Under the synergistic effect of drivers and water stress, … there is a larger proportion of biomass allocated to, and stored in, light-gathering vegetation organs."), but it's unclear how that is evident from the figure. Is it because the blue line is so much higher than the other lines? But why only in sub-humid and humid zones? And there, what does it mean that the ratio goes back down when additional factors are added after S3?
  - o L488: What does "variation range of ratio between LVBC and WVBC" mean? Interannual variation? Variation among factorial experiments? Neither seems to match the trend mentioned.

**Other issues with interpretation: Discussion**

- L503-29: LVBC and WVBC trends overall
  - o This is very confusing and disjointed. L514-7 and L527-9 make it seem like plants are tending to shift their allocation from WVBC to LVBC, but then L519-20 ("LVBC… dominates the long-term trends") and L525-7 seem to suggest the opposite. What's correct? Here's a hint—focus on the *ratio*. The absolute numbers I think are not very informative about allocation changes, because wood biomass is always so much higher than fine root biomass.
  - o What does "Compared with WVBC" mean here? Does it mean "116.18 ±2.34 Pg C" is ΔLVBC minus ΔWVBC?
  - o L520-2: Yes, but then why is increasing ΔWVBC concentrated in high latitudes? The difference here might be something that partitioning theory could explain, or maybe not—maybe it's just a relaxation of climate limitations in the high latitudes that low latitudes never experienced (as possibly suggested at L545-7).
- L531-47: Factorial experiments
  - o This needs to be cleaned up a lot. For one thing, just listing these numbers feels much more like something for the Results rather than the Discussion. Here, you should be focusing on the implications of your results for scientific understanding,

and comparing your results to previous literature. For another, the presentation of results is really confusing. For example, radiation doesn't "dominate" the trend at *any* latitude band. It *explains 20.67% of the global variation*, though. Explain where these numbers are coming from (Fig. 8b/d, adding the (–) and (+) numbers at the bottom of each). But then maybe I have it wrong—it doesn't make any sense for any of these "fraction of variation explained" numbers to be negative, as they are for, e.g., precip → WVBC (–2.76%). Do you mean instead that the net influence of precip on WVBC is negative?

- o L531-2: Is this conclusion drawn from Fig. 8? If so, I would rephrase to talk about the amount of variation explained, rather than the amount of increase. Always mention what figure(s) your assertions come from.
- o L535-540: Still need to do a better job of tying these results back to the hypotheses and/or to other explanations. The sentence at L539-40 might be directly relevant to optimal partitioning theory, depending on what it ends up saying after the authors clean up the section.

- L578-90: Caveats
  - o L587-90: This sentence could be interpreted as "We didn't vary N deposition over the experiment," when in reality no N deposition was included *at all.* This means that the simulated ecosystems would have an incorrectly *low* amount of N input, leading to incorrectly *high* amounts of N limitation, leading to an *underestimate* of CO2 fertilization (because if they're N-limited, they can't take advantage of higher CO2 levels). This is the opposite of what the authors seem to conclude.

**Technical corrections and minor suggestions**

- To avoid confusion (such as I exhibited in my first review), aridity index axes on all figures with them should include (something like) "drier" at 0 and (something like) "wetter" at 1. Or at least an explanation of this should be in the caption. The axes should also include labels indicating what the ranges are for the different classifications (humid, semi-arid, etc.).

Sect. 1: Introduction

- L50: "are" should be "is"
- L56-8: Citation?
- L76: "region" should be "regions"
- L82-83: The two parts of this sentence seem to contradict each other.
- L84: "predictable" should be "predicted"
- L104-6: It's not really a *proxy*; this is just how you define it. (Which is fine, of course!)

Sect. 2.1: Forcing data (minimal changes; no comments)

Sect. 2.2: "Overview of modeling concept in SEIB-DGVM"

- Good clarification of model timesteps
- L175-6: "doesn't include" should be "isn't included"

Sect. 2.3.1: Allocation in SEIB-DGVM

- L207: Define "soffit" (or, ideally, use a simpler word, like "layer").
- L220-1: Not a complete sentence. Suggest changing the period to a comma and deleting "This".
- L221: "NSC" not previously defined (should happen at L187).
- L221-2: Clarify that these organs are not explicitly simulated (unless they are!), and instead are represented as a flux to litter.

Sect. 2.3.2: Description of partitions

- L254: Delete "in PFTs", maybe? What does it mean?
- L257-8: GVBC should be LVBC.
- L259-63: "wood" should be "woody vegetation".
- L257-263: $W_{mass}$ should be renamed, e.g. to $T_{mass}$ (T for "tree" [or "trees and shrubs"] instead of W for "woody vegetation"), to avoid confusion with other use of W for "water-gathering".

Sect. 2.4.1: Run setup (minimal changes; no comments)

Sect. 2.4.2: Factorial simulation scheme

- L274-7: The "remarkable effects" comment about wind and relative humidity is a result; don't include it in Methods. Deleting that comment will allow wind and RH to replace "other factors" in the previous sentence.

Sect. 2.4.3: Non-parametric test methods

- L294: "A2, 3" should be "A2–3" or "A2 and A3"
- Figs. A2–3: Caption should include experiment labels S2 etc.

Sect. 2.4.4: Hydrological regions (no technical corrections, but see Specific Comments above)

Sect. 2.5: Observational data

- L328-9: "It was defined as vegetation grid that the land cover type of this grid is" should be "We defined vegetated grid cells as those whose largest component was". This grammatical correction turns out to also be a simplification, as it then allows the deletion of the sentences at L332-4.
- L331-2: Replace this with a simpler sentence along the lines of "Other grid cells were excluded from our analysis."
- L334: Start a new paragraph here, as you're talking about something new.
- Fig. A4: There should be a clear break in the color bar at whatever threshold you end up using (currently 1%). Also, red-green axes should be avoided, because red-green colorblindness is relatively common.
- Fig. A5:
  - "NNG (no natural vegetation)" should more accurately be something like "NI (not included)."
  - "END" should be "ENF"

Sect. 3.1:

- Fig. 4: Pixels that were excluded based on land cover should be colored gray, to distinguish from included pixels with low correlation.
- Fig. 5: This is much better than the previous bar graph version. My only suggestion is to delete "Dynamic of" from the Y-axis label.

Sect. 3.2

- Fig. 6:
  - (a): Delete "Dynamic of" from right Y-axis label
- Could also refer to Fig. 7 for extra support in this section.

Sect. 3.3:

- L385: "while **they** declined"
- L387: "decrease" should be "decreasing"

Sect. 3.5

- Figs. 9-10, A6-7
  - Figs. 9-10: Are Y-axis units *per year*? It would be easier to relate to other figures if they were total over the simulation.
  - L460: "enhance" should be "increase".
  - L460-1: That's not really from the factorial simulations; it's obvious from the historical simulation.
  - L461: "water pressure" should be "aridity".
  - L462: "different" should be "difference"
  - L476: Should be "matches", not "matchs"
- Fig. 11 (L484-97)
  - L490: "Positive influence" is unclear, and "allocate" should be "allocation"

Discussion

- L503-29: LVBC and WVBC trends overall
  - L520: For clarity, say "annual **change in** LVBC"
- L549-76: Aridity zones
  - Great improvements here with regard to comparison to other literature and tying back to theory.
  - L552: "region" should be "regions"
  - L550-2: Refer to figures supporting this (presumably Figs. 9 and 10).
  - L565: Start a new paragraph here, since you're switching from analyzing your results to comparing them with previous literature.
  - L567-8: Description of Madani et al. (2020) is too vague. "Variable"? In what way, and how does it compare to your results?
- L578-90: Caveats
  - L584: I would say "**apparent** underestimate," as the numbers from SEIB-DGVM aren't *wrong*—they're measuring something different.
- L592-601: Conclusion
  - L594: "to" should be "vs." or "and"

---

## Referee Report (RR2)

**Review of revision 2: "Impact of changes in climate and $CO_2$ on the carbon-sequestration potential of vegetation under limited water availability using SEIB-DGVM version 3.02"**

**General comments**

General comments from my original review:

> In this manuscript, the authors perform simulations with the dynamic global vegetation model SEIB-DGVM to explore the impact of historical changes in climate and atmospheric $CO_2$ concentration on potential carbon sequestration in live vegetation. Intriguingly, they look not just at total biomass, but also "aboveground" vs. "belowground" biomass (although those terms are misleading; see below). This allows the authors to examine how plants have shifted their growth strategies over the last century to maintain a competitive edge under environmental change.
>
> The results show that both biomass pools have increased, but with "belowground" increasing more than "aboveground" on a relative basis. Factorial experiments reveal that atmospheric $CO_2$ increase is unsurprisingly the dominant driver of potential biomass increase in most of the world, but temperature and other factors are more important at latitudes above 60°N. The results also show that "aboveground" and "belowground" responses to environmental change differ along an aridity gradient, as well as from each other.
>
> The authors designed a suite of experiments well-suited to explore how plant individuals and communities have changed their growth strategies to deal with environmental change. However, the manuscript needs substantial rework. Most importantly, while the Introduction briefly mentions previous findings regarding shifts in above- and belowground allocation under environmental change, this should build up to a set of hypotheses that are then tested with the model experiments. It is also unclear why this was submitted to *Geoscientific Model Development*. Perhaps if it were more focused on comparing SEIB-DGVM biomass to observations it would fit as an evaluation paper, but the work performed is much more high-level than that. I thus think it would be more appropriate to move to *Biogeosciences*.

And from my second review:

> The authors decided not to move journals, which is fine. They have made significant improvements in terms of explaining how water stress theoretically affects allocation, as well as tying their results back to this theory. However, I still have some significant questions about the methodology and confusion about the interpretation of results. As such, I again suggest this manuscript be reconsidered after major revisions.

In the latest (second) revision, the authors have done a good job of responding to my comments. They improved their methods significantly by (a) excluding grid cells that changed aridity classes and (b) refining their pasture-cell exclusion rule. Additionally, they have made great improvements in terms of explaining their analyses. As a result, I suggest this paper be ***published after minor revisions***.

**Specific comments**

*Pasture vs. rangeland*

You masked based on the LUH2 "managed pasture" layer, but most grazing land by area is actually *rangeland*—see below for 2010. Please consider the exclusion based on the total pasture+rangeland area. (I'm sorry, I should have caught this in the first revision!) Alternatively, there may be an argument that rangeland doesn't need to be excluded, as might be considered less intensely grazed. If you want to go that route, mention it in the text.

[Figure]

*Include temperature and "other factors" as "climate factors"*

At L557-64, the authors seem to consider only precipitation and radiation as "climate change" factors. The numbers from that result in a pretty good correspondence to the results from Zhu et al. (2016), but I would be surprised to learn that those authors included only precipitation and radiation in their analyses. Later in that paragraph, the authors talk about temperature, but it's unclear why it was not included before. Finally, "other factors" (wind speed and relative humidity) are not mentioned at all, but these are also climate factors. The authors should rewrite this paragraph to include all climate factors together in the initial analysis. (The final sentence is a good summary but should also mention "other factors.")

In almost all instances, the authors should replace "grid(s)" with "grid cell(s)." "Grid" is more appropriate when describing the overall setup (e.g. "grid resolution" is fine), but for referring to individual 0.5° boxes, "grid cell" is what should be used.

In "Minor suggestions and technical corrections," I've noted some places this should be fixed, but not all places.

**Minor suggestions and technical corrections**

- L82-4: This sentence is still confusing. "Global warming" seems to speak directly to temperature, but Keenan et al. (2017) found that slower temperature growth meant MORE C sequestration on land (due to lower ecosystem respiration). The Madani et al. (2020) bit is weird as well. Maybe "found that plants productively with water stress show a negative response to temperature rise in tropical zones" should be changed to "found that productivity showed a negative response to temperature in tropical zones due to increasing water stress"?
- L222: "10% **of** non-structural"
- L262,6: "tree" should be "trees".

- Fig. 1: I'm glad to see the newly-excluded grid cells marked in white in this figure. Please add an indication to the legend and/or caption pointing this out.

- L332-3: "vegetation grid cells" doesn't really make sense. Suggest changing "We defined vegetation grid cells as those whose largest component" to "We included grid cells whose largest vegetation component". Also, refer to Fig. A6 here.

- L345: "grids" should be "grid cells".

- L355: "We declare that" is unnecessary and can be deleted.

- L367: "showed" should be "shown".

- L427, 433: "grid" should be "grid cell".

- L428: "dominated" should be "dominant".

- L430-2: "zones" should be "grid cells"… Unless the analysis looks at area (i.e., hectares or whatever), in which case it should say "land area" or something. "Zones" is confusing because it can also be used to refer to latitudinal bands. (I know you're not referring to latitudinal bands because with 10 bands all your results would be multiples of 10%.)

- Fig. 8:

  - Now that you've improved the description, I understand what you were going for with panels B and D. The labels indicating the fraction of grid cells in each category (1.21%, 6.33%, etc.) should actually be changed back to how they were previously. "–1.21" etc. is confusing because it doesn't have the percentage symbol, and incorrect because –1.21% of global area is impossible. Sorry for my confusion before.

- - The caption says that the fractions are of global *area*, but in the text it sounds more like fractions of *grid cells*. (See comment above for L430-2.)
- Figs. 9, 10:
  - X-axis labels should indicate the range of values in each bin. This can be accomplished by either (a) changing each label to be, e.g., "0–0.1," "0.1–0.2", etc. or (b) moving the tick marks so that the tick to the left of a box shows its lower bound and the tick to the right shows its upper bound. I'd prefer (b), personally.
  - Suggest deleting "over the hydrological grid cells (Figure 1)." It's poorly-worded and doesn't really add anything.
- L467: "of historical" should be "of the historical".
- L480: Is "maximum change magnitude of LVBC density" here saying the same thing as "fluctuation range" later? If so, define and use "fluctuation range" here.
- L486: "lived in aridity" should be "in arid".
- L504-8: Where is it demonstrated that "aridity mitigation" is happening in semi-arid zones? This phrasing to me implies that semi-arid regions are becoming moister. I think what you mean is that semi-arid regions are less arid than hyper-arid and arid regions. Suggest rewriting: "Whereas LVBC decreases and WVBC increases in hyper-arid and arid regions (Figs. A7 and A8), causing a downward trend in LVBC:WVBC ratio, semi-arid regions see an increase in LVBC." Note that "in all factorial simulations" in several places in this paragraph is unnecessary; it doesn't contribute anything to the analysis here.
- L508: "semi-arid **regions**"
- L538: Delete "are"
- L541-2: Suggest deleting this sentence. It doesn't add anything explanatory like what I was looking for.
- L547: "more dramatically" should be "relatively more" for clarity.
- L554: Please check whether "zonal" is correct here (referring to latitude bands) or whether "grid cell" should be used instead. (Similar: "zone" at L566.)
- L558: Where does this "over one third" number come from? The weighted average of results from 8b and 8d? I think you should add panels C and E to show the *total* C effects (or maybe put this an Appendix figure).
- L562-4: This sentence is confusing. Suggested rewrite: "This spatially compensatory effect of climate changes is consistent with a previous analysis (Zhu et al. 2016) which found that climate changes explain only 8% of the increasing trend in carbon storage of foliage at a global level but that they dominate the trend over 28.4% of global land area."
- L566-7: This sentence is unnecessary and opinionated; please delete.
- L567: Revert "we suggest" to "our results reveal" or "our results show" or something.
- L579-81: A critical aspect is not just that light competition is high, but that *water limitation (competition) is low*. Indeed, that's what *allows* high competition for light—

trees can grow in close proximity to each other (and thus shade each other) *because* there's enough water to allow each tree as much as it wants. Please include this in your explanation here.

- L592-4: Description of Madani et al. (2020) is still too vague. What exactly did they show? Compare the vagueness here to the excellent summaries you give for Humphrey et al. (2021) and Ma et al. (2021) in the following sentences.

- L599: "process of terrestrial ecosystem" should be "of terrestrial ecosystems".

- L617: You don't have any way of showing that the underestimate of $CO_2$ fertilization would be "slight." Unless you cite some other work showing that the N deposition effect is indeed slight, I would rewrite this to "which should cause an underestimate".

- Fig. A5: Much improved. Last thing: Please edit the legend labels to be "0–10%", "10–20%", etc.

---

## Author Response (AR2)

**Response to reviewer**

\_\_\_\_\_

**Reviewer # Questions and our responses**

We extend our deep appreciation to Reviewer for the constructive comments and suggestions toward improving our paper.

**Reviewer:**

The authors decided not to move journals, which is fine. They have made significant improvements in terms of explaining how water stress theoretically affects allocation, as well as tying their results back to this theory. However, I still have some significant questions about the methodology and confusion about the interpretation of results. As such, I again suggest this manuscript be *reconsidered after major revisions*.

Response: We greatly appreciate the reviewer's detailed and insightful comments which helped us to clarify the logic and presentation of the manuscript. According to suggestions, we have added more detailed description about methods and results in the revised manuscript.

Point-to-point responses to all the comments are given below.

**1. Methods: Hydrological regions**

The aridity zones were determined based on a 115-year average, but it's possible that they could see long-term trends. For example, a gridcell classified as "semi-arid" on average might have been arid at the beginning of the simulation and sub-humid at the end. This is potentially a very important confounding factor, and might explain the sometimes-large variation around  $\Delta=0$  and the resulting weak trends in Figs. 9–10. If a lot of gridcells see shifts like this, it might be necessary to restrict analysis to grid cells that didn't change in terms of aridity, or didn't change much.

Another option might be to let gridcells shift their classification over time. This would require the classification to be based on a rolling mean of the previous, say, 15 or 30 years of climate. So, e.g., a bin in Figs. 9–10 would be "[mean 2015 L/WVBC value for all cells that were in this bin in 2015; i.e., whose 2001–2014 climate fit into this bin] minus [mean 1916 L/WVBC value for all cells that were in this bin in 1916; i.e., whose 1901–1914 climate fit into this bin]." Fig. 11 would show, at each year Y in each subfigure F, "mean Y L/W ratio for all cells that were in class F in Y; i.e., that qualify as class F based on climate in years Y–15 to Y–1." This would be necessary in order to avoid spurious interannual switches in classification, as well as to minimize the effects of a lag in vegetation community response to changing conditions.

Response: Thanks for the detailed and constructive suggestion. We agree with the review that the property of some grid cells was shifted from arid at the beginning of the simulation to sub-humid at the end of the simulation. Based on precipitation and potential evapotranspiration data, we calculated the multiyear average aridity indices at the 1916-1945 period and at the 1986-2015 period, and analyzed the transformation of hydrological conditions from the period of 1916-1945 to the period of 1986-2015 (Figure A4). We added more explanation and figure in the revision as below:

[revised manuscript text omitted]

(see Revision, Page 26)

**2. Pasture exclusion**

I appreciate the authors having made this change to their analyses, as I suggested. However, I'm a bit unclear as to how they actually did it. L338-42 are confusing, but I interpret them as saying: "To minimize confounding effects of livestock grazing, we excluded grassland grid cells with pasture fraction greater than 1%."

If my interpretation is correct, it's hypothetically possible for a "forest" gridcell that's 51% forest and 49% grassland, with LUH2 saying "49% of the area in this gridcell is pasture," and it wouldn't be excluded. This doesn't seem right. Instead, the authors

should exclude any gridcell with a significant pasture fraction, whether it's classified as "grassland" or not. 1% might be too strict, though—maybe something like 10% instead.

Also, make it clear that this exclusion only applies for the NPP comparison.

Response: Thanks for your constructive suggestion. Following your suggestion, we redefined the gird cells affected by livestock grazing where the fraction of managed pasture is greater than 10%. We added more detailed explanation as below:

"When the fraction of managed pasture is over 10%, the grid was considered to be affected by the managed pasture. To reduce the interference effects of livestock grazing, we first removed the grids affected by managed pasture. Then, we map the distribution of natural vegetation zones without anthropogenic disturbance (Figure A6). We declare that this exclusion method is only used for potential NPP comparison." (see Revision, Page 14, Lines 343-347)

**3. Unimodal pattern in Fig. 10**

In my initial review, I raised the question of why  $\Delta$ WVBC decreases along the aridity axis from semi-arid to humid regions. The authors have not actually addressed this, despite an attempt at L556-9. "Drought mitigation promotes the growth of WVBC" okay, sure, that explains why it initially increases from left to right. "humid region with high light competition limits root growth"—but this figure is about WVBC, not LVBC. Why would high light competition in humid regions lead to decreased  $\Delta$ WVBC relative to semi-arid regions? (My thinking is it's because vegetation in these regions is lightlimited: They're not seeing any alleviation of that limitation with climate change.)

Response: Thanks for your constructive comment. Yes, light resources are limited in each grid cell. Humid regions are covered by vegetation with higher plant height. Plants usually face intensified light-competition in humid regions, so plants must invest and allocate as much non-structural carbon as possible into leaf and trunk. This allocation scheme leads to the decreased investment of  $\Delta WVBC$  in wet regions. We added more detailed illustration of the unimodal pattern in the revised manuscript.

"Drought mitigation promotes the growth of WVBC. In sub-humid and humid regions, plants face intensified light-competition and have to invest as much non-structural carbon as possible into leaf and trunk. This allocation scheme leads to the decreased investment of  $\Delta$ WVBC in wet regions." (see Revision, Page 28, Lines 578-581)

**4. Other issues with interpretation: Results**

Some of these suggestions might seem like they're better suited for a Discussion section than a Results section, but I think it would not be good writing to simply list a set of observations with no context, then explain them somewhere later in the paper. The current Discussion section is well structured in the sense that it provides a general overview of the results and then compares to previous literature.

L461-2: It's not that plants stop growing; they're still alive! It's that they don't end up increasing their carbon stocks—i.e., it's so dry that they can't take advantage of higher CO2.

Response: Thanks for your encouraging comments and constructive suggestion. We modified this sentence as below:

"In extreme water stress, the increase of LVBC tends to zero and plants stop increasing their carbon storage." (see Revision, Page 23, Lines 468-469)

5. L462-3: What does it *mean* that there's no obvious difference among the slopes? Response: Thanks for your detailed comment. We have clarified this in the revised manuscript added more detailed explanations in order to help readers understand better as below: "There is no obvious difference in the slopes of fitting curves between factorial simulations, which shows the robustness in the response of LVBC to the change of water stress." (see Revision, Page 23, Lines 469-471)

6. L472-483: What do all these factorial results mean? It is not sufficient to just saying how big the difference is between the minimum and maximum experiment. What do they imply regarding the drivers of partitioning according to your hypotheses? You approach this for LVBC at L475-7, but you don't actually connect the results back to the hypotheses. (And you don't do this at all for WVBC.) Guide the reader!

L472-4: I'm not sure where these numbers come from.

L473: "changed" should be "ranged", I think. You're not comparing a change, you're comparing across a range of aridity classes. Unless I'm misunderstanding—as I said, I don't know where the numbers came from.

Response: Thanks for your constructive comment. We are sorry for these confusions. We have re-written this sentence.

"Figure A7b shows that the maximum change magnitude of LVBC density across all factorial simulation is 1.202 kg C m-2 in the hyper-arid regions for the 1916-2015 period. As shown in Figure A7f, the maximum change magnitude of LVBC density in humid regions is 6.068 kg C m-2 during the same period. In Figure A8b, the maximum change magnitude of WVBC density across all factorial simulation is 0.011 kg C m-2 in the hyper-arid regions during the time of 1916-2015. In Figure A8f, the maximum change magnitude of WVBC density is 0.046 kg C m-2 in humid regions during the same period. Compared with plants lived in aridity regions, plants in humid regions show more dramatic responses to the stimulation from drivers' change. With a lessening of water stress (from hyper-arid to humid region), the response magnitudes of the carbon stock to the changes of climate and CO2 gradually become more noticeable. The robust pattern in the zonal average density of the carbon stock shows that terrestrial water limitations strongly regulate the enhanced magnitude of the carbon stock." (see Revision, Pages 23-24, Lines 480-490)

**7. Fig. 11 (L484-97)**

Again, guide the reader. Here you've done a good job of phrasing the results in a way that connects back to hypotheses ("Under the synergistic effect of drivers and water stress, ... there is a larger proportion of biomass allocated to, and stored in, light-gathering vegetation organs."), but it's unclear how that is evident from the figure. Is it because the blue line is so much higher than the other lines? But why only in sub-humid and humid zones? And there, what does it mean that the ratio goes back down when additional factors are added after S3?

Response: Thanks for your encouraging comments and constructive suggestion. Figure 6 shows that LVBC and WVBC significantly increased in the past hundred years, while the increasing rates of LVBC and WVBC are obviously different (Figures 7d and 11a). Based on the simulated results, we found that LVBC increased faster than WVBC because more non-structural carbon was allocated to leaf and trunk in sub humid and humid zones under the synergistic effect of drivers and water stress. S4 and S5 represent the effect of  $CO_2$  + temperature and the effect of  $CO_2$  + radiation on carbon stocks, separately. Compared with S3, the ratio goes back down in S4 and S5, which indicates that precipitation is the main contributing factors to the change of LVBC/WVBC ratio. We added more detailed explanations in revised manuscript as below:

"Under the synergistic effect of drivers and water stress, the trends of light- and watergathering vegetation carbon stock are upward in the past hundred years (Figure 6). However, there is a difference in the increasing rate between LVBC and WVBC, resulting in a dramatic and complicated fluctuation in global LVBC/WVBC ratio (Figure 11a). The density of LVBC decreases and that of WVBC increases in hyperarid and arid zones for all factorial simulations (Figures A7 and A8). So, the ratio of LVBC and WVBC shows a downward trend in these zones. LVBC in semi-arid regions shows upward tendency in the past years (Figure A7d) because of the aridity mitigation. There is an upward trend in WVBC in semi-arid (Figure A8d). Plants in semi-arid still utilize a tolerance strategy and allocates more non-structural carbon to water-gathering vegetation organ to resist water stress, resulting in the decline of LVBC/WVBC ratio. In humid zones, light- and water-gathering biomass carbon stocks both increased in all factorial simulations (Figures A7 and A8). The proportion of LVBC increases more than that of WVBC for capturing more resources like CO2 and radiation energy, leading to an increase in the LVBC/WVBC ratio. The value of LVBC/WVBC in S3 is higher than that in S4 and S5, which represents that precipitation makes more contributions to the change of LVBC/WVBC ratio among meteorological factors." (see Revision, Page 25, Lines 500-515)

8. L488: What does "variation range of ratio between LVBC and WVBC" mean? Interannual variation? Variation among factorial experiments? Neither seems to match the trend mentioned.

Response: Thanks for the detailed comments. "the variation range of ratio between LVBC and WVBC" was changed to "the fluctuation range of LVBC/WVBC ratio", which represents the response magnitude of LVBC/WVBC ratio to changes in climate and CO2. We have re-written this sentence to help readers understand better.

"From hyper-arid zones to humid zones, the fluctuation range (the difference between maximum value and the minimum value in each factorial simulation) of LVBC/WVBC ratio significantly changes. The fluctuation magnitudes of LVBC/WVBC in humid and hyper-arid zones are greater than that in other hydrological zones. Compared with plants in hyper-arid zones, plants in humid zones exhibit more significant responses to changes in climate and CO2." (see Revision, Pages 24-25, Lines 494-499)

**9. Other issues with interpretation: Discussion**

**L503-29: LVBC and WVBC trends overall**

This is very confusing and disjointed. L514-7 and L527-9 make it seem like plants are tending to shift their allocation from WVBC to LVBC, but then L519-20 ("LVBC...

dominates the long-term trends") and L525-7 seem to suggest the opposite. What's correct? Here's a hint-focus on the ratio. The absolute numbers I think are not very informative about allocation changes, because wood biomass is always so much higher than fine root biomass.

Response: Thanks for your detailed comments. In this study, we found that LVBC significantly increased  $116.18 \pm 2.34$  Pg C, accounting for 97.42% of the total carbon stock increase. WVBC increased  $3.08 \pm 0.14$  Pg C. So, we suggested that the LVBC predominates the spatio-temporal pattern of total carbon storage potential and dominates the long-term trends of vegetation carbon stock at the global scale. Meanwhile, Figure 7d showed that there was a slight upward trend in the ratio of LVBC/WVBC, because LVBC increased more dramatically than WVBC in the past decades. We suggest that plants change the allocation scheme and store more non-structure carbon into LVBC. We have re-written sentences as below:

"LVBC increases 116.18  $\pm$  2.34 Pg C from 1916 to 2015, accounting for 97.42% of the total carbon stock increase (119.26  $\pm$  2.44 Pg C). The long-term trends and spatial pattern of vegetation carbon stock are predominated the variability characteristic of LVBC." (see Revision, Page 27, Lines 537-539)

"During the past hundred years, the ratio of LVBC/WVBC showed a slight upward trend since LVBC increased more dramatically than WVBC." (see Revision, Page 27, Lines 546-547)

10. What does "Compared with WVBC" mean here? Does it mean "116.18  $\pm$ 2.34 Pg C" is  $\Delta$ LVBC minus  $\Delta$ WVBC?

Response: Thanks for your detailed comments. We removed the sentence "Compared with WVBC" in the revision. The "116.18  $\pm$  2.34 Pg C" is the increasing value of LVBC from 1916 to 2015. We added more detailed illustration about "116.18  $\pm$ 2.34 Pg C" as below:

"LVBC increases 116.18  $\pm$  2.34 Pg C from 1916 to 2015, accounting for 97.42% of the total carbon stock increase (119.26  $\pm$  2.44 Pg C)." (see Revision, Page 27, Lines 537-538)

11. L520-2: Yes, but then why is increasing  $\Delta$ WVBC concentrated in high latitudes? The difference here might be something that partitioning theory could explain, or maybe not—maybe it's just a relaxation of climate limitations in the high latitudes that low latitudes never experienced (as possibly suggested at L545-7).

Response: Thanks for your detailed suggestion. Yes, we agree that climate limitations may be able to explain the spatio-temporal patterns of WVBC. We added explanations about the variant patterns of WVBC as below:

"Under the influences of environmental stressors, WVBC increases significantly in boreal latitudes" (see Revision, Page 27, Lines 541-542)

**12. L531-47: Factorial experiments**

This needs to be cleaned up a lot. For one thing, just listing these numbers feels much more like something for the Results rather than the Discussion. Here, you should be focusing on the implications of your results for scientific understanding, and comparing your results to previous literature. For another, the presentation of results is really confusing. For example, radiation doesn't "dominate" the trend at any latitude band. It explains 20.67% of the global variation, though. Explain where these numbers are coming from (Fig. 8b/d, adding the (–) and (+) numbers at the bottom of each). But then maybe I have it wrong—it doesn't make any sense for any of these "fraction of variation explained" numbers to be negative, as they are for, e.g., precip  $\rightarrow$  WVBC (– 2.76%). Do you mean instead that the net influence of precip on WVBC is negative?

Response: Thanks for your detailed comment. In this study, we reveal the effects of drivers' changes on carbon stocks from global and grid cell perspectives, respectively.

We listed some results from Figure 8 to better reveal the discrepancy from two perspectives in the Discussion. The number "-2.76%" represents the negative effects of precipitation on the changes of WVBC from 1916-2015 at the global scale. We added more detailed illustration in revision as below:

"Figure 8c shows there are negative effects and contributions of precipitation on the change in WVBC at the global level (-2.76%, -0.013 g m-2 yr-1)." (see Revision, Page 21, Lines 443-444)

We used the method from Piao et al. (2006, Geophysical Research Letters, 33(23), L23402) to identify the dominated factors of each grid and quantify its contribution, which is shown in Figure 8 b/d. We added more detailed illustration about results to help readers understand better. The Figure 8 in the revised manuscript have been improved.

"While the increase or decrease in the carbon stock may be attributed to more than one driving factor, within any specified grid, the one with the highest positive or negative contribution is the dominated driver that consistently resulted in the highest increase or decrease in the carbon stock for that grid." (see Revision, Page 20, Lines 426-429)

Figure 8. The proportion of change in the vegetation biomass carbon stocks attributed to driving factors. Ratios of the driving factors of  $CO_2$  fertilization effects ( $CO_2$ ), climate change effects (CLI), precipitation (Pre), temperature (Tem), radiation (Rad) for LVBC (a) and WVBC (c) under the five scenarios using the Mann-Kendall and Sen's slope estimator statistical tests. Attribution of LVBC (b) and WVBC (d) dynamics to driving factors calculated as averages along 15° latitude bands. At local scales, the driving factors include  $CO_2$ , Pre, Tem, Rad, and other climate factors (OF). The fraction of global area (%) that is predominantly influenced by the driving factors is showed at the bottom of the bar. The '-' symbol before fraction indicates a negative effect of the driving factor on carbon stock, and vice versa.

(see Revision, Pages 20-21)

We have re-written this sentence in Discussion.

"At the grid cell scale, shown in Figure 8b and 8d, radiation and precipitation dominate the long-term trend of carbon stocks over one third of global grid cells. At the global scale, radiation and precipitation explain approximately 10% of long-term trend in LVBC and WVBC (Figure 8a and 8c). LVBC and WVBC variations driven by precipitation and radiation are ultimately offset by spatially compensatory effects, which dampens the response of the carbon stock to these factors at global scale (Jung et al., 2017). This spatially compensatory effect of climate changes is consistent with previous analyses (Zhu et al., 2016) that climate changes explain 8% of the increasing carbon storage of global foliage, while climate changes dominate the greening trend over 28.4% of the global land." (see Revision, Page 28, Lines 557-564)

13. L531-2: Is this conclusion drawn from Fig. 8? If so, I would rephrase to talk about the amount of variation explained, rather than the amount of increase. Always mention what figure(s) your assertions come from.

Response: Thanks for your detailed comment. Following your suggestion, we reworded the sentence in the revision as below:

"Based on our factorial simulations (Figure 8), the influences of  $CO_2$  fertilization induce the most significant variation of the vegetation carbon stock. In addition, the responses of carbon stocks to the changes of climatic factors are obvious, particularly at the zonal scale." (see Revision, Pages 27-28, Lines 552-554)

14. L535-540: Still need to do a better job of tying these results back to the hypotheses and/or to other explanations. The sentence at L539-40 might be directly relevant to optimal partitioning theory, depending on what it ends up saying after the authors clean up the section.

Response: Thanks for your constructive comment. We added more detailed explanation about this citation to compare our findings with previous explanations.

"This spatially compensatory effect of climate changes is consistent with previous analyses (Zhu et al. 2016) that climate changes explain 8% of the increasing carbon storage of global foliage, while climate changes dominate the greening trend over 28.4% of the global land." (see Revision, Page 28, Lines 562-564)

**15. L578-90: Caveats**

L587-90: This sentence could be interpreted as "We didn't vary N deposition over the experiment," when in reality no N deposition was included at all. This means that the simulated ecosystems would have an incorrectly low amount of N input, leading to incorrectly high amounts of N limitation, leading to an underestimate of CO2 fertilization (because if they're N-limited, they can't take advantage of higher CO2 levels). This is the opposite of what the authors seem to conclude.

Response: Thanks for your constructive comment. We changed the sentence to "which leads to a slight underestimate of the contributions of  $CO_2$  fertilization on biomass production." (see Revision, Page 30, Lines 616-617)

16. To avoid confusion (such as I exhibited in my first review), aridity index axes on all figures with them should include (something like) "drier" at 0 and (something like) "wetter" at 1. Or at least an explanation of this should be in the caption. The axes should also include labels indicating what the ranges are for the different classifications (humid, semi-arid, etc.).

Response: Thanks for your constructive suggestion. Following your suggestion, we added more explanations about the aridity index and hydrological zones in the caption of Figures 9 and 10.

"Categories of hydrological zones include: hyper-arid (AI  $\leq 0.05$ ), arid (0.05 < AI  $\leq 0.2$ ), semi-arid (0.2 < AI  $\leq 0.5$ ), sub-humid (0.5 < AI  $\leq 0.65$ ), and humid (AI > 0.65)." (see Revision, Pages 22-23)

"Categories of hydrological zones include: hyper-arid (AI  $\leq 0.05$ ), arid (0.05 < AI  $\leq 0.2$ ), semi-arid (0.2 < AI  $\leq 0.5$ ), sub-humid (0.5 < AI  $\leq 0.65$ ), and humid (AI > 0.65)." (see Revision, Page 24)

17. L50: "are" should be "is"

Response: Corrected. (see Revision, Page 2, Line 50)

18. L56-8: Citation?

Response: Thanks. We added the citation. (see Revision, Page 2, Line 58)

19. L76: "region" should be "regions"

Response: Corrected. (see Revision, Page 3, Line 76)

20. L82-83: The two parts of this sentence seem to contradict each other.

Response: Thanks for your detailed comment. We added more detailed illustration about this example in order to help readers understand better as below:

"For example, global warming positively stimulates plant productivity (Keenan et al. 2017), while Madani et al. (2020) found that plant productively with water stress show a negative response to temperature rise in tropical zones." (see Revision, Page 3, Lines 82-84)

21. L84: "predictable" should be "predicted"

Response: Corrected. (see Revision, Page 3, Line 85)

22. L104-6: It's not really a proxy; this is just how you define it. (Which is fine, of course!)

Response: Thanks. "a proxy" was changed to "an indicator". (see Revision, Page 4, Line 107)

23. Good clarification of model timesteps

Response: Thanks for your encouraging comment.

24. L175-6: "doesn't include" should be "isn't included"

Response: Corrected. (see Revision, Page 7, Lines 176-177)

25. L207: Define "soffit" (or, ideally, use a simpler word, like "layer").

Response: Thanks, "soffit" was changed to "layer". (see Revision, Page 8, Line 208)

26. L220-1: Not a complete sentence. Suggest changing the period to a comma and deleting "This".

Response: Corrected. (see Revision, Page 8, Lines 221-222)

27. L221: "NSC" not previously defined (should happen at L187).

Response: Thanks. "NSC" was changed to "non-structural carbon" in the revision. (see Revision, Page 8, Line 222)

28. L221-2: Clarify that these organs are not explicitly simulated (unless they are!), and instead are represented as a flux to litter.

Response: Thanks for your constructive suggestion. We added explanation as below: "These organs are not explicitly modelled in SEIB-DGVM." (see Revision, Page 8, Lines 223-224)

29. L254: Delete "in PFTs", maybe? What does it mean?

Response: Thanks. We deleted "in PFTs". (see Revision, Page 9, Line 256)

30. L257-8: GVBC should be LVBC.

Response: Corrected. (see Revision, Page 10, Line 260)

31. L259-63: "wood" should be "woody vegetation".Response: Corrected. (see Revision, Page 10, Lines 261-262)

32. L257-263: *Wmass* should be renamed, e.g. to *Tmass* (T for "tree" [or "trees and shrubs"] instead of W for "woody vegetation"), to avoid confusion with other use of W for "water gathering".

Response: Corrected. (see Revision, Pages 9-10, Lines 259-266)

33. L274-7: The "remarkable effects" comment about wind and relative humidity is a result; don't include it in Methods. Deleting that comment will allow wind and RH to replace "other factors" in the previous sentence.

Response: Thanks. Following your suggestion, we have re-written this sentence as below:

"In order to further quantify the relative contributions of varying atmospheric  $CO_2$  concentrations, precipitation, temperature, radiation, and other factors (wind velocity and relative humidity), we performed six factorial simulations." (see Revision, Pages 10-11, Lines 276-278)

34. L294: "A2, 3" should be "A2–3" or "A2 and A3"

Response: Corrected. (see Revision, Page 11, Line 295)

35. Figs. A2–3: Caption should include experiment labels S2 etc.

Response: Thanks. We rewrote the caption of Figures A2 and A3 as below:

"Figure A2. Potential LVBC trend maps during the period of 1916 to 2015 under different factorial simulations. (a)  $CO_2$  driving factorial simulation (S2); (b)  $CO_2$ +precipitation driving factorial simulation (S3); (c)  $CO_2$ +temperature driving factorial simulation (S4); and (d)  $CO_2$ +radiation driving factorial simulation (S5)." (see Revision, Page 32)

"Figure A3. Potential WVBC trend maps during the period of 1916 to 2015 under different factorial simulations. (a)  $CO_2$  driving factorial simulation (S2); (b)  $CO_2$ +precipitation driving factorial simulation (S3); (c)  $CO_2$ +temperature driving factorial simulation (S4); and (d)  $CO_2$ +radiation driving factorial simulation (S5)." (see Revision, Page 33)

36. L328-9: "It was defined as vegetation grid that the land cover type of this grid is" should be "We defined vegetated grid cells as those whose largest component was". This grammatical correction turns out to also be a simplification, as it then allows the deletion of the sentences at L332-4.

Response: Thanks. We changed this sentence in original manuscript L328-9 and removed the sentences in original manuscript L332-4.

"We defined vegetation grid cells as those whose largest component was evergreen needleleaf forest, evergreen broadleaf forest, deciduous needleleaf forest, deciduous broadleaf forest, mixed forest, closed shrublands, open shrublands, woody savannas, savannas or grasslands." (see Revision, Pages 13-14, Lines 332-335)

37. L331-2: Replace this with a simpler sentence along the lines of "Other grid cells were excluded from our analysis."

Response: Corrected. (see Revision, Page 14, Lines 335-336)

38. L334: Start a new paragraph here, as you're talking about something new.Response: Corrected. (see Revision, Page 14, Line 338)

39. Fig. A4: There should be a clear break in the color bar at whatever threshold you end up using (currently 1%). Also, red-green axes should be avoided, because red-green colorblindness is relatively common

Response: Thanks for your constructive comment. Following your suggestion, we remap the distribution of managed pasture.

Figure A5. Spatial distribution of multi-year average fraction of managed pasture from 2001-2015 at  $0.5 \times 0.5$  arc-degree resolution.

(see Revision, Page 34)

40. Fig. A5: "NNG (no natural vegetation)" should more accurately be something like "NI (not included)". END" should be "ENF".

Response: Thanks. We corrected the legend in Figure A6 (Fig. A5 in original manuscript) as below:

Figure A6. Map of land vegetation without anthropogenic disturbance from MCD12C1 and LUH2. ENF: Evergreen needleleaf forest, EBF: Evergreen broadleaf forest, DNF: Deciduous needleleaf forest, DBF: Deciduous broadleaf forest, MF: Mixed forest, CS: Closed shrublands, OS: Open shrublands, WS: Woody savannas, SA: Savannas, GL: Grasslands, NI: Not included, which means the zone is not covered by vegetation without anthropogenic disturbance.

(see Revision, Page 34)

41. Fig. 4: Pixels that were excluded based on land cover should be colored gray, to distinguish from included pixels with low correlation.

 NP correlation coefficient of MODIS and SEIB-DGVM

 0.8 - 0.6 - 0.4 - 0.2 - 0.1 0 0 11 0.2 0.4 0.6 0.8

Response: Corrected.

**Figure 4. Spatial patterns in the potential NPP correlation coefficients (P

Figure 5. Estimates of the potential vegetation biomass carbon stock from the literature (blue plot), state-of-the-art datasets (red plot) and this study (black line). Datasets are from the following studies: (1)(Erb et al., 2018; Erb et al., 2007), (2)(Bazilevich et al., 1971), (3)(Saugier et al., 2001), (4)(Erb et al., 2018; Bartholome and Belward, 2005), (5)(Olson et al., 1983), (6)(Erb et al., 2018; Pan et al., 2011), (7)(Ajtay et al., 1979), (8)Erb et al., 2018; Ruesch and Gibbs, 2008), (9)(Kaplan et al., 2011), (10)(Shevliakova et al., 2009), (11)(Kaplan et al., 2011), (12)(Pan et al., 2013), (13)(Prentice et al., 2011), (14)(Erb et al., 2018; Erb et al., 2007), (15)(Erb et al., 2018; West et al., 2010), (16)(Hurtt et al., 2011). (see Revision, Page 16)

43. Fig. 6: (a): Delete "Dynamic of" from right Y-axis label.

**Response: Corrected.**

**Figure 6. Global potential biomass carbon stocks of vegetation during the past 100 years.** (a) The evolution of global potential biomass stocks (LVBC+WVBC), along with changes in biomass stocks that can be attributed to the variability and trend of LVBC and WVBC through the twentieth century. The red line represents the monthly value of LVBC, the blue line represents the monthly value of WVBC, and the black line represents the annual value of CO2 concentration. (b, c) Zonal averaged sums of the annual LVBC and WVBC for latitudinal bands during the first decade (1916–1925, red line) and the last decade (2006–2015, blue line) shows the increased carbon stock capacity.

(see Revision, Page 17)

44. Could also refer to Fig. 7 for extra support in this section.

Response: Thanks. We added the citation of Figure 7 in section 3.2 as below:

"The latitudinal bands of increasing annual LVBC are mainly distributed in the tropical and boreal latitudes, which is consistent with Figure 7b." (see Revision, Page 17, Lines 381-382) "There is a single peak in the spatial variation of annual WVBC (Figure 6c and Figure 7c)." (see Revision, Page 18, Line 386)

45. L385: "while they declined"

Response: Corrected. (see Revision, Page 18, Line 390)

46. L387: "decrease" should be "decreasing"

Response: Corrected. (see Revision, Page 18, Line 392)

47. Figs. 9-10: Are Y-axis units per year? It would be easier to relate to other figures if they were total over the simulation.

Response: Thanks. Y-axis units represents the accumulated change value of LVBC or WVBC from 1916 to 2015. We added explanation in the revision.

"As shown in Figures 9 and 10, with the accumulated change of LVBC or WVBC in the period of 1916 to 2015 across the aridity index (i.e., an increase in available water)" (see Revision, Page 23, Lines 464-466)

48. L460: "enhance" should be "increase".

Response: Corrected. (see Revision, Page 23, Line 467)

49. L460-1: That's not really from the factorial simulations; it's obvious from the historical simulation.

Response: Thanks for your constructive suggestion. We added more detailed explanation as below:

"Based on the results of historical simulation (Figure 9), we find a positive relationship between LVBC and aridity index." (see Revision, Page 23, Lines 467-468)

50. L461: "water pressure" should be "aridity".

Response: Corrected. (see Revision, Page 23, Line 468)

51. L462: "different" should be "difference"

Response: Corrected. (see Revision, Page 23, Line 469)

52. L476: Should be "matches", not "matchs"

Response: Thanks. We are very sorry for our incorrect writing and removed it in revised manuscript.

53. L490: "Positive influence" is unclear, and "allocate" should be "allocation"

Response: Corrected.

"Meanwhile, the long-term effects of driver changes have a remarkable influence on this carbon allocation pattern at global level. (Figure 7d)" (see Revision, Page 25, Lines 499-500)

54. L503-29: LVBC and WVBC trends overall. L520: For clarity, say "annual **change** in LVBC"

Response: Corrected. (see Revision, Page 27, Lines 539-540)

55. Great improvements here with regard to comparison to other literature and tying back to theory.

Response: Thanks for your encouraging comment.

56. L552: "region" should be "regions"

Response: Corrected. (see Revision, Page 28, Line 574)

57. L550-2: Refer to figures supporting this (presumably Figs. 9 and 10).

Response: Thanks. We added the citation of Figures 9 and 10 as below:

"These results indicate that vegetation in humid regions is responsible for most of the trend in global LVBC, while plants in semi-arid regions play a dominate global role in controlling the long-term trend in WVBC (Figures 9 and 10)." (see Revision, Page 28, Lines 572-575)

58. L565: Start a new paragraph here, since you're switching from analyzing your results to comparing them with previous literature.

Response: Corrected. (see Revision, Page 29, Line 591)

59. L567-8: Description of Madani et al. (2020) is too vague. "Variable"? In what way, and how does it compare to your results?

Response: Thanks for your detailed comment. We added more explanations and compared my results with previous conclusion, and suggested that terrestrial hydrological conditions significantly affect the carbon cycle process of terrestrial ecosystem.

"Madani et al. (2020) found that changes in water constraints significantly affect the response patterns of ecosystem productivity and net carbon exchange. Humphrey et al. (2021) found that increasing water stress limits the response magnitude of carbon uptake rates through a down-regulation of stomatal conductance and suggested that land carbon uptake is driven by temperature and vapour pressure deficit effects that are controlled by terrestrial water availability. Ma et al. (2021) found that plants increase investment into building roots in arid region because the extent of water limitation there is exacerbated by global warming. Terrestrial hydrological conditions significantly affect the carbon cycle process of terrestrial ecosystem, including carbon uptake, allocation, and stock." (see Revision, Page 29, Lines 592-600)

60. L578-90: Caveats. L584: I would say "apparent underestimate," as the numbers from SEIB-DGVM aren't wrong—they're measuring something different.

Response: Corrected. (see Revision, Page 29, Line 611)

61. L592-601: Conclusion. L594: "to" should be "vs." or "and"Response: Corrected. (see Revision, Page 30, Line 621)

Thanks again for your time and efforts put on this manuscript.

---

## Author Response (AR3)

**Response to reviewer**
* * *
**Reviewer # Questions and our responses**

**We extend our deep appreciation to Reviewer for the constructive comments and suggestions toward improving our paper.**

**Reviewer:**

In the latest (second) revision, the authors have done a good job of responding to my comments. They improved their methods significantly by (a) excluding grid cells that changed aridity classes and (b) refining their pasture-cell exclusion rule. Additionally, they have made great improvements in terms of explaining their analyses. As a result, I suggest this paper be ***published after minor revisions***.

Response: Thanks for your encouragement. We extend our deep appreciation to you for constructive comments and suggestions toward improving our paper.

Point-to-point responses to all the comments are given below.

1. Pasture vs. rangeland

You masked based on the LUH2 "managed pasture" layer, but most grazing land by area is actually rangeland—see below for 2010. Please consider the exclusion based on the total pasture+rangeland area. (I'm sorry, I should have caught this in the first revision!) Alternatively, there may be an argument that rangeland doesn't need to be excluded, as might be considered less intensely grazed. If you want to go that route, mention it in the text

Response: Thanks for the detailed and constructive suggestion. We agree with that rangeland is considered less intensely grazed, while there is a strong anthropogenic disturbance in managed pasture. To reduce the interference effects from human activity, we only removed the grids with the fraction of managed pasture over 10% in this study. We added more detailed explanation as below:

"There is a weak anthropogenic disturbance in rangeland, while managed pasture is intensely grazed by livestock. To remove pasture area with strong anthropogenic disturbance, we obtained land-use forcing data from Land-Use Harmonization (LUH2) to map the distribution of managed pasture data from 2001 to 2015 (Hurtt et al., 2020)." (See Revision, Page 14, Lines 338-341)

2.  Include temperature and "other factors" as "climate factors"

At L557-64, the authors seem to consider only precipitation and radiation as "climate change" factors. The numbers from that result in a pretty good correspondence to the results from Zhu et al. (2016), but I would be surprised to learn that those authors included only precipitation and radiation in their analyses. Later in that paragraph, the authors talk about temperature, but it's unclear why it was not included before. Finally, "other factors" (wind speed and relative humidity) are not mentioned at all, but these are also climate factors. The authors should rewrite this paragraph to include all climate factors together in the initial analysis. (The final sentence is a good summary but should also mention "other factors.")

Response: Thanks for the constructive suggestion. We rewrote this paragraph to include all climate factors and added more explanation about the "other factors" in the revised manuscript.

"At the grid cell scale, as shown in Figures 8b and 8d, temperature, radiation, precipitation, and other climate factors (humidity and wind speed) dominate the long-term trend of carbon stocks over two thirds of global grid cells. At the global scale, climate factors explain 17.55% and 10.72% of long-term trend in LVBC and WVBC,

respectively (Figures 8a and 8c). LVBC and WVBC variations driven by climate factors are ultimately offset by spatially compensatory effects, which dampens the response of the carbon stock to these factors at the global scale (Jung et al., 2017). Thus, contributions of precipitation and radiation to the variability of LVBC and WVBC are relatively low at the global scale, and the effects of humidity and wind speed on global carbon stock are minor. This spatially compensatory effect of climate changes is consistent with a previous analysis (Zhu et al. 2016) which found that climate changes explain only 8% of the increasing trend in carbon storage of foliage at a global level but that they dominate the trend over 28.4% of global land area. Results show that trends in temperature drive historical long-term trends in the potential carbon stocks, with faster increases and considerable variation occurring by grid cell. Thus, our results reveal that temperature dominates the long-term trends of carbon stock among climatic drivers, while a relatively strong compensatory effect exists in the global change in the carbon stock induced by precipitation, radiation, humidity, and wind speed." (See Revision, Page 28, Lines 557-571)

3. "Grids"

In almost all instances, the authors should replace "grid(s)" with "grid cell(s)." "Grid" is more appropriate when describing the overall setup (e.g. "grid resolution" is fine), but for referring to individual 0.5° boxes, "grid cell" is what should be used. In "Minor suggestions and technical corrections," I've noted some places this should be fixed, but not all places.

Response: Corrected.

4. L82-4: This sentence is still confusing. "Global warming" seems to speak directly to temperature, but Keenan et al. (2017) found that slower temperature growth meant MORE C sequestration on land (due to lower ecosystem respiration). The Madani et al.

(2020) bit is weird as well. Maybe "found that plants productively with water stress show a negative response to temperature rise in tropical zones" should be changed to "found that productivity showed a negative response to temperature in tropical zones due to increasing water stress"?

Response: Thanks for the suggestion. We corrected this paragraph following review's suggestion. (see Revision, Page 3, Lines 83-84)

5. L222: "10% **of** non-structural"

Response: Corrected. (see Revision, Page 8, Line 222)

6. L262,6: "tree" should be "trees".

Response: Corrected. (see Revision, Page 10, Line 262 and Line 266)

7. Fig. 1: I'm glad to see the newly-excluded grid cells marked in white in this figure. Please add an indication to the legend and/or caption pointing this out.

Response: Thanks, we added more explanation of the white grid cells in the caption of Figure 1.

"The white grid cell was not assigned hydrological category." (see Revision, Page 12, Figure 1)

8. L332-3: "vegetation grid cells" doesn't really make sense. Suggest changing "We defined vegetation grid cells as those whose largest component" to "We included grid cells whose largest vegetation component". Also, refer to Fig. A6 here.

Response: Corrected. (see Revision, Page 14, Lines 332-333 and Page 34, Line 639)

9. L345: "grids" should be "grid cells".

Response: Corrected. (see Revision, Page 14, Line 346)

10. L355: "We declare that" is unnecessary and can be deleted.

Response: Corrected. (see Revision, Page 14, Line 347)

11. L367: "showed" should be "shown".

Response: Corrected. (see Revision, Page 16, Line 368)

12. L427, 433: "grid" should be "grid cell".

Response: Corrected. (see Revision, Page 20, Line 428 and Line 435)

13. L428: "dominated" should be "dominant".

Response: Corrected. (see Revision, Page 20, Line 429)

14. L430-2: "zones" should be "grid cells"… Unless the analysis looks at area (i.e., hectares or whatever), in which case it should say "land area" or something. "Zones" is confusing because it can also be used to refer to latitudinal bands. (I know you're not referring to latitudinal bands because with 10 bands all your results would be multiples of 10%.)

Response: Corrected. (see Revision, Page 20, Lines 433-434)

15. Fig. 8:

Now that you've improved the description, I understand what you were going for with panels B and D. The labels indicating the fraction of grid cells in each category (1.21%, 6.33%, etc.) should actually be changed back to how they were previously. "–1.21" etc. is confusing because it doesn't have the percentage symbol, and incorrect because – 1.21% of global area is impossible. Sorry for my confusion before.

The caption says that the fractions are of global area, but in the text it sounds more like fractions of grid cells. (See comment above for L430-2.)

[Figure]

**Figure 8. The proportion of changes in vegetation biomass carbon stocks attributed to driving factors.** Ratios of the driving factors of $CO_2$ fertilization effects ($CO_2$), climate change effects (CLI), precipitation (Pre), temperature (Tem), radiation (Rad) for LVBC (a) and WVBC (c) are calculated by the Mann-Kendall and Sen's slope estimator statistical tests. Attribution of LVBC (b) and WVBC (d) dynamics to driving factors calculated as averages along 15° latitude bands. At the local scale, the driving factors include $CO_2$, Pre, Tem, Rad, and other climate factors (OF). The fraction of global grid cells (%) that is predominantly influenced by the driving factors is showed at the bottom of the bar. The '-' symbol before fraction indicates a negative effect of the driving factor on carbon stock, and vice versa.

16. Figs. 9, 10:

X-axis labels should indicate the range of values in each bin. This can be accomplished by either (a) changing each label to be, e.g., "0–0.1," "0.1–0.2", etc. or (b) moving the tick marks so that the tick to the left of a box shows its lower bound and the tick to the right shows its upper bound. I'd prefer (b), personally.

Suggest deleting "over the hydrological grid cells (Figure 1)." It's poorly-worded and doesn't really add anything.

Response: Corrected.

[Figure]

**Figure 9. Relationships of the incremental change between AI and LVBC.** Magnitude of change in LVBC in the historical scenario S1 (a), $CO_2$ in scenario S2 (b), $CO_2$ + precipitation in scenario S3 (c), $CO_2$ + temperature in scenario S4 (d), and $CO_2$ + radiation in scenario S5 (e). The range of the box is 25%-75% of values; the range of the whiskers is 10%-90% of values; the small red square is average value; the red line is the median line; and the black line is the fitted curve. Positive value of the Y axis represents the magnitude of increased LVBC from 1916 to 2015 under

water-limitations conditions, and vice versa. AI of grid cells is calculated by multiyear average precipitation and multiyear average potential evapotranspiration in the period of 1916-2015. Categories of hydrological zones include: hyper-arid (AI $\leqslant$ 0.05), arid (0.05 < AI $\leqslant$ 0.2), semi-arid (0.2 < AI $\leqslant$ 0.5), sub-humid (0.5 < AI $\leqslant$ 0.65), and humid (AI > 0.65).

[Figure]

**Figure 10. Relationships of the incremental change in AI and WVBC**. Magnitude of change in WVBC in the historical scenario S1 (a), $CO_2$ in scenario S2 (b), $CO_2$ + precipitation in scenario S3 (c), $CO_2$ + temperature in scenario S4 (d), and $CO_2$ + radiation in scenario S5 (e). The range of the box is 25%-75% of values; the range of the whiskers is 10%-90% of values; the small red square is average value; the red line is the median line, and the black line is the fitted curve. Positive value of the Y axis represents the magnitude of increased WVBC from 1916 to 2015 under water-limitations conditions, and vice versa. AI of grid cells is calculated by multiyear average precipitation and multiyear average potential evapotranspiration in the period of 1916-2015. Categories of hydrological zones include: hyper-arid (AI $\leqslant$ 0.05), arid (0.05 < AI $\leqslant$ 0.2), semi-arid (0.2 < AI $\leqslant$ 0.5), sub-humid (0.5 < AI $\leqslant$ 0.65), and humid (AI > 0.65).

17. L467: "of historical" should be "of **the** historical".

Response: Corrected. (see Revision, Page 23, Line 468)

18. L480: Is "maximum change magnitude of LVBC density" here saying the same thing as "fluctuation range" later? If so, define and use "fluctuation range" here.

Response: Thanks. The meaning of "maximum change magnitude of LVBC density" is same to "fluctuation range". We corrected it and added the define of "fluctuation range".

"Figure A7b shows that the fluctuation range (the difference between maximum value and minimum value in each factorial simulation) of LVBC density across all factorial simulation is 1.202 kg C m$^{-2}$ in the hyper-arid regions for the 1916-2015 period. As shown in Figure A7f, the fluctuation range of LVBC density in humid regions is 6.068 kg C m$^{-2}$ during the same period." (see Revision, Page 23, Lines 481-484)

19. L486: "lived in aridity" should be "in arid".

Response: Corrected. (see Revision, Page 23, Line 488)

20. L504-8: Where is it demonstrated that "aridity mitigation" is happening in semi-arid zones? This phrasing to me implies that semi-arid regions are becoming moister. I think what you mean is that semi-arid regions are less arid than hyper-arid and arid regions. Suggest rewriting: "Whereas LVBC decreases and WVBC increases in hyper-arid and arid regions (Figs. A7 and A8), causing a downward trend in LVBC:WVBC ratio, semiarid regions see an increase in LVBC." Note that "in all factorial simulations" in several places in this paragraph is unnecessary; it doesn't contribute anything to the analysis here.

Response: Corrected. (see Revision, Page 25, Line 505-507)

21. L508: "semi-arid **regions**"

Response: Corrected. (see Revision, Page 25, Line 510)

22. L538: Delete "are"

Response: Corrected.

23. L541-2: Suggest deleting this sentence. It doesn't add anything explanatory like what I was looking for.

Response: Corrected.

24. L547: "more dramatically" should be "relatively more" for clarity.

Response: Corrected. (see Revision, Page 27, Line 547)

25. L554: Please check whether "zonal" is correct here (referring to latitude bands) or whether "grid cell" should be used instead. (Similar: "zone" at L566.)

Response: Corrected. (see Revision, Page 27, Line 554 and Page 28, Line 569)

26. L558: Where does this "over one third" number come from? The weighted average of results from 8b and 8d? I think you should add panels C and E to show the total C effects (or maybe put this an Appendix figure).

Response: Thanks for the comment. Results show that climate factors dominated the variability of LVBC over 92.72% of the grid cells (Figure 8b) and dominated the

variability of WVBC over 72.40% of the grid cells (Figure 8d) in the past years. In the revised manuscript, we concluded that temperature, radiation, precipitation and other climate factors (humidity and wind speed) dominate the long-term trend of carbon stocks over two thirds of global grid cells. We think that readers can be able to understand the total contributions of climate factors to the variability of carbon stock at the grid cell scale based on Figure 8b and 8d. Meanwhile, we think that four numbers are too few to draw an Appendix figure. Thus, figures are not added in the revised manuscript.

"At the grid cell scale, as shown in Figure 8b and 8d, temperature, radiation, precipitation and other climate factors (humidity and wind speed) dominate the long-term trend of carbon stocks over two thirds of global grid cells." (see Revision, Page 28, Lines 557-559)

27. L562-4: This sentence is confusing. Suggested rewrite: "This spatially compensatory effect of climate changes is consistent with a previous analysis (Zhu et al. 2016) which found that climate changes explain only 8% of the increasing trend in carbon storage of foliage at a global level but that they dominate the trend over 28.4% of global land area."

Response: Corrected. (see Revision, Page 28, Lines 564-567)

28. L566-7: This sentence is unnecessary and opinionated; please delete.

Response: Corrected.

29. L567: Revert "we suggest" to "our results reveal" or "our results show" or something.

Response: Thanks, "we suggest" was changed to "our results reveal". (see Revision, Page 28, Line 569)

30. L579-81: A critical aspect is not just that light competition is high, but that water limitation (competition) is low. Indeed, that's what allows high competition for light— trees can grow in close proximity to each other (and thus shade each other) because there's enough water to allow each tree as much as it wants. Please include this in your explanation here.

Response: Thanks for the constructive suggestion, we added more explanation about water competition as below:

"In sub-humid and humid regions, plants face low water limitations and intensified light-competition and have to invest as much non-structural carbon as possible into leaf and trunk." (see Revision, Page 28, Lines 581-582)

31. L592-4: Description of Madani et al. (2020) is still too vague. What exactly did they show? Compare the vagueness here to the excellent summaries you give for Humphrey et al. (2021) and Ma et al. (2021) in the following sentences.

Response: Thanks for the comment, we added more explanation about water competition as below:

"Based on observations from satellite remote sensing, Madani et al. (2020) found that the constraining impact of water limitation determines whether global ecosystem productivity responds positively or negatively to the changes in climate factors." (see Revision, Page 29, Lines 594-596)

32. L599: "process of terrestrial ecosystem" should be "of terrestrial ecosystems".

Response: Corrected. (see Revision, Page 29, Line 602)

33. L617: You don't have any way of showing that the underestimate of $CO_2$ fertilization would be "slight." Unless you cite some other work showing that the N deposition effect is indeed slight, I would rewrite this to "which should cause an underestimate".

Response: Corrected. (see Revision, Page 30, Line 620)

34. Fig. A5: Much improved. Last thing: Please edit the legend labels to be "0–10%", "10–20%", etc.

Response: Corrected.

[Figure]

Figure A5. Spatial distribution of multi-year average fraction of managed pasture from 2001-2015 at 0.5 × 0.5 arc-degree resolution.

(see Revision, Page 34)

**Thanks again for your time and efforts put on this manuscript.**